# *EgoFact*: A Benchmark for Multi-Hop Multimodal Retrieval-Augmented Generation

## Abstract

Retrieval-Augmented Generation (RAG) has emerged as a powerful approach to improve large language models (LLMs) by grounding their outputs in external knowledge. However, progress in the multimodal domain remains limited, largely due to the lack of suitable benchmarks. Existing multimodal corpora are often built by merging unimodal datasets, which rarely support queries requiring multi-hop reasoning and thus reduce most tasks to single-modality, one-hop retrieval. To address this gap, we introduce **EgoFact**, the first benchmark explicitly designed for multi-hop reasoning across visual and textual corpora. Success on *EgoFact* requires models to retrieve and integrate evidence spanning multiple modalities. We systematically evaluate existing RAG systems and uncover fundamental limitations in multimodal evidence integration and reasoning. Motivated by these findings, we propose a localization-first framework for cross-modal video reasoning that enables more precise evidence grounding and substantially improves reasoning accuracy. Extensive experiments demonstrate the effectiveness of our approach, establishing new state-of-the-art results on multimodal RAG tasks. Together, the benchmark and framework lay a foundation for advancing research in this emerging area and for building more reliable multimodal reasoning systems.

## 1 Introduction

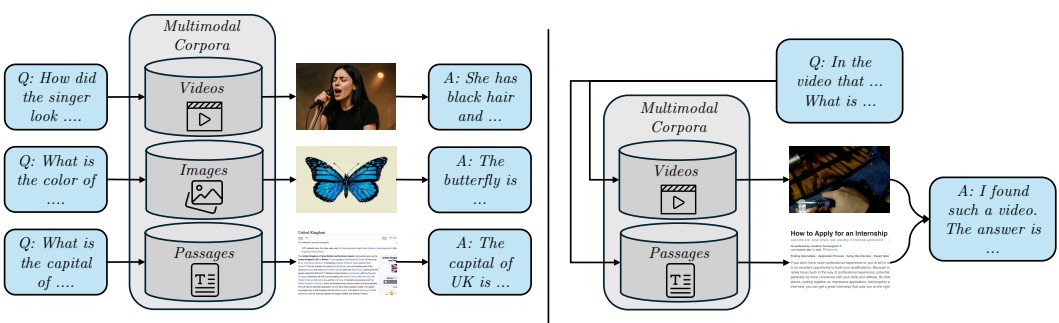

Figure 1: Illustration of multi-hop reasoning in RAG. *Left:* Existing multimodal benchmarks include multiple modalities in their corpora but can often be solved by retrieving and reasoning over a single modality. *Right:* Our proposed benchmark, *EgoFact*, features queries that require integrating evidence across modalities through multi-hop reasoning to arrive at the correct answer.

Large language models (LLMs) have transformed information access and content generation, yet they remain prone to hallucinations and struggle to reason beyond their training data (Zhang et al., 2023; Arora et al., 2023). Retrieval-augmented generation (RAG) addresses these limitations by grounding model outputs in external evidence. While RAG has achieved remarkable progress in text-based domains across diverse applications (Sarthi et al., 2024; Wang et al., 2024; 2025b; Ng et al., 2025; Wiratunga et al., 2024; Wang et al., 2025a), its extension to other modalities—particularly video—remains underexplored. Yet many real-world queries demand precisely this capability: integrating visual observations from video with complementary textual knowledge.

Consider a simple example: a cooking assistant determining *"Did I already add salt?"* can resolve this by grounding its reasoning purely in the video timeline. By contrast, an egocentric video showing

a person using a specific tool to tighten bolts reveals the tool's appearance, but answering *"What safety precautions should be followed when using this tool?"* requires consulting external textual sources such as manuals or *WikiHow*. These examples highlight the spectrum of challenges: while some questions depend only on visual understanding, many require multi-hop reasoning that integrates video evidence with external knowledge.

The need for multimodal reasoning arises naturally in everyday contexts. People increasingly turn to instructional videos, short-form content, and augmented reality (AR) interfaces for information. In these settings, an AI system must not only interpret events in the video but also connect them to external knowledge. These trends underscore the importance of multimodal, multi-hop RAG for real-world applications such as education, assistive technologies, and robotics.

Despite growing interest, existing benchmarks only partially capture the challenges outlined above: the need to reason directly over raw video while integrating external textual knowledge. Luo et al. (2024) focus on intra-video reasoning supported by auxiliary signals such as OCR and ASR. Jeong et al. (2025) study retrieval over large video corpora but rely on transcripts that closely mirror video content. Ren et al. (2025) emphasize cross-video reasoning with LongerVideos, yet their benchmark remains unimodal and text-rich. Yeo et al. (2025) propose UniversalRAG with routing across modalities, but their corpora are assembled from unimodal datasets and their video tasks largely reformulated into text. Collectively, these works advance the field but either extend existing resources or treat modalities in isolation. As a result, current benchmarks fall short of capturing the multimodal, multi-hop reasoning required for authentic video understanding.

To address these limitations, we introduce *EgoFact*, a benchmark for evaluating RAG systems on multi-hop reasoning that integrates video with textual knowledge. Built on egocentric video paired with external resources like *WikiHow*, it enables rigorous assessment of how models retrieve, align, and reason across modalities. At its core, *EgoFact* features queries of two types: *single-hop*, answerable directly from video alone, and *two-hop* that link observed events in video with procedural or factual knowledge in text. To capture different reasoning demands, queries span two categories: *local* (single-event object-focused) and *temporal* (event-ordering across activities). Together, these dimensions capture realistic multimodal reasoning scenarios that existing testbeds overlook. Finally, the use of egocentric videos grounds the benchmark in everyday experiences, while *WikiHow* provides procedural, step-by-step knowledge that supports faithful evaluation of cross-modal retrieval and reasoning.

Beyond benchmark construction, evaluating existing VideoRAG methods on *EgoFact* reveals a critical weakness: *retrieval alone is not enough.* Even when the correct clips are retrieved, models often fail to leverage them effectively, producing hallucinations or inconsistent answers. Strikingly, these failures arise even on questions that are trivial for humans given the same clips, underscoring that the real bottleneck lies not in retrieval but in visual grounding. Current systems operate at a coarse granularity, treating entire clips as evidence rather than isolating the precise snippets that contain the answer.

To overcome this deficiency, we propose a *localization-first framework* that identifies and grounds reasoning in the most relevant video segments within retrieved clips. By directing attention to these informative regions, the method reduces spurious generation and improves answer accuracy. For example, rather than processing an entire 30-second cooking clip, the model isolates the 5-second snippet capturing an event (such as when milk is added), enabling more precise and reliable reasoning.

Ultimately, by ensuring that answers are anchored both in what users see and in reliable external sources, *EgoFact* lays the groundwork for building AI systems that can interact more naturally with human experiences. This work makes three key contributions: (1) *EgoFact*, the first benchmark for retrieval-augmented reasoning over egocentric, action-oriented video with both single-hop and two-hop multimodal queries; (2) a *systematic diagnosis* showing that existing video-RAG systems underperform not because of retrieval, but due to weak visual grounding; and (3) a *localization-first method* that grounds reasoning in relevant video snippets, yielding notable improvements.

## 2 ON MULTI-HOP MULTIMODAL RETRIEVAL

Real-world information-seeking problems are rarely solvable in a single step or from a single source; instead, they demand combining evidence scattered across heterogeneous modalities—text, images,

videos, or tables—through multiple reasoning steps. Consider a scenario where a person is navigating a new city with a wearable camera. To answer the question, *"Which building is this, and when was it built?"*, a system must connect visual recognition of the building with textual knowledge such as Wikipedia entries, effectively linking evidence across modalities. These challenges highlight two complementary directions for advancing retrieval-augmented generation: the need for *multi-hop reasoning*, where answers are constructed by chaining together multiple pieces of evidence, and the need for handling *rich contexts*, where relevant signals are embedded in long, noisy, and multimodal streams. Below we discuss each of these challenges.

**RAG on Multiple Hops.** Many real-world queries require multi-hop reasoning, where systems must chain together evidence distributed across different sources. This idea has been extensively explored in text-based settings, with benchmarks such as *Hot-potQA* (Yang et al., 2018) and *MuSiQue* (Trivedi et al., 2022) explicitly designed to evaluate models' ability to retrieve and integrate information from multiple documents. However, to the best of our knowledge, these efforts remain largely confined to *textual corpora*, leaving open the challenge of extending multi-hop reasoning to multimodal domains.

Table 1: Query statistics of *Ego-Fact*, spanning hop count (1-hop vs. 2-hop) and grounding type (Local vs. Temporal). Each cell corresponds to one query family; in total the dataset contains 396 queries.

| Grounding | 1-hop | 2-hop |
|-----------|-------|-------|
| Local     | 113   | 73    |
| Temporal  | 119   | 91    |

**RAG on Rich Contexts.** Another fundamental challenge for RAG arises in *rich contexts*, where evidence is embedded beyond textual corpora. Research has explored RAG in various modalities, including images (Hu et al., 2025), videos (Ren et al., 2025; Luo et al., 2024; Jeong et al., 2025), tabular data (Joshi et al., 2024), *etc*. These efforts typically focus on single-hop retrieval, where the model retrieves a single piece of evidence from a specific modality to answer a query.

To advance toward real-world applications, we introduce *EgoFact*, a dataset designed to evaluate RAG systems on their ability to perform both single and multi-hop reasoning across multimodal corpora. By integrating egocentric videos with complementary textual corpora, *EgoFact* presents unique challenges that require models to understand context, actions, and interactions from both visual and textual cues.

## 2.1 VIDEO CORPORA CONSTRUCTION

Egocentric recordings are particularly valuable. Unlike third-person instructional or documentary videos used in prior VideoRAG benchmarks, egocentric videos capture fine-grained, first-person perspectives of human activities. This viewpoint naturally reflects situated, real-world contexts and foregrounds objects and actions most relevant to the wearer, making it especially suitable for studying grounded multimodal reasoning. We base our selection on the *Ego4D* dataset (Grauman et al., 2022), focusing specifically on the subset annotated in *Ego4DSounds* (Chen et al., 2024), which offers reliable event-level labels to help curate videos with grounded actions. Importantly, these annotations are used only for dataset curation and are not included as part of the benchmark itself. To manage dataset scale, we further restrict the pool to videos from a set of *pre-defined scenarios* directly relevant to common real-world applications (e.g., daily household tasks, cooking, or tool usage). This selection ensures that the resulting dataset captures both the multimodal reasoning challenges and the domain constraints central to day-to-day use cases.

Each selected video is segmented into 30-second clips. After applying filtering and selection (details in C), we obtain a curated set of 127 representative clips. We then generate a caption for each clip using *GPT-4o-mini* to support later query construction. These clips constitute the backbone of the video corpora in *EgoFact*, providing a controlled yet realistic setting for evaluating cross-modal retrieval and reasoning.

## 2.2 QUERIES DESIGN

To evaluate multimodal reasoning under controlled conditions, *EgoFact* queries are organized along two complementary axes: the number of hops (*1-hop* vs. *2-hop*) and the grounding type (*local* vs. *temporal*). Their combination yields four query families, posed as multiple-choice selection tasks. The overall generation workflow is illustrated in fig. 2, with concrete examples provided in table 4.

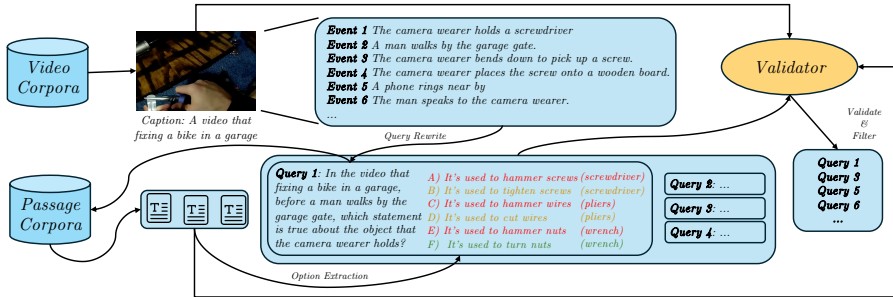

Figure 2: Overall design of the query construction in *EgoFact*. Each query follows a structured pattern: *A, C,* and *E* are counterfactuals (red), with *A* and *C* tied to distractor objects. *B* and *D* are factual statements about these distractors (orange). Only *F* corresponds to the correct object and fact (green). The events list (1–6) is derived from Ego4DSounds annotations, used solely for dataset curation and not included in the benchmark.

**1-hop vs. 2-hop queries.** In 1-hop queries, the answer is derived directly from the video by recognizing an object involved in a specific event. In 2-hop queries, the model must identify the relevant object in the video and also retrieve a factual attribute of that object from an external text passage. To structure this, multiple-choice options are grouped into three object–passage pairs: *(A,B)*, *(C,D)*, and *(E,F)*. Each pair corresponds to a distinct object and its associated passage. Within each pair, option one (e.g., *A, C, E*) states a counter-factual claim about the object, while option two (e.g., *B, D, F*) states the factual one. The first two pairs *(A,B)* and *(C,D)* are distractors referring to other objects, while only the last pair *(E,F)* corresponds to the correct object shown in the video. Within that final pair, *E* gives a false property and *F* gives the true one. Here, each true or counter-factual option is generated by an LLM conditioned on the passage from WikiHow. To construct the textual side of the corpora, we aggregate all passages that appear in these queries, ensuring a consistent pool of evidence for retrieval. This design ensures that solving 2-hop queries requires not just object recognition but also cross-modal factual reasoning.

**Local vs. temporal queries.** Local queries are tied to a single event within a clip, asking about the object directly involved in that event. Temporal queries, in contrast, exploit event ordering: one event serves as a temporal anchor while another provides the target of the question. Note that temporal queries in *EgoFact* always use two consecutive events to form the anchor–target pair. For example, a temporal query may ask, "Before a woman plays with a baby, what object does she place onto the neck?"—requiring the model to resolve sequence as well as grounding.

**Reliability check of queries.** To ensure the reliability of the benchmark, each generated query is validated with three complementary conditions. 1) the query must be *valid*: when the ground-truth event information is provided, an LLM should be able to produce the correct answer. 2) it must be *non-trivial*: given only the query without the supporting event, the LLM should *not* be able to guess the correct answer based on common-sense knowledge. 3) the query must be *non-ambiguous*: the same query should not admit multiple plausible answers across the target document and other contextually similar documents. Queries that fail any of these checks are discarded. This triple criterion ensures that each query in *EgoFact* is answerable only when grounded in the correct evidence, remains challenging without it, and avoids ambiguity across related contexts. The detailed query construction pipeline is presented in fig. 2, and further described in appendix C.2.

Finally, we generate a total of 396 queries spanning the four query families, as shown in table 1.

## 3 EMPIRICAL INSIGHTS ON *EgoFact*

In this section, we assess the effectiveness of existing RAG systems on *EgoFact* by adapting a state-of-the-art video-based RAG framework and evaluating its performance under our benchmark design. Our purpose is to understand where current methods succeed and where they fail, thereby identifying the key bottlenecks for multimodal, multi-hop reasoning. To this end, we analyze both retrieval performance and how well retrieved evidence is understood.

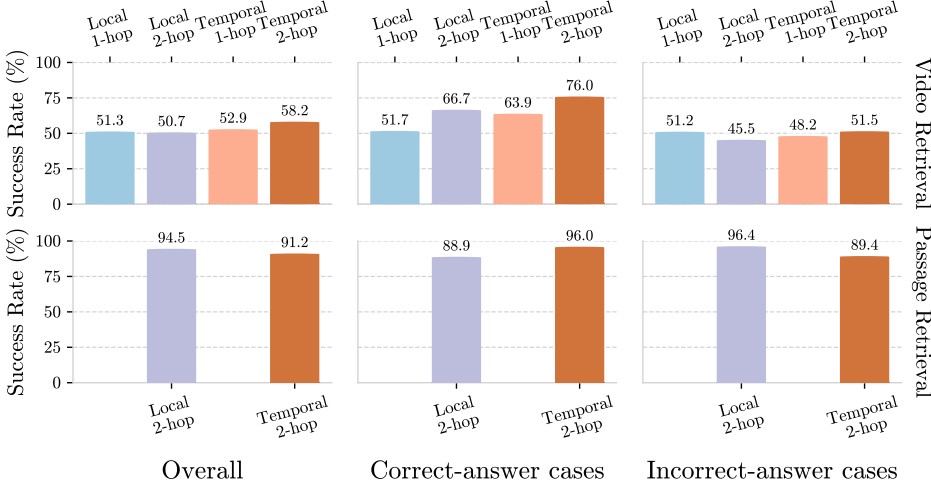

Figure 3: Retrieval performance across video and passage tasks. The first row shows *Video Retrieval* results, and the second row shows *Passage Retrieval* results, aligned with their corresponding 2-hop categories. We observe that (1) passage retrieval maintains a consistently high success rate, (2) video retrieval success rate is generally in the 50%–60% range, and (3) even in the *Incorrect-answer cases*, video retrieval still achieves around 50% success rate, suggesting that the retrieved videos are often reasonably relevant.

## 3.1 EVALUATION SETUP

For evaluation, we focus on VideoRAG (Ren et al., 2025), as it was found to perform better than other baselines on our benchmark. We begin by briefly describing the original framework.

**Framework of Ren et al. (2025).** The original framework is designed for single-hop, video-based RAG tasks. It first uses *MiniCPM* to generate video descriptions, which are stored as a video description corpus. Given a query, the system retrieves related descriptions from this corpus, then performs video retrieval through two branches: (1) a visual embedding–based branch using *ImageBind* (Girdhar et al., 2023), and (2) an entity-based branch that extracts entities from the video descriptions with *MiniCPM* and retrieves videos by entity matching. The retrieved video segments are then captioned with query-guided video captioning, again using *MiniCPM*. Finally, the pre-generated video descriptions and the new captions are concatenated with the query and passed to *MiniCPM* to produce the final answer.

We adapted this framework to fit *EgoFact*. Specifically: (1) we replace the original *MiniCPM* video captioning backbone with *GPT-4o-mini* (OpenAI et al., 2024) for improved query-guided video captioning; (2) we remove the pre-captioned video descriptions and disable the entity-based video retrieval branch, as both components were found to be ineffective for *EgoFact*; (3) given a query that requires information from passage corpora, besides retrieving video segments, we perform an additional passage retrieval step from the text corpora (as described in section 2), where we retrieve the top-6 passages per query; and (4) the final answer is generated from the concatenated video and text evidence. This workflow is illustrated in the upper part of fig. 7, while hyperparameters and prompts are provided in appendix D.2.

## 3.2 RETRIEVAL PERFORMANCE

We examine retrieval performance, decomposed into video retrieval and passage retrieval. For both tasks, we measure *success rate* as the proportion of queries for which the correct item is retrieved. We further categorize results into three groups: *Overall* (all queries), *Correct-answer cases* (queries where the final answer is correct), and *Incorrect-answer cases* (queries where the final answer is incorrect). This breakdown helps identify whether retrieval errors are driving overall failures. The results are summarized in fig. 3. We find some key observations as follows:

**Passage retrieval is reliable.** Across all 2-hop categories, passage retrieval maintains consistently high success rate (generally above 90%), with little variation between the *Correct-answer* and

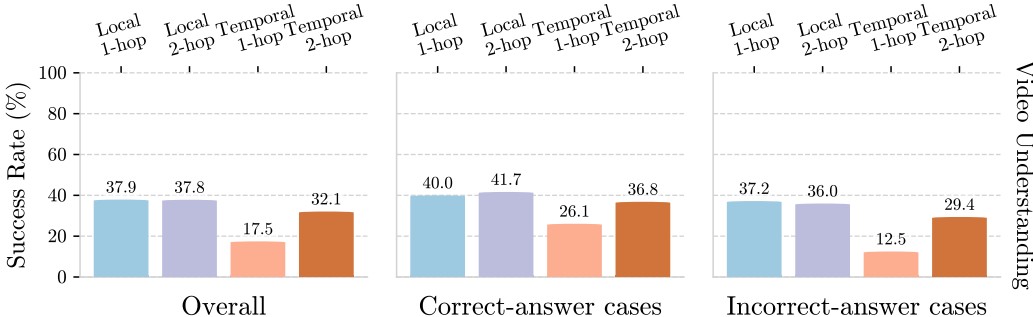

Figure 4: Video understanding performance in successful retrieval cases, with three columns aligned to the setting in fig. 3. Each bar indicates the proportion of queries for which an LLM (GPT-4o-mini in this case), given the query and the ground-truth video description, can correctly identify the key information required to answer. Overall, the proportions remain low (generally below 50%), with performance especially poor in the *Incorrect-answer cases*, where retrieved videos rarely provide sufficient information. This highlights video understanding as a critical area in need of improvement.

*Incorrect-answer cases*. This indicates that the passage retriever is generally effective, and retrieval errors might not be major contributor to possible failures for *Incorrect-answer cases*.

**Video retrieval is challenging but not the main bottleneck.** Video retrieval success rate generally falls in the 50%–60% range, indicating that identifying the correct video clip remains somewhat challenging. However, even in the *Incorrect-answer cases*, video retrieval still achieves around 50% accuracy, suggesting that at least half of the failed queries still retrieve reasonably relevant video clips. These results indicate that retrieval alone does not fully explain the failures observed.

### 3.3 Analysis of Evidence Understanding

We next analyze how well the model understands retrieved evidence, both video and text. The goal of this evaluation is to probe not only whether retrieval succeeds, but also whether the model can effectively interpret the retrieved material to answer the queries. Here we only consider queries where retrieval is successful. Since no ground-truth metric exists, we rely on proxies. For video understanding, we use an LLM (*GPT-4o-mini* here) as a judge: given the query and the query-conditioned caption of the ground-truth video clip, the judge is prompted to determine whether the key information needed to answer is present. For passage understanding, we directly examine the model's answer distribution across options *A–F* (Section 2), where *A*, *C*, and *E* denote counterfactuals and *B*, *D*, and *F* denote factual statements. Therefore, a model that understands the passage should consistently favor *B*, *D*, and *F*. The results are summarized in fig. 4 and fig. 5 and we present the detailed prompts in appendix D.1. We can summarize some key observations as follows:

**Passage understanding is proficient.** When the correct passage is retrieved, the model consistently favors *B*, *D*, and *F* over counterfactuals in about 80% of the cases. This suggests that passage understanding is reliable, though still leaving room for improvement.

**Video understanding is the main bottleneck.** Overall, video understanding performance remains low (generally below 50%), with especially poor results in the *Incorrect-answer cases*, where retrieved videos rarely provide sufficient information. This highlights video understanding as the critical limitation in current systems.

In short, our analysis reveals that both passage and video retrieval are reasonably effective, and that passage understanding is proficient. *The main bottleneck lies in video understanding, where models struggle to extract the key information needed to answer queries, even when the correct video is retrieved.* This points to the need for improved visual grounding and reasoning ability in *EgoFact*.

## 4 Towards Better Video Understanding

*How can we improve video grounding and understanding for RAG?*

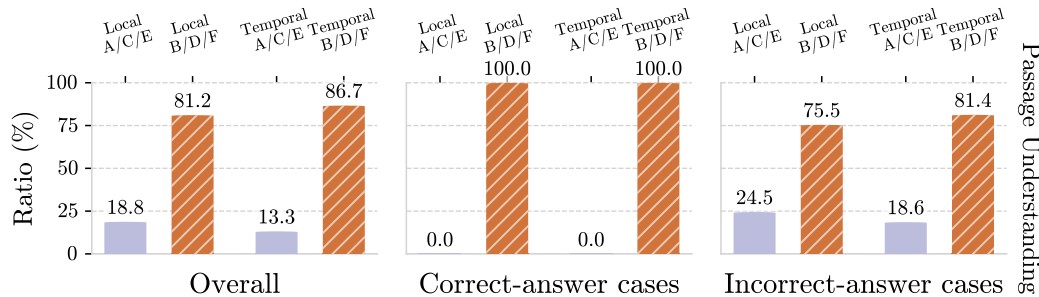

Figure 5: Textual understanding performance under successful retrieval. The three columns follow the same setting as in fig. 3. Each bar shows the distribution of answers across options *A–F*, where *A*, *C*, and *E* denote counterfactual statements and *B*, *D*, and *F* denote factual ones. When the correct passage is retrieved, the model consistently favors *B*, *D*, and *F*, suggesting that passage understanding is reliable. Note that in *Correct-answer cases*, the answer is always *F* by design.

This question may seem daunting at first glance, as video understanding remains a longstanding open challenge in AI. Yet our analysis offers a different perspective: by examining where existing RAG systems succeed, we identify a promising path forward.

Specifically, effective video understanding requires a model to *grasp the query*, *ground it in video content*, and *generate an textual response (e.g., , answer or caption)*; failure in any component leads to errors. Our retrieval analysis reveals a crucial insight: current models can retrieve relevant video clips reliably, even when the query depends on short, localized evidence within the clip. This indicates that existing retrieval mechanisms already handle the first two components—*grasping the query* and *grounding it in video content*—reasonably well.

These findings suggest a fresh perspective: rather than requiring models to ground and generate jointly, we can pre-localize the evidence in video and allow the model to focus exclusively on generation.

Table 2: Video understanding performance with and without localization under the same setting as fig. 4.

| Method | All | Correct-answer cases | Incorrect-answer cases |
|---|---|---|---|
| VideoRAG (Ren) | 30.33 | 41.67 | 28.17 |
| + Localization | **40.95** | **44.05** | **38.89** |

**Our proposal.** Based on this observation, we advocate for a lightweight yet effective solution: *video localization*. Before feeding content into the generator, we pre-localize a short segment of the video that is most relevant to the query. By narrowing the temporal scope, we reduce noise and sharpen cross-modal alignment, enabling the LLM to focus entirely on reasoning and generation. Importantly, this design does not require retraining the backbone model; rather, it integrates seamlessly as a plug-and-play module in the RAG pipeline. As we demonstrate in the following sections, this simple adjustment consistently improves video understanding and leads to more reliable multimodal reasoning.

## 4.1 EXPERIMENT

**Baselines.** We compare our approach against several representative RAG frameworks for multimodal reasoning. (1) *GraphRAG* (Edge et al., 2025): a traditional graph-based RAG model that constructs local–global graphs over the corpus. For adaptation to our setting, we follow prior practice and use *GPT-4o-mini* to generate a textual description for each video, which is then treated as a node in the graph. (2) *VideoRAG (Jeong)* (Jeong et al., 2025): a video-centric RAG model that retrieves relevant video clips given a query. To align with our benchmark, we also extend it with an additional passage-retrieval branch so that both modalities are considered. (3) *VideoRAG (Ren)* (Ren et al., 2025): the state-of-the-art video-based RAG framework that serves as the main baseline in our analysis. (4) *VideoRAG (Ren) + Localization*: our variant that integrates the proposed localization step into the *VideoRAG (Ren)* model, enabling the system to focus on the most query-relevant video segments before answer generation. We detail implementation and hyperparameters in appendix D.2.

**Metrics.** We evaluate all methods on *EgoFact* using accuracy as the main metric, defined as the proportion of queries for which the model selects the correct answer option. We report results separately for each of the four query families (Local 1-hop, Local 2-hop, Temporal 1-hop, Temporal 2-hop) as well as overall accuracy across all queries.

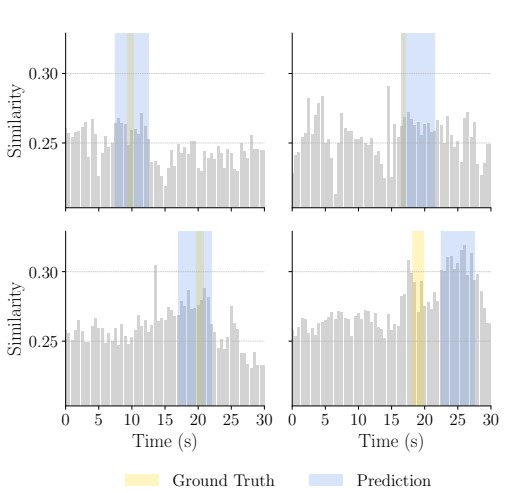

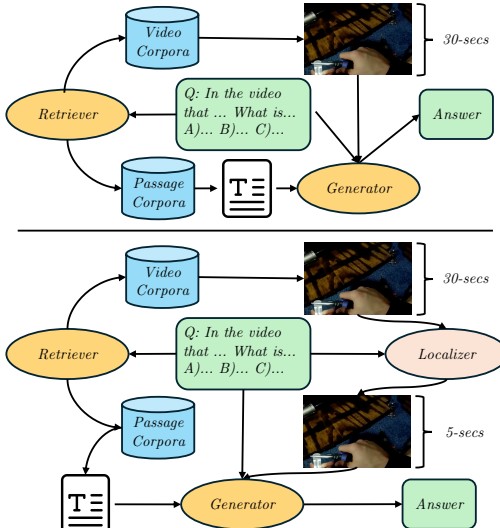

Figure 6: Demonstration of the localization process. Given a query and a retrieved video, we sample frames at regular intervals, compute query–video relevance, and select the most relevant contiguous segment as the localized snippet for reasoning. Four examples from *EgoFact* are shown: the first three are successes, while the last is a failure.

Figure 7: Comparison of workflows. *Top:* baseline pipeline of (Ren et al., 2025), where candidate video clips and passages are retrieved and fed directly into reasoning without localization. *Bottom:* our approach adds a localization module to identify the most relevant segment within each retrieved video before answer generation.

**Methodology.** For localization, we first sample each video at 2 frames per second. Given a target segment length of 5 seconds (about 10 frames), we slide a window of this length across the entire sequence of sampled frames. For every possible window, we compute the average query–video similarity using *MetaCLIP* (Xu et al., 2024), and select the window with the highest mean similarity as the localized segment. We then fix the downstream frame budget at 15 frames for all video-based methods, including both baselines and our approach. All the other setting remain the same with section 3. We illustrate this process in fig. 6, and provide the detailed prompts, workflow, and hyperparameters in appendix D.2.

Table 3: Comparison of RAG variants on *EgoFact*. Columns correspond to query families (1-hop vs. 2-hop, Local vs. Temporal), with the last column showing the overall average. VideoRAG (Ren) with localization consistently outperforms its counterpart, especially on 1-hop queries. Best results are in bold.

| Method | Local 1-hop | Local 2-hop | Temporal 1-hop | Temporal 2-hop | Overall |
|---|---|---|---|---|---|
| GraphRAG | 15.04 | 8.22 | 21.01 | 4.40 | 13.13 |
| VideoRAG (Jeong) | 10.62 | 12.33 | 29.41 | 9.89 | 16.41 |
| VideoRAG (Ren) | 25.66 | 24.66 | 30.25 | **27.47** | 27.27 |
| + Localization | **31.86** | **27.40** | **43.70** | **27.47** | **33.59** |

**Results.** The results are summarized in table 3. Incorporating localization into *VideoRAG (Ren)* consistently improves performance across three query families—*Local 1-hop*, *Local 2-hop*, and *Temporal 1-hop*—as well as the overall average. The gains are most pronounced in 1-hop settings, where queries rely more directly on video evidence. This aligns with our earlier finding that textual passages are generally easier to interpret, while localization sharpens the temporal scope for video reasoning. Although 2-hop queries see smaller improvements, the consistent gains confirm localization as an effective strategy for enhancing multimodal reasoning under *EgoFact*. Consistent with this, the analysis in table 2 shows that localization makes the LLM more likely to identify the key information needed to answer.

## 4.2 ANALYSIS

We analyze our proposed localization method by asking two natural questions. (1) *how does localization perform under different temporal ranges?*, and (2) *to what extent does localization improve over naive alternatives?* To address these questions, we design two experiments. In the first, we compare localization against a *Video Split* baseline that divides each retrieved video into fixed-length, non-overlapping segments (e.g., 5 seconds) and uses them as retrieval units. In the

second, we vary the window size used in localization to examine how temporal granularity influences performance.

**Findings.** The results are summarized in fig. 8. On the one hand, *Localization* consistently outperforms naive splitting, confirming that scope preservation and relevance scoring are crucial for effective video understanding. On the other hand, localization performance follows an inverted U-shape across window sizes: very short windows risk missing essential context, very long windows dilute relevance with noise, while intermediate ranges strike the best balance. Together, these results highlight both the effectiveness of localization and the importance of selecting appropriate temporal granularity.

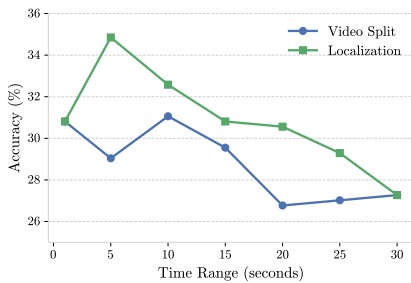

Figure 8: Comparison of *Video Split* and *Localization*. Localization outperforms naive segmentation and shows an inverted U-shaped performance curve, peaking at mid-sized windows.

Although both *Video Split* and *Localization* enable fine-grained retrieval, they differ in how discriminative the search space becomes. When a video is split into many short chunks, similar-looking fragments from different videos may appear nearly indistinguishable, making retrieval more noisy. In contrast, *Localization* first identifies a query-relevant video clip at the global level and then refines the evidence within that scope, which reduces confusion across videos and yields clearer grounding.

## 5 RELATED WORK

**Multimodal RAG Approaches.** Prior work has explored RAG for multimodal settings, especially video, but under settings different from ours. Jeong et al. (2025) introduces video with adaptive frame selection and corpus-level retrieval over *HowTo100M* (Miech et al., 2019), but both retrieval and generation rely on auxiliary text like subtitles or transcripts. Ren et al. (2025) design the *LongerVideos* benchmark for lecture, documentary, and entertainment series, emphasizing cross-video reasoning; their framework integrates graph-based textual knowledge grounding with multimodal embeddings, operating mainly in text-rich contexts. Luo et al. (2024) focus on intra-video only retrieval: from a single target video, they extract OCR, ASR, and object-detection metadata, filtering them for relevance before feeding into an LVLM. *UniversalRAG* (Yeo et al., 2025) extends RAG to text, image, and video corpora with a modality-and granularity-aware router; however, its testbed merely concatenates existing datasets and can be addressed using simple 1-hop reasoning.

**Challenges for Prior Approaches on *EgoFact*.** *EgoFact* removes these scaffolds: our videos have no subtitles, ASR, or OCR, and complementary passages do not mirror video content but provide external knowledge for the second reasoning hop. This design enforces a stricter setting, where models must effectively retrieve from both the video corpus and the complementary text corpus without relying on subtitles or transcriptions. Methods optimized for video–text co-availability, or for cross-video aggregation, underperform here because (i) their retrieval modules depend on auxiliary text to bridge video and query, (ii) their generation modules assume the availability of text-aligned frames, and (iii) they are not designed for fine-grained intra-video grounding in the absence of textual aids. *EgoFact* thus complements prior benchmarks by explicitly highlighting visual grounding as the critical challenge and testing whether systems can succeed in multi-hop multimodal reasoning without relying on text that merely shadows video content.

## 6 CONCLUSION

As multimodal AI continues to advance, robust benchmarks are crucial for guiding progress. In this work, we introduced *EgoFact*, the first testbed designed to intertwine visual and textual knowledge, and provided a systematic diagnosis of existing RAG systems. We envision *EgoFact* as a foundation for developing and evaluating more capable multimodal RAG approaches. Looking ahead, future directions include incorporating richer modalities, supporting deeper reasoning chains, and enabling more interactive settings—paving the way for AI systems that can better understand and assist in complex real-world scenarios.

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

## A  THE USAGE OF LLMS

Besides the query construction process mentioned in section 2, the LLMs are used solely for rewriting purposes to ensure clarity and coherence of the text. They have not been employed for generating any original content or research ideas. All technical details, methodologies, and findings are the result of our own research and analysis.

## B  COMPUTATIONAL RESOURCES

All experiments were conducted on a single NVIDIA A100 GPU with 80GB memory.

## C  *EgoFact* DETAILS

### C.1  VIDEO CORPORA CONSTRUCTION DETAILS

To constrain the volume of video data while ensuring relevance to everyday activities, we use videos from the *Ego4DSounds* subset (Chen et al., 2024), restricted to the following annotated scenarios:

- *Bike mechanic*
- *Getting car fixed*
- *Car mechanic*
- *Car - commuting, road trip*
- *Fixing something in the home*

Each selected video is further segmented into uniform 30-second windows, which serve as the atomic units of video corpora. This procedure yields an initial collection of $2,124$ video segments. Then, we screenshot every 3-second to get $10$ screenshots for each segment. For each screenshot, we use *GPT-4o-mini* with the prompt below to generate a concise and detailed narration.

```
**ROLE**
You are a scene understanding expert.

**INPUT**
•Scenario: a list of short strings identifying the video context.
•Screenshot: the image itself, provided as an attached image.

**OUTPUT**
Output a single paragraph that describes the scenario in detail.
```

Then we concatenate the narrations of all screenshots within a segment to form a comprehensive video description, then concatenate them with *Ego4DSounds* labels and sort to get a video timeline. The timeline is used for two purposes: (1) to generate keywords for video embedding and clustering, and (2) to generate narrations for query construction. To generate narrations, we use *OpenAI o3* with the following prompt:

```
**TASK**
Generate a narration for a video timeline.

**INPUT**
A video timeline in the format:
Scenarios: <comma-separated list of contexts>
```

```
Timeline a list of entries, each line in the format:
  [<timestamp>] (<type>) <text>

Where:
- `<timestamp>` is the time in seconds, formatted as HH:MM:SS.ss
- `<type>` is either "Sound" or "Scene"
    - Scene - a detailed visual description of screenshot at that second
    - Sound - a label indicating a sound in the video at that second
- `<text>` is the description of the sound or scene

**OUTPUT**
Return two narrations:
1. A concise narration that summarizes the video content in a single
    sentence, starts with "the video that".
2. A detailed narration that describes the video content in a few
    sentences, capturing the main events, actions, and objects, starts
    with "the video that".
Return only the parsable JSON object with two keys "concise" and
    "detailed", each containing the respective narration, without any
    additional text.

**EXAMPLE**
INPUT:
Scenarios: Bike mechanic, Fix something in the home
Timeline:
[00:01:30.00] (Scene) The scene depicts a bike mechanic working in a
    room, where various tools are scattered across a wooden workbench.
[00:01:32.82] (Sound) #C C picks up a kick scooter from the floor.
[00:01:35.33] (Sound) #O A man walks in the door.
...

OUTPUT:
{
    "concise": "the video that a bike mechanic works in a room with
        scattered tools, while a person picks up a kick scooter and
        another person enters.",
    "detailed": "the video that shows a bike mechanic in a room filled
        with various tools on a wooden workbench. At one point, the
        camera ..."
}
```

To generate keywords, we also use *OpenAI o3* with the following prompt:

```
**TASK**
Generate a list of keywords from the provided video timeline.

**INPUT**
A video timeline in the format:
Scenarios: <comma-separated list of contexts>
Timeline a list of entries, each line in the format:
  [<timestamp>] (<type>) <text>

Where:
- `<timestamp>` is the time in seconds, formatted as HH:MM:SS.ss
- `<type>` is either "Sound" or "Scene"
    - Scene - a detailed visual description of screenshot at that second
    - Sound - a label indicating a sound in the video at that second
- `<text>` is the description of the sound or scene

**OUTPUT**
Return a JSON array of keywords that capture the main objects, actions,
    contexts, etc., each paired with a relevance score from 1 to 10. The
    keywords should be as detailed as possible.
Return only the parsable JSON array without any additional text.
```

```
**EXAMPLE**
INPUT:
Scenarios: Bike mechanic, Fix something in the home
Timeline:
[00:01:30.00] (Scene) The scene depicts a bike mechanic working in a
    room, where various tools are scattered across a wooden workbench.
[00:01:32.82] (Sound) #C C picks up a kick scooter from the floor.
[00:01:35.33] (Sound) #O A man walks in the door.
...

OUTPUT:
{
    "bike mechanic": 10,
    "kick scooter": 8,
    "tools": 6,
    ...
}
```

After obtaining the narrations and keywords for each segment, we embed and weight the keywords (weight is the relevance score generated by previous step) to get video embeddings using *OpenAI-Text-Embedding-3-small*. Then we apply $k$-nearest neighbor (kNN) clustering in the embedding space to identify representative samples and eliminate highly redundant segments. The hyperparameters for clustering are set as: number of clusters $k = 0.2 * 2124$. After filtering, we obtain a curated set of $424$ representative segments, each 30 seconds in length. Then we filter again to remove segments that have too few events (less than 2) resulting in a final set of $127$ segments. These segments constitute the backbone of the video corpora in *EgoFact*, providing a controlled yet realistic setting for evaluating cross-modal retrieval and reasoning.

## C.2 QUERY CONSTRUCTION DETAILS

After obtaining the video corpora, we generate queries based on the event annotations provided in *Ego4DSounds*. Our query generation pipeline consists of several stages:

### C.2.1 EVENT STRUCTURE EXTRACTION

We first convert the raw event annotations into a structured format using *OpenAI o3*. Each event is represented as:

- **rewritten**: the human-readable rewritten event text
- **agent**: the person performing the action
- **action**: the verb describing the action
- **patient**: the object involved in the action
- **patient_canonical**: the canonical form of the patient for retrieval
- **patient_location**: the location of the patient
- **patient_destination**: the destination of the patient
- **instrument**: any instrument used in the action
- **event_location**: the location where the event takes place

For example, the event `"#C C wearer picks the cellphone from the table"` is converted to:

- **rewritten**: the camera wearer picks up the cellphone from the table
- **agent**: camera wearer
- **action**: pick
- **patient**: cellphone

- **patient_canonical**: cellphone

- **patient_location**: table

- **patient_destination**: null

- **instrument**: null

- **event_location**: null

### C.2.2 QUERY TEMPLATE CONSTRUCTION

For *local 1-hop queries*, we construct questions asking about the interacting object in a specific video context: `"In [video description], what is the object that [agent] [action] [location context]?"`

For *temporal 1-hop queries*, we identify event pairs within the same video segment occurring within 4 seconds of each other, and construct questions with temporal relationships: `"In [video description]. Before/After [reference event], what is the object that [target agent] [target action] [target location context]?"`

### C.2.3 DEDUPLICATION AND DISAMBIGUATION

We apply deduplication to remove semantically similar queries by checking for clips with similar visual indicators (similarity > 0.7) that contain events with similar action descriptions (similarity > 0.8) but different objects (similarity < 0.9). This ensures each query has a unique, unambiguous answer that cannot be confused with events in visually similar clips.

### C.2.4 DISTRACTOR GENERATION AND QUALITY CONTROL

For each query, we use *OpenAI o3* to generate 5 distractor options plus the correct answer with the following criteria:

```
**TASK**
Given a question and answer pair for video clip understanding, generate
    5 wrong
distractors and score the overall question quality from 0-10 based on:
- Reasonable distractors in context
- Visually distinguishable from correct answer
- Not synonyms or near-equivalents
- Cannot be answered without watching the video

**OUTPUT**
JSON object with "Options" (6 total), "Reasoning", and "Score"
```

We filter out queries with scores below 7 and further remove queries that can be answered by *OpenAI o3* without video context, ensuring all queries require visual understanding.

### C.2.5 MULTI-HOP QUERY CONSTRUCTION

For *2-hop* queries, we extend *1-hop* queries by replacing the answer with factual statements about the answer. The process involves:

**1. Passage Retrieval**: For each answer option, we retrieve semantically related WikiHow passages using *OpenAI-Text-Embedding-3-small* with similarity above 0.5.

**2. Statement Generation**: For each option-passage pair, we generate 4 statements (3 counterfactual, 1 factual) using *OpenAI o3*:

```
**TASK**
Given a term and passage, generate exactly four statements:
1-3: Plausible but incorrect statements
4: True fact from the passage
Requirements:
- Not commonsense knowledge
```

– Use "it"/"this" to refer to term
– Self-contained sentences
– No direct passage references

```
**OUTPUT**
JSON array with exactly four strings
```

**3. Validity Verification**: We test each statement set with *OpenAI o3* both with and without the reference passage:

```
**TASK**
Given a multiple-choice question with options A-E, determine the correct
    answer.

**INPUT**
Passage: [reference passage or empty]
Question: [multiple-choice question]

**OUTPUT**
Single letter A-E indicating correct answer
```

We ensure *validity* (correct answer identifiable with passage) and *non-triviality* (correct answer not identifiable without passage).

**4. Final Assembly**: We select 2 factual statements and 4 counterfactual statements from verified option-passage pairs, constructing the final 6-option multiple choice question with exactly one correct factual statement, prefixed with contextual information: `"When [doing activity]. [Statement about object]"`.

## C.3   EXAMPLE QUERIES

We provide examples of the four query types in *EgoFact* in table 4.

## C.4   FULL QUERY LIST

The complete list of queries in *EgoFact* is included here for later reference.

Query ID: query_00000; Query Text: In a video that cleaning and repairing equipment on a cluttered workbench, what is the object that the camera wearer plugs into the socket? A) power drill B) soldering iron C) desk lamp D) hot glue gun E) vacuum cleaner F) cable; Passage UIDs: None; Passage Titles: None

Query ID: query_00000_2hop; Query Text: In a video that cleaning and repairing equipment on a cluttered workbench, which statement is true about the object that the camera wearer plugs into the socket? A) When cleaning your room as teens. Professional cleaning routines advise operating it before picking up visible rubbish so that loose trash is drawn together for easier disposal. B) When cleaning your room as teens. When refreshing a bedroom, it should first be run over the floor and then moved up to skirting boards, cabinet tops, desk surfaces, and even the inside of wardrobes to remove lingering dust. C) When making a paper christmas tree. It is generally avoided for embellishing the pointed tip of a paper cone because the thick molten glue pools and distorts the cone's shape. D) When making a paper christmas tree. It is considered the most effective adhesive for securing decorations directly to the pointed top of a paper-cone Christmas tree. E) When fixing a broken ipod. Because its connector is permanently fused to the logic board, standard repair guides recommend cutting it and soldering in a replacement. F) When fixing a broken ipod. It lies hidden under a metal tray near the bottom of the device, and after peeling away a layer of tape you have to pry it out of its socket to release the display.; Passage UIDs: ['6328e5ea5c164bdcb6ca48bd52dc5e98', '6328e5ea5c164bdcb6ca48bd52dc5e98', '7e358fd8a5da4c3399e10e15a3d47d7b', '7e358fd8a5da4c3399e10e15a3d47d7b', '308ebf3186434f91af8b15eefbdfa5', '308ebf3186434f91af8b15eefbdfa5']; Passage Titles: ['how to clean your room teens', 'how to clean your room teens', 'how to make a paper christmas tree 2', 'how to make a paper christmas tree 2', 'how to fix a broken ipod 7', 'how to fix a broken ipod 7']

Query ID: query_00001; Query Text: In a video that attaching safety straps to a child's high chair, what is the object that the camera wearer put into the kids seat? A) toy car B) bib C) cushion D) bottle E) spoon F) belt; Passage UIDs: None; Passage Titles: None

Query ID: query_00001_2hop; Query Text: In a video that attaching safety straps to a child's high chair, which statement is true about the object that the camera wearer put into the kids seat? A) When making homemade chalk. It serves as the recommended stirring implement for combining tempera paint and plaster of Paris until no lumps remain. B) When making homemade chalk. It is employed to help pour each colored chalk mixture neatly into its individual mold. C) When cleaning your room as teens. The instructions suggest refilling this with a water-cleanser mix to spray and wipe down mirrors and windows. D) When cleaning your room as teens. The guide explicitly tells you to put it in the trash bin whenever you come across it while tidying up. E) When making cat toys out of common household items. This is recommended as the ideal material to thread through a hole in a stuffed animal when crafting a dangling cat toy. F) When making cat toys out of common household items. It is never actually mentioned among the suggested materials—like string or ribbon—when describing how to assemble the homemade cat toy.; Passage UIDs: ['3b4f9768b45744afb3420786af56482d', '3b4f9768b45744afb3420786af56482d', '6328e5ea5c164bdcb6ca48bd52dc5e98', '6328e5ea5c164bdcb6ca48bd52dc5e98', '53603216dabf47a5b3660c9c4e7ac362', '53603216dabf47a5b3660c9c4e7ac362']; Passage Titles: ['how to make homemade chalk 1', 'how to make homemade chalk 1', 'how to clean your room teens', 'how to clean your room teens', 'how to make cat toys out of common household items 8', 'how to make cat toys out of common household items 8']

Query ID: query_00002; Query Text: In a video that repairing a car wheel and brakes using tools, what is the object that the camera wearer picks from the bonnet? A) screwdriver B) car jack C) brake pad D) oil funnel E) flashlight F) wheel spanner; Passage UIDs: None; Passage Titles: None

Query ID: query_00002_2hop; Query Text: In a video that repairing a car wheel and brakes using tools, which statement is true about the object that the camera wearer picks from the bonnet? A) When preparing a car for winter driving. The article claims that pointing it at an ice-coated windshield will generate enough heat to melt the frost before driving. B) When preparing a car for winter car preparedness conspicuously omits it, never suggesting that drivers carry one at all. C) When assembling a bmx bike. Slightly toeing it in so that only the front edge contacts the rim first is the standard way to position it during setup. D) When assembling a bmx bike. With properly installed on a BMX bike, it sits parallel to the wheel rim and stays roughly one millimetre (0.04 inch) away from the rim surface. E) When checking your car before a road trip. It is specifically designed to loosen a car's oil filter housing during routine oil changes. F) When checking your car before a road trip. It is conspicuously absent from the recommended emergency-tool list that includes a flashlight, screwdrivers, pliers and an adjustable spanner for road trips.; Passage UIDs: ['a14401b6093b40c588cdfb5c1fc60180', 'a14401b6093b40c588cdfb5c1fc60180', 'b95d78cd77040c70bb681db201039f7', 'b95d78cd77040c70bb681db201039f7', '640c06aa084849b887585a92534a399', '640c06aa084849b887585a92534a399']; Passage Titles: ['how to prepare a car for winter driving', 'how to prepare a car for winter driving', 'how to assemble a bmx bike', 'how to assemble a bmx bike', 'how to check your car before a road trip', 'how to check your car before a road trip']

Query ID: query_00003; Query Text: In a video that loading a box into a car's trunk, what is the object that the man puts into between the chairs? A) suitcase B) cooler C) backpack D) folded blanket E) grocery bag F) box; Passage UIDs: None; Passage Titles: None

Query ID: query_00003_2hop; Query Text: In a video that loading a box into a car's trunk, which statement is true about the object that the man puts into between the chairs? A) When helping senior citizens. It is advised that large-font labels be attached to it so that seniors can easily distinguish its contents at a glance. B) When helping senior citizens. It is mentioned that helpers should carry it into the senior citizen's home and unpack the groceries for them. C) When redoing your room for a hundred or less. It is recommended to be spread over stuffed animals on the bed so the toys appear tidier and more mature. D) When redoing your room for a hundred or less. It should match the comforter, sheets, bed skirt and throw pillows so the bed becomes an immediately striking focal point when anyone walks into the room. E) When doing quick chores your parents will appreciate. It stores the custom CDs and playlists of the songs your parents enjoyed as teenagers, ready for the next long road trip. F) When doing quick chores your parents will appreciate. It is crammed with old family photographs that your parents have been postponing sorting into a proper album.; Passage UIDs: ['7e31983d436774de2a9e3e6ef6ed8fd8b', '7e31983d436774de2a9e3e6ef6ed8fd8b', '5e1acbb3948e4bb9879fcf522be802bf', '5e1acbb3948e4bb9879fcf522be802bf', '196676f31b344429366030ac2c45f3', '196676f31b344429366030ac2c45f3']; Passage Titles: ['how to help senior citizens 3', 'how to help senior citizens 3', 'how to redo your room for a hundred or less', 'how to redo your room for a hundred or less', 'how to do quick chores your parents will appreciate 3', 'how to do quick chores your parents will appreciate 3']

Query ID: query_00004; Query Text: In a video that repairing household items on a cluttered workbench, what is the object that the camera wearer attaches to the white mask? A) blue plastic visor B) red rubber band C) silver metal clip D) green fabric patch E) transparent acetate sheet F) black cardboard cut-out; Passage UIDs: None; Passage Titles: None

Query ID: query_00004_2hop; Query Text: In a video that repairing household items on a cluttered workbench, which statement is true about the object that the camera wearer attaches to the white mask? A) When creating a secret box. It is stretched diagonally across the thicker wooden square to keep the epoxy-held magnets firmly seated in their corners. B) When creating a secret box. It is wrapped around the four wooden strips after gluing so the box walls stay tightly aligned until the adhesive cures. C) When making stim toys. Its back can be glued with rubbing alcohol before gluing when using it as a tactile decoration on a hard plastic phone case. D) When making stim toys. It is generally too large to be used as a decorative element on the back of a standard smartphone case. E) When preparation, it is laid across the phone's display as a temporary shield to prevent silicone from touching the screen. F) When making a cell phone case. Before the silicone fully sets, this can be pressed into the surface to leave personalized imprints that decorate the finished case.; Passage UIDs: ['7dcfdc36339f4ee8ad41cf5706b6116a', '7dcfdc36339f4ee8ad41cf5706b6116a', '7dcfdc36339f4ee8ad41cf5706b6116a', '90b429d87cb545d6ae0fa29e32745d41', '90b429d87cb545d6ae0fa29e32745d41', 'ba71c7b20ed14897b2ce5f4ca184aff7']; Passage Titles: ['how to create a secret box 3', 'how to create a secret box 3', 'how to make stim toys 2', 'how to make stim toys 2', 'how to make a cell phone case 2', 'how to make a cell phone case 2']

Query ID: query_00005; Query Text: In a video that cutting and gluing a cardboard piece onto a face mask, what is the object that the camera wearer adjusts from the white mask? A) white paper strip B) foam nose bridge C) elastic ear loop D) transparent plastic visor E) colored sticker sheet F) black cardboard cut-out; Passage UIDs: None; Passage Titles: None

Query ID: query_00005_2hop; Query Text: In a video that cutting and gluing a cardboard piece onto a face mask, which statement is true about the object that the camera wearer adjusts from the white mask? A) When making your own iphone case. It generally withstands more wear and tear than glitter, rhinestones, or decoupage because its built-in adhesive resists constant rubbing from hands and fabric. B) When making your own iphone case. You can create it yourself by sandwiching designs between sticker paper or packing tape and a light layer of glue to form a ready-to-apply, self-adhesive sheet. C) When making a paper gun that shoots. When the handle is complete, it is slipped inside the trigger assembly to reinforce the lower frame and remains hidden when the paper gun that shoots. It is first folded lengthwise to create a single 45-degree diagonal crease that later functions as the gun's sight. E) When making a gauntlet from soda bottles. It must be coated with all-purpose acrylic morning silver paint in horizontal strokes to give the soda-bottle gauntlet a weathered metallic finish. F) When making a gauntlet from soda bottles. It is not required at any stage of constructing the soda-bottle gauntlet and is never mentioned in the process, so omitting it has no effect on the final result.; Passage UIDs: ['7bab4f7236554479ab71f9f442daf a90d', '7bab4f7236554479ab71f9f442daf a90d', 'f588ab1039f3444cb22e327305b5ab0', 'f588ab1039f3444cb22e327305b5ab0', 'f51698fa3a3c40b0aebc3e762519ddfc', 'f51698fa3a3c40b0aebc3e762519ddfc']; Passage Titles: ['how to make your own iphone case 3', 'how to make your own iphone case 3', 'how to make a paper gun that shoots 2', 'how to make a paper gun that shoots 2', 'how to make a gauntlet from soda bottles', 'how to make a gauntlet from soda bottles']

Query ID: query_00006; Query Text: In a video that is repairing a table lamp and adjusting its shade, what is the object that camera wearer holds? A) light bulb B) screwdriver C) lamp base D) extension cord E) paint brush F) lamp shade; Passage UIDs: None; Passage Titles: None

Query ID: query_00006_2hop; Query Text: In a video that is repairing a table lamp and adjusting its shade, which statement is true about the object that camera wearer holds? A) When building a cat condo. In the instructions for building the cat condo, it is said to have visible streaks when used with acrylic paint, so its use on the tables is discouraged. B) When building a cat condo. The guide explains that if spray paint is not chosen, acrylic paint can be applied to the tables with it, provided it is a wide brush. C) When making lampshades. It is described as a handy tie used to bring the blanket's ends together after you fold it in the hot-dog or burrito way. D) When making lampshades. It is recommended as the solution for supplying power to a lamp inside a closet that lacks an outlet, with the excess length hidden among the room's piles of stuff. E) When making lampshades. The typical drum design of it employs a single continuous frame rather than the pair of separate wire rings found in other styles. F) When making lampshades. For accurate sizing, the styrene beneath the fabric is cut roughly one inch narrower and half an inch shorter than the fabric piece used to wrap it.; Passage UIDs: ['b9a9da3a93824324f fda994172ca3a0a', 'b9a9da3a93824324f fda994172ca3a0a', 'f8a30d316ae4483aa0af97535e178dbd', 'f8a30d316ae4483aa0af97535e178dbd', '33e0d06d0e2847d98bb023a211127976', '33e0d06d0e2847d98bb023a211127976']; Passage Titles: ['how to build a cat condo 3', 'how to build a cat condo 3', 'how to make a clubhouse in a small closet', 'how to make a clubhouse in a small closet', 'how to make lampshades 1', 'how to make lampshades 1']

Query ID: query_00007; Query Text: In a video that arranging containers during dimly lit home repair, which statement is true about the object that the camera wearer drops into the basket? A) screwdriver D) tape measure E) glove F) bottle; Passage UIDs: None; Passage Titles: None

Query ID: query_00007_2hop; Query Text: In a video that arranging containers during dimly lit home repair, what is the object that the camera wearer drops into the basket? A) hammer B) flashlight C) screwdriver D) ... outside enclosure for cats. The instructions explain that using it becomes unnecessary if a staple gun rather than a nail gun is chosen for attaching the mesh panels. B) When building an outside enclosure for cats. While fastening overlapping sheets of chicken wire to the frame, it is recommended to wear it to avoid being cut by the sharp mesh edges. C) When untangling a newtons cradle. It is explained that it can double as a cutting edge for trimming excess string once the knots are loosened. D) When untangling a newtons cradle. It is specifically recommended for taking precise measurements of replacement string or wire lengths before they...

| Query Type | Query (Example) | Options |
|---|---|---|
| Local 1-hop | In a video that shows cleaning and repairing equipment on a cluttered workbench, what is the object that the camera wearer plugs into the socket? | A) power drill
B) soldering iron
C) desk lamp
D) hot glue gun
E) vacuum cleaner
F) cable |
| Local 2-hop | In a video that shows cleaning and repairing equipment on a cluttered workbench, which statement is true about the object that the camera wearer plugs into the socket? | A) When cleaning your room as teens. Professional cleaning routines advise operating it before picking up visible rubbish... (false statement about vacuum cleaner)
B) When cleaning your room as teens. When refreshing a bedroom, it should first be run over the floor and then moved up... (true statement about vacuum cleaner)
C) When making a paper christmas tree. It is generally avoided for embellishing the pointed tip... (false statement about hot glue gun)
D) When making a paper christmas tree. It is considered the most effective adhesive for securing decorations... (true statement about hot glue gun)
E) When fixing a broken iPod. Because its connector is permanently fused to the logic board... (false statement about cable)
F) When fixing a broken iPod. It lies hidden under a metal tray near the bottom of the device... (true statement about cable) |
| Temporal 1-hop | In a video that shows fixing a device on a living room floor. Before a woman plays with a baby, what is the object that the woman places onto the neck? | A) scarf
B) headphones
C) necklace
D) baby bottle
E) remote control
F) hand |
| Temporal 2-hop | In a video that shows fixing a device on a living room floor. Before a woman plays with a baby, which statement is true about the object that the woman places onto the neck? | A) When childproofing a living room. Because its components are considered too large to swallow... (false statement about remote control)
B) When childproofing a living room. Its small buttons can detach and end up under furniture... (true statement about remote control)
C) When reusing household objects for cleaning. Spritzing the interior of it with cooking spray... (false statement about baby bottle)
D) When reusing household objects for cleaning. Putting it into a mesh laundry bag prevents it... (true statement about baby bottle)
E) When making cat toys out of common household items. It should remain outside the sock while you funnel catnip... (false statement about hand)
F) When making cat toys out of common household items. It is placed inside the sock, with the fingers holding the catnip... (true statement about hand) |

Table 4: Examples of the four query types in *EgoFact*. For 2-hop queries, only F corresponds to the correct object and fact, while other options are distractors or counter-facts.

972 are cut and attached to the cradle's bar. E) When cleaning your room as teens. The instructions suggest refilling this with a water-cleanser mix to spray and wipe down mirrors and windows. F) When cleaning your room
973 as teens. The guide explicitly tells you to put it in the trash bin whenever you come across it while tidying up.; Passage UIDs: ['eb406b3b920a48da81c5d478ad9dcdbd', 'eb406b3b920a48da81c5d478ad9dcdbd', 'ac2cdf78802f4264b65300f3519eef61', '6328e5ea5c164bdcb6ca48bd52dc5e98', '6328e5ea5c164bdcb6ca48bd52dc5e98']; Passage Titles: ['how to build an outside enclosure for cats 3', 'how to build an outside enclosure for cats 3', 'how to untangle a newtons cradle', 'how to clean your room teens', 'how to clean your room teens']

...

1026

1027 Query ID: query_00029; Query Text: In a video that measuring a table using a tape measure and level, what is the object that man picks? A) spirit level B) wooden ruler C) hammer D) screwdriver E) pencil F) tape measure;

1028 Query ID: query_00029_2hop; Query Text: In a video that measuring a table using a tape measure and level, which statement is true about the object that man picks? A) When making duct tape flip flops. Crafters tightly wrap the duct-tape straps around it to form a perfectly cylindrical tube before attaching the straps to the sandal. B) When making duct tape flip flops. To round and slightly enlarge the toe-strap hole, it is pushed

1029 through the opening of the flip flop before the strap is threaded. C) When changing a kids room to a tweens room. It is said to be discouraged for tween redecorating projects because its sharp point poses an unacceptable safety risk. D) When changing a kids room to a tweens room. It, paired with a power drill, is mentioned as sufficient for installing most simple additions during the room makeover. E) When untangling a newtons cradle. It is explained that it can double as a cutting edge for trimming excess string once the knots are loosened. F) When untangling a newtons cradle. It is specifically recommended for taking precise measurements of replacement string or wire lengths before they are cut and attached to the cradle's bar.; Passage UIDs: [...]; Passage Titles: ['how to make duct tape flip flops', 'how to make duct tape flip flops', 'how to change a kids room to a tweens room', 'how to change a kids room to a tweens room', 'how to untangle a newtons cradle', 'how to untangle a newtons cradle']

1030 Query ID: query_00030; Query Text: In a video that is repairing a household device using tools and cables, what is the object that the camera wearer wraps onto the light? A) screwdriver B) tape C) rope D) cloth rag E) zip tie F) cable; Passage UIDs: None; Passage Titles: None

1031 Query ID: query_00030_2hop; Query Text: In a video that is repairing a household device using tools and cables, which statement is true about the object that the camera wearer wraps onto the light? ...; Passage UIDs: [...]; Passage Titles: ['how to untangle a newtons cradle', 'how to untangle a newtons cradle', 'how to make a chalkboard', ...]

1032 Query ID: query_00031; Query Text: In a video that tossing and catching a baseball on grassy field, what is the object that the man touches? A) baseball B) baseball glove C) cap D) fence E) water bottle F) ground; Passage UIDs: None; Passage Titles: None

1033 Query ID: query_00031_2hop; Query Text: In a video that tossing and catching a baseball on grassy field, which statement is true about the object that the man touches? A) When being a little league umpire. Base umpires are instructed to move toward it after contact so they can watch runners without obstructing the play. B) When being a little league umpire. This is presented as a scenario in which little league rules differ from the major leagues, specifically addressing what to do when a ball rolls beneath it. C) When getting into baseball career. ...; Passage UIDs: [...]; Passage Titles: ['how to be a little league umpire', 'how to be a little league umpire', 'how to get into baseball career', 'how to get into baseball career', 'how to hit a home run 1', 'how to hit a home run 1']

1034 Query ID: query_00032; Query Text: In a video that shows disassembling a metal frame in a workshop, what is the object that woman holds? A) wooden plank B) plastic hose C) aluminum ladder D) rubber mallet E) electric drill F) iron pipe; Passage UIDs: None; Passage Titles: None

1035 Query ID: query_00032_2hop; Query Text: In a video that shows disassembling a metal frame in a workshop, which statement is true about the object that woman holds? A) When building a simple dog house. It is said to be optional for building the dog house because all connections can be completed with a hammer and nails instead. B) When building a simple dog house. It is specifically used to drive 1.5-inch screws into the wooden roof frame when assembling the dog house. C) When webbing a chair seat. ...; Passage UIDs: [...]; Passage Titles: ['how to build a simple dog house', 'how to build a simple

1036 dog house', 'how to web a chair seat', 'how to web a chair seat', 'how to create open shelving', 'how to create open shelving']

1037 Query ID: query_00033; Query Text: In a video that connecting a wire to earphones indoors, what is the object that the camera wearer picks from the table in the bedroom? A) remote control B) pair of glasses C) wallet D) hairbrush E) smartphone F) ceramic vase; Passage UIDs: None; Passage Titles: None

1038 Query ID: query_00033_2hop; Query Text: In a video that connecting a wire to earphones indoors, which statement is true about the object that the camera wearer picks from the table in the bedroom? A) When managing time as a working student. It is highlighted as a tool students should use during lunch breaks for quick online quizzes to reinforce learning in 15-30-minute intervals. B) When managing time as a working student. It is singled out as a source of distraction that students are urged to put away entirely to maintain focus while studying. C) When organizing your school bag girls. ...; Passage UIDs: [...]; Passage Titles: ['how to manage time as a working student', 'how to manage time as a working student', 'how to organize your school bag girls', 'how to organize your school bag girls', 'how to excel in high school', 'how to excel in high school']

1039 Query ID: query_00034; Query Text: In a video that removing a bicycle's front wheel and adjusting its components, what is the object that the camera wearer opens? A) water bottle B) toolbox lid C) brake lever D) saddle bag E) handlebar grip F) valve cover; Passage UIDs: None; Passage Titles: None

1040 Query ID: query_00034_2hop; Query Text: In a video that removing a bicycle's front wheel and adjusting its components, which statement is true about the object that the camera wearer opens? A) When designing and sew cold weather mitts for drop handlebars. It traditionally has its hand opening finished with overlock stitching instead of bias tape to minimize thickness at the wrist. B) When designing and sew cold weather mitts for drop handlebars. ...; Passage UIDs: [...]; Passage Titles: ['how to design and sew cold weather mitts for drop handlebars', 'how to design and sew cold weather mitts for drop handlebars', 'how to carry cargo on a bike 1', 'how to carry cargo on a bike 1', 'how to rebuild an engine', 'how to rebuild an engine']

1041 Query ID: query_00035; Query Text: In a video that crafting a helmet mask by cutting and gluing cardboard, what is the object that the camera wearer passes from the left hand to the right hand? A) glue stick B) ruler C) paintbrush D) utility knife E) hot glue gun F) scissors; Passage UIDs: None; Passage Titles: None

1042 Query ID: query_00035_2hop; Query Text: In a video that crafting a helmet mask by cutting and gluing cardboard, which statement is true about the object that the camera wearer passes from the left hand to the right hand? A) When making a furry mouse toy for cats. The only time it is used in the project is when it has been stuffed, when the remaining opening is closed with adhesive instead of thread. B) When making a furry mouse toy for cats. ...; Passage UIDs: [...]; Passage Titles: ['how to make a furry mouse toy for cats', 'how to make a furry mouse toy for cats', 'how to camp in your car', 'how to camp in your car', 'how to potty train a guinea pig', 'how to potty train a guinea pig']

1043 Query ID: query_00036; Query Text: In a video that painting an interior wall with a paintbrush, what is the object that the camera wearer put onto the floor? A) paint roller B) paint tray C) drop cloth D) measuring tape E) screwdriver F) lid; Passage UIDs: None; Passage Titles: None

1044 Query ID: query_00036_2hop; Query Text: In a video that painting an interior wall with a paintbrush, which statement is true about the object that the camera wearer put onto the floor? A) When changing a kids room to a tweens room. It is said to be discouraged for tween redecorating projects because its sharp point poses an unacceptable safety risk. B) When changing a kids room to a tweens room. It, paired with a power drill, is mentioned as sufficient for installing most simple additions during the room makeover. C) When making a circle skirt. ...; Passage UIDs: [...]; Passage Titles: ['how to change a kids room to a tweens room', 'how to change a kids room to a tweens room', 'how to make a circle skirt', 'how to make a circle skirt', 'how to create a secret box', 'how to create a secret box']

1045 Query ID: query_00037; Query Text: In a video that working on a cluttered workshop repair task, what is the object that camera wearer zips? A) backpack B) toolbox C) duffel bag D) pencil case E) sleeping bag F) jacket; Passage UIDs: None; Passage Titles: None

1046 Query ID: query_00037_2hop; Query Text: In a video that working on a cluttered workshop repair task, which statement is true about the object that camera wearer zips? A) When having fun at home. It is presented as an inventive planter that can be stuffed with soil to create a portable flower bed. B) When having fun at home. It is explicitly recommended for hiding money in the yard so that one can fall asleep beneath the stars without leaving home. ...; Passage UIDs: [...]; Passage Titles: ['how to have fun at home 4', 'how to have fun at home 4', 'how to make your bedroom an apartment', 'how to make your bedroom an apartment', 'how to live within your means', 'how to live within your means']

1047 Query ID: query_00038; Query Text: In a video that placing napkins and utensils on a dining table, what is the object that the camera wearer operates? A) fork B) wine glass C) napkin D) candle E) camera F) phone; Passage UIDs: None; Passage Titles: None

1048 Query ID: query_00038_2hop; Query Text: In a video that placing napkins and utensils on a dining table, which statement is true about the object that the camera wearer operates? A) When preparing for a winter road trip. It is presented as a convenient way to film your cooking somewhere while you expand your repertoire of regional dishes throughout winter. B) When being productive during winter. ...; Passage UIDs: [...]; Passage Titles: ['how to be productive during winter', 'how to be productive during winter', 'how to prepare for a winter road trip', 'how to prepare for a winter road trip', 'how to keep a parakeet safe out of its cage', 'how to keep a parakeet safe out of its cage']

1049 Query ID: query_00039; Query Text: In a video that placing napkins and utensils on a dining table, what is the object that person moves from the table? A) dinner plate B) flower vase C) salt shaker D) wine glass E) metal spoon holder F) table cloth; Passage UIDs: None; Passage Titles: None

1050 Query ID: query_00039_2hop; Query Text: In a video that placing napkins and utensils on a dining table, which statement is true about the object that person moves from the table? A) When shopping for new houseware. The purchasing guide labels it as a smaller kitchen essential that should be bought alongside plates, mugs, and cutlery. B) When shopping for new houseware. In the recommended inventory, it is placed in the decorative category together with vases rather than among indispensable kitchen utensils. C) When making a wedding money tree. ...; Passage UIDs: [...]; Passage Titles: ['how to shop for new houseware', 'how to shop for new houseware', 'how to make a wedding money tree', 'how to make a wedding money tree', 'how to clean up after a dinner party', 'how to clean up after a dinner party']

1051 Query ID: query_00040; Query Text: In a video that assembling a baby cot with rods on a carpet, what is the object that the camera wearer picks from the middle of the baby cot? A) screwdriver B) cot mattress C) plastic connector cap D) baby blanket E) instruction manual F) rod; Passage UIDs: None; Passage Titles: None

1052 Query ID: query_00040_2hop; Query Text: In a video that assembling a baby cot with rods on a carpet, which statement is true about the object that the camera wearer picks from the middle of the baby cot? A) When setting up a baby crib. It tells the assembler to secure the casters by striking them directly with a hammer until they lock onto their studs. B) When setting up a baby crib. It indicates exactly two points on the crib headboard where the spring frame should be bolted. C) When making cat jungle gyms and playgrounds. ...; Passage UIDs: [...]; Passage Titles: ['how to set up a baby crib', 'how to set up a baby crib', 'how to make cat jungle gyms and playgrounds 1', 'how to make cat jungle gyms and playgrounds 1', 'how to hang a plasma tv over the fireplace', 'how to hang a plasma tv over the fireplace']

1053 Query ID: query_00041; Query Text: In a video that assembling a baby cot with rods on a carpet, what is the object that camera wearer drops from left hand onto baby cot? A) screwdriver B) baby blanket C) stuffed teddy bear D) plastic bottle E) mobile phone F) rod; Passage UIDs: None; Passage Titles: None

1054 Query ID: query_00041_2hop; Query Text: In a video that assembling a baby cot with rods on a carpet, which statement is true about the object that camera wearer drops from left hand onto baby cot? A) When deep cleaning clean and organize your room. When sorting possessions, the text recommends donating any outdated versions of it together with unneeded clothes. B) When deep clean and organize your room. The organizing guide states that you may keep it on the top of the bedside table as an essential item, provided the surface remains uncluttered. C) When making musical instruments with recycled materials. ...; Passage UIDs: [...]; Passage Titles: ['how to deep clean and organize your room', 'how to deep clean and organize your room', 'how to make musical instruments with recycled materials', 'how to make musical instruments with recycled materials', 'how to hang a plasma tv over the fireplace', 'how to hang a plasma tv over the fireplace']

1055 Query ID: query_00042; Query Text: In a video that repairing car interior while marking a wooden plank, what is the object that the camera wearer passes to the left hand? A) screwdriver B) tape measure C) wrench D) utility knife E) smartphone F) pen; Passage UIDs: None; Passage Titles: None

1056 Query ID: query_00042_2hop; Query Text: In a video that repairing car interior while marking a wooden plank, which statement is true about the object that the camera wearer passes to the left hand? A) When beating night watch island on poptropica. It is protected by a numeric seven-digit passcode that the mall manager provides after the first security test. B) When beating night watch island on poptropica. It includes a mini-game called 'Planet Slurp' alongside tools like security cameras, contacts, notes, and a power button. ...; Passage UIDs: [...]; Passage Titles: ['how to beat night watch island on poptropica', 'how to beat night watch island on poptropica', 'how to camp in your car 1', 'how to camp in your car 1', 'how to make a hamster playground 1', 'how to make a hamster playground 1']

1057 Query ID: query_00043; Query Text: In a video that is replacing knife sharpener pads on a cluttered workbench, what is the object that camera wearer blows? A) screwdriver B) knife sharpener pad C) sandpaper sheet D) safety goggles E) plastic bottle F) glove; Passage UIDs: None; Passage Titles: None

1058 Query ID: query_00043_2hop; Query Text: In a video that is replacing knife sharpener pads on a cluttered workbench, which statement is true about the object that camera wearer blows? A) When turning a plastic bottle into a rabbit shaped plant holder. It should be thoroughly sanded with fine-grit paper before any glue or the coating properly adheres. B) When turning a plastic bottle into a rabbit shaped plant holder. ...; Passage UIDs: [...]; Passage Titles: ['how to turn a plastic bottle into a rabbit shaped plant holder', 'how to turn a plastic bottle into a rabbit shaped plant holder', 'how to make a frosted vase with a rubber band', 'how to make a frosted vase with a rubber band', 'how to make a freddy krueger glove 1', 'how to make a freddy krueger glove 1']

1059 Query ID: query_00044; Query Text: In a video that retrieving letters from a building's mailbox and locking it, what is the object that camera wearer removes? A) envelope B) letter C) phone D) coin E) small padlock F) key; Passage UIDs: None; Passage Titles: None

1060 Query ID: query_00044_2hop; Query Text: In a video that retrieving letters from a building's mailbox and locking it, which statement is true about the object that camera wearer removes? A) When writing a letter to your penpal. It is listed as something you should mention but never physically send, because sharing it might reveal your income level. B) When writing a letter to your penpal. The text recommends including it as a small gift for a pen pal who lives in another country. C) When avoiding playing phone tag at work. ...; Passage UIDs: [...]; Passage Titles: ['how to write a letter to your penpal', 'how to write a letter to your penpal', 'how to avoid playing phone tag at work 3', 'how to avoid playing phone tag at work 3', 'how to activate voicemail on a blackberry 1', 'how to activate voicemail on a blackberry 1']

1061 Query ID: query_00045; Query Text: In a video that wiping a knife and closing a plastic container, what is the object that the camera wearer cuts? A) carrot B) bread loaf C) aluminum foil D) kitchen sponge E) plastic wrap F) tissue paper; Passage UIDs: None; Passage Titles: None

1062 Query ID: query_00045_2hop; Query Text: In a video that wiping a knife and closing a plastic container, which statement is true about the object that the camera wearer cuts? A) When making earplugs. It should be trimmed to the exact size of the ear canal so no excess material protrudes once the wrapped cotton ball is inserted. B) When making earplugs. ...; Passage UIDs: [...]; Passage Titles: ['how to make earplugs', 'how to make earplugs', 'how to make a balloon-powered boat 1', 'how to make a balloon-powered boat 1', 'how to make a diwali paper lantern', 'how to make a diwali paper lantern']

1063 Query ID: query_00046; Query Text: In a video that drilling a screw into a wooden table, what is the object that man removes from the wall? A) wall clock B) picture frame C) light switch cover D) thermostat E) coat hook F) drilling machine; Passage UIDs: None; Passage Titles: None

1064 Query ID: query_00046_2hop; Query Text: In a video that drilling a screw into a wooden table, which statement is true about the object that man removes from the wall? A) When fixing a good vacation home renter. Guests are informed that the rental agreement may require them to leave it set at a particular temperature. B) When improving a bathroom without remodeling. Swapping it for a sleek graphite-finished version is said to provide multiple adjustable five settings for the shower. ...; Passage UIDs: [...]; Passage Titles: ['how to be a good vacation home renter', 'how to be a good vacation home renter', 'how to improve a bathroom without remodeling 2', 'how to improve a bathroom without remodeling 2', 'how to build your own computer case', 'how to build your own computer case']

1065 Query ID: query_00047; Query Text: In a video that drilling a screw into a wooden table, what is the object that the camera wearer raises from the floor? A) screwdriver B) metal screw C) drill bit D) plastic bottle E) smartphone F) wood; Passage UIDs: None; Passage Titles: None

1066 Query ID: query_00047_2hop; Query Text: In a video that drilling a screw into a wooden table, which statement is true about the object that the camera wearer raises from the floor? A) When increasing your neat. The text cautions that carrying it in your hand while walking can actually diminish the non-exercise activity thermogenesis you gain from the stroll. B) When increasing your neat. It is recommended as a convenient tool for recording work or study materials so you can listen to them while walking to boost your daily non-exercise activity. ...; Passage UIDs: [...]; Passage Titles: ['how to increase your neat 1', 'how to increase your neat 1', 'how to build a carpenter bee trap', 'how to build a carpenter bee trap', 'how to make a squirrel feeder 3', 'how to make a squirrel feeder 3']

1067 Query ID: query_00048; Query Text: In a video that painting and caulking wall baseboards with various hand tools, what is the object that camera wearer paint? A) door B) floor C) ceiling D) window E) baseboard F) wall; Passage UIDs: None; Passage Titles: None

1068 Query ID: query_00048_2hop; Query Text: In a video that painting and caulking wall baseboards with various hand tools, which statement is true about the object that camera wearer paint? A) When installing pergo flooring. Installers leave it attached until after all laminate planks are locked in, then remove it to fill the expansion gap with flexible caulk. B) When installing pergo flooring. The vapor barrier is intentionally left oversize at the room's edges so any moisture rising from the slab is channeled into the narrow cavity hidden behind it. ...; Passage UIDs: [...]; Passage Titles: ['how to install pergo flooring 2', 'how to install pergo flooring 2', 'how to build a cat condo 2', 'how to build a cat condo 2', 'how to use crutches', 'how to use crutches']

1069 Query ID: query_00049; Query Text: In a video that nailing and drilling timber while repairing furniture indoors, what is the object that the man in the video attach? A) hinge B) drawer slide C) shelf bracket D) door knob E) metal plate F) latch; Passage UIDs: None; Passage Titles: None

1079 Query ID: query_00049_2hop; Query Text: In a video that nailing and drilling timber while repairing furniture indoors, which statement is true about the object that the man in the video attach? A) When making a desk chair from a car seat. Because its edges are often secured by welding, the recommended approach is to cut it completely and replace it with fresh stock steel before assembly. B) When making a desk chair from a car seat. The front-to-back measurement of it is recorded and then reduced by one widest-side dimension of the box section in order to calculate the two lengths. C) When making a cozy cabin facial tissue box cover. According to the crafting steps for the cardboard cabin, it must be painted black alongside the walls and roof before any tape or shingles are attached. D) When making a cozy cabin facial tissue box cover. In the set of directions for the cozy cabin tissue box cover, it is never required at all because the door is fitted with a handle instead. E) When making a rabbit cage. It works by lifting a hinged bar vertically until it drops

into a slot, locking the rabbit cage without any rotation. F) When making a rabbit cage. When you turn it to the right, it locks the rabbit cage shut.; Passage UIDs: ['2f2cd28839d94d82b6c78576c7321987', '2f2cd28839d94d82b6c78576c7321987', '5939ef72a29c440ab6aab2c5fd0bbd39', '5939ef72a29c440ab6aab2c5fd0bbd39', '1b4d4c411d6a488b883b16dd700a c1d5', '1b4d4c411d6a488b883b16dd700ac1d5']; Passage Titles: ['how to make a desk chair from a car seat', 'how to make a desk chair from a car seat', 'how to make a cozy cabin facial tissue box cover', 'how to make a cozy cabin facial tissue box cover', 'how to make a rabbit cage', 'how to make a rabbit cage']

Query ID: query_00050; Query Text: In a video that nailing and drilling timber while repairing furniture indoors, what is the object that the man in the video place? A) screw B) bolt C) wooden dowel D) glue stick E) metal bracket F) nail; Passage UIDs: None; Passage Titles: None

Query ID: query_00050_2hop; Query Text: In a video that nailing and drilling timber while repairing furniture indoors, which statement is true about the object that the man in the video place? A) When making a memory box table. This is first attached to the tops of the table legs and then the combined pieces are pressed into the box base as a single unit. B) When making a memory box table. It rests on each interior bottom corner of the box, after which the leg's threaded stud is twisted into it to lock the leg firmly in place. C) When maintaining your mobile phone. Mobile repair enthusiasts sometimes insert narrow shavings of it between a wobbly battery and the compartment wall to secure the battery and prevent intermittent shutdowns. D) When maintaining your mobile phone. Simply rubbing the oxidized battery contact terminals with it can clear the corrosion and restore reliable charging in a phone experiencing power issues. E) When fixing a dented ping pong ball. Driving it halfway into a dented ping pong ball is recommended because the metal shaft pushes the plastic outward from the inside. F) When fixing a dented ping pong ball. After using a hair dryer to pop a dent, wrapping the ball in tissue and hanging it from it for a few minutes keeps the ball from sagging while it cools.; Passage UIDs: ['e5c7a574d980d45498b9caab693be123', 'e5c7a574d980d45498b9caab693be123', 'd0751115ada0491db9b183eaab6e598', 'd0751115ada0491db9b183eaab6e598', '00985a31d03a4b568c7ee8b487bbc8f0', '00985a31d03a4b568c7ee8b487bbc8f0']; Passage Titles: ['how to make a memory box table', 'how to make a memory box table', 'how to maintain your mobile phone', 'how to maintain your mobile phone', 'how to fix a dented ping pong ball', 'how to fix a dented ping pong ball']

Query ID: query_00051; Query Text: In a video that tending hanging flower baskets outside a brick house, what is the object that the camera wearer touches? A) leaf B) watering can C) brick wall D) metal chain E) birdhouse F) flower; Passage UIDs: None; Passage Titles: None

Query ID: query_00051_2hop; Query Text: In a video that tending hanging flower baskets outside a brick house, which statement is true about the object that the camera wearer touches? A) When building a milk carton birdhouse. To suspend it safely, one should punch holes halfway down the carton's sides and run a single loop of twine through both openings. B) When building a milk carton birdhouse. When fashioned from a milk carton, it can only accommodate birds that nest in size from a sparrow up to a robin. C) When receiving estimates for home improvements. It is customary to offer a routine repair job for which obtaining a single contractor estimate is considered sufficient. D) When receiving estimates for home improvements. The illustrative example offered focuses on a stone wall in the front yard rather than it, meaning this type of wall is not actually mentioned at all. E) When reusing empty aluminum cans. It is generally produced by perforating an unopened aluminum can with a series of evenly spaced holes so that, when a candle is placed inside, light filters through the lacy surface. F) When reusing empty aluminum cans. It can be created by cutting an aluminum can into one large circular piece and several smaller petal shapes, gluing them together so that the finished decoration serves as a base for a candle.; Passage UIDs: ['372e476728ac42719bd4cecc9b75c133', '372e476728ac42719bd4cecc9b75c133', '541131b1df5a4ab6a0fdd351ae1ae2f7', '541131b1df5a4ab6a0fdd351ae1ae2f7', 'cf64f682e9be4cfab5907ed6ee2dacac2', 'cf64f682e9be4cfab5907ed6ee2dacac2']; Passage Titles: ['how to build a milk carton birdhouse 1', 'how to build a milk carton birdhouse 1', 'how to receive estimates for home improvements', 'how to receive estimates for home improvements', 'how to reuse empty aluminum cans 1', 'how to reuse empty aluminum cans 1']

Query ID: query_00052; Query Text: In a video that cutting a metal bar clamped in a bench vise, what is the object that the camera wearer turns? A) vise handle B) hacksaw blade C) adjustable wrench D) wooden dowel E) copper pipe F) metal bar; Passage UIDs: None; Passage Titles: None

Query ID: query_00052_2hop; Query Text: In a video that cutting a metal bar clamped in a bench vise, which statement is true about the object that the camera wearer turns? A) When changing the water pump on a 2 . 01 , 4 cylinder ford probe 1993 to 1997. The procedure cautions that applying it to the crankshaft harmonic-pulley bolt is dangerous because its jaws may slip under the high torque required. B) When changing the water pump on a 2 . 01 , 4 cylinder ford probe 1993 to 1997. The instructions recommend using it, adjusted to fit roughly a one-inch (25.5-26 mm) hex, to nudge the camshaft back and forth while aligning the cam sprockets. C) When easilying remove a cylinder head on an auto with frozen bolts. During exhaust repairs, mechanics sometimes wedge it between the manifold and pipe to serve as a temporary spacer before welding stainless-steel sections together. D) When easilying remove a cylinder head on an auto with frozen bolts. When removing only one cylinder head while leaving the exhaust manifold attached, it can be used to cut the specific arm of the cross-over pipe that connects to that head. E) When fixing a door. It is identified as the strike plate on the doorjamb that latch engages with and may need filing to adjust alignment. F) When fixing a door. It is never referenced as either a part of the door assembly or a tool in the step-by-step procedure for fixing the misaligned door.; Passage UIDs: ['5a8d0b6b61e40889964b251c2afc831', '5a8d0b6b61e40889964b251c2afc831', '965c5c1607 4b49bc925d4fdc77eb356f', '965c5c16074b49bc925d4fdc77eb356f', '056d5df4e09e46fba24e9b1adcb24a8', '056d5df4e09e46fba24e9b1adcb24a8']; Passage Titles: ['how to change the water pump on a 2 . 01 , 4 cylinder ford probe 1993 to 1997', 'how to change the water pump on a 2 . 01 , 4 cylinder ford probe 1993 to 1997', 'how to easily remove a cylinder head on an auto with frozen bolts', 'how to easily remove a cylinder head on an auto with frozen bolts', 'how to fix a door 1', 'how to fix a door 1']

Query ID: query_00053; Query Text: In a video that preparing a parked motorcycle in a dimly lit garage, what is the object that camera wearer put onto the motorbike? A) motorcycle helmet B) metal toolbox C) backpack D) leather jacket E) fuel canister F) shopping bag; Passage UIDs: None; Passage Titles: None

Query ID: query_00053_2hop; Query Text: In a video that preparing a parked motorcycle in a dimly lit garage, which statement is true about the object that camera wearer put onto the motorbike? A) When dressing like kristen stewart. This is portrayed as too formal for relaxed evenings and is therefore discouraged for a casual night out in the suggested looks. B) When dressing like kristen stewart. For a professional ensemble inspired by Stewart, it is supposed to complete the look together with metal accessories such as gold or silver earrings and a long necklace. C) When sleeping comfortably in a car. It is recommended that you attach it to the steering wheel with a short length of cord to deter opportunistic theft through a cracked window. D) When sleeping comfortably in a car. This should contain no more than a single change of clothes if you are not on an extended trip. E) When avoiding being carjacked. Certain tips claim that filling it with sand and wedging it against a garage-door track can block burglars from tripping the release mechanism. F) When avoiding being carjacked. One safety measure recommends placing it inside the car's trunk so that it stays out of sight and does not attract carjackers.; Passage UIDs: ['1f35a40d731446b9b3ce1df843a35441', '1f35a40d731446b9b3ce1df843a35441', 'bc9d30a3e484138a550caf1ce79e5bc', 'bc9d30a3e484138a550caf1ce79e5bc', '9699665322dc452 9a65a15544593dd778', '9699665322dc4529a65a15544593dd778']; Passage Titles: ['how to dress like kristen stewart', 'how to dress like kristen stewart', 'how to sleep comfortably in a car', 'how to sleep comfortably in a car', 'how to avoid being carjacked 1', 'how to avoid being carjacked 1']

Query ID: query_00054; Query Text: In a video that preparing a parked motorcycle in a dimly lit garage, what is the object that the camera wearer adjusts? A) motorcycle mirror B) helmet strap C) handlebar D) kickstand E) riding glove F) camera; Passage UIDs: None; Passage Titles: None

Query ID: query_00054_2hop; Query Text: In a video that preparing a parked motorcycle in a dimly lit garage, which statement is true about the object that the camera wearer adjusts? A) When storing your classic car. It needs to be disconnected from the heater hoses and flushed with rust-inhibiting antifreeze before winter. B) When storing your classic car. It is worn while handling antifreeze and should be used to shield hands from potential burns. C) When riding a manual , 6 speed dirt bike. It locks into place against the rear wheel, acting as a mechanical brake to keep the machine from rolling on steep inclines. D) When riding a manual , 6 speed dirt bike. It is normally dropped only after the rider has killed the engine and shut off the fuel tap, marking the final step in parking the bike. E) When surviving a long road trip teenage girls. Because of its high power consumption, the text advises attaching a dedicated car charger for it at all times rather than packing it in the personal bag. F) When surviving a long road trip teenage girls. It appears on the list of entertainment items—alongside books, an iPod, a tablet, and a phone—that teenage girls are advised to pack in their personal bag for a long road trip.; Passage UIDs: ['3b150b000a1045a49b832d84 0b4d7e', '81290bdacc8c454a9c4852cb69cc0405', '81290bdacc8c454a9c4852cb69cc0405', '23860401dd58464 9a3fe7370c0d7b091', '23860401dd584649a3fe7370c0d7b091']; Passage Titles: ['how to store your classic car', 'how to store your classic car', 'how to ride a manual , 6 speed dirt bike', 'how to ride a manual , 6 speed dirt bike', 'how to survive a long road trip teenage girls', 'how to survive a long road trip teenage girls']

Query ID: query_00055; Query Text: In a video that assembling and repairing wooden furniture in a cozy kitchen, what is the object that the camera wearer moves from hand to kitchen? A) chair B) stool C) cutting board D) cabinet door E) wooden shelf F) table; Passage UIDs: None; Passage Titles: None

Query ID: query_00055_2hop; Query Text: In a video that assembling and repairing wooden furniture in a cozy kitchen, which statement is true about the object that the camera wearer moves from hand to kitchen? A) When making a soda bottle volcano. It is ruled out for the project if it is made of plastic, since the acidic eruption could damage the surface. B) When making a soda bottle volcano. The text advises coating this in vivid accent paint to make window sills stand out against neutral walls. D) When giving your room a makeover. The makeover advice includes adding it, along with beanbags and couches, to give the bedroom an extra spot to sit and relax. E) When making an advent calendar. It is singled out as the best place to let the glued matchboxes dry for several hours before final assembly. F) When making an advent calendar. It is mentioned, along with a countertop, as an optional surface on which the finished matchbox advent calendar can be displayed.; Passage UIDs: ['159ebaa18a7e44498c3eae6cf712e4582', '159ebaa18a7e44498c3eae6cf712e4582', '21703962931f4005a1b544b896965913', '099a641 29ef241868deaeba55615eac7', '099a64129ef241868deaeba55615eac7']; Passage Titles: ['how to make a soda bottle volcano 2', 'how to make a soda bottle volcano 2', 'how to give your room a makeover', 'how to give your room a makeover', 'how to make an advent calendar 2', 'how to make an advent calendar 2']

Query ID: query_00056; Query Text: In a video that shows entering a parked truck and sanitizing hands, what is the object that the camera wearer drops onto co drivers sit? A) backpack B) coffee cup C) clipboard D) phone E) hat F) jacket; Passage UIDs: None; Passage Titles: None

Query ID: query_00056_2hop; Query Text: In a video that shows entering a parked truck and sanitizing hands, which statement is true about the object that the camera wearer drops onto co drivers sit? A) When packing your bag before going into labour. The list explains that hospitals routinely supply a sterilized version of it, making a personal one unnecessary in the delivery bag. B) When packing your bag before going into labour. The infant clothing checklist specifically pairs it with mittens and a blanket to create layered attire that can suit unpredictable weather. C) When finding out where your crush lives. It is advised to keep it switched off during shared rides so that your crush will not suspect you are overly interested in them. D) When finding out where your crush lives. It is suggested as a discreet place to jot down your crush's address while pretending to send a text after glimpsing their driver's license. E) When winning your crush over to the danceperson. Detailed instructions tell you to press it with an iron instead of trying to loosen it with it alone. F) When winning your crush over to the danceperson. It receives no mention whatsoever among the clothing and accessories considered when preparing for the dance.; Passage UIDs: ['7c4ab912fe9944257b101f150b6e9953b', '7c4ab912fe9944257b101f150b6e9953b', 'acf2858c9355465e85c5faa9f327158e', 'acf2858c9355465e85c5faa9f327158e', '195270c9b008 4f83aa08c7aca7b33b11', '195270c9b0084f83aa08c7aca7b33b11']; Passage Titles: ['how to pack your bag before going into labour', 'how to pack your bag before going into labour', 'how to find out where your crush lives', 'how to find out where your crush lives', 'how to win your crush over to the danceperson 5', 'how to win your crush over to the danceperson 5']

Query ID: query_00057; Query Text: In a video that fixing a worn kitchen sink with a wrench, what is the object that the camera wearer rubs? A) faucet B) wrench C) sponge D) kitchen countertop E) sink pipe F) hand; Passage UIDs: None; Passage Titles: None

Query ID: query_00057_2hop; Query Text: In a video that fixing a worn kitchen sink with a wrench, which statement is true about the object that the camera wearer rubs? A) When painting oak cabinets. During finishing, the directions highlight it as the preferred substitute for foam rollers to achieve a smooth coat on door fronts without leaving brush marks. B) When painting oak cabinets. Although many specific tools and materials are listed for refinishing oak cabinets, it is never actually included among the recommended items at any stage of the process. C) When building a simple workbench. To eliminate wobble, builders are told to tighten every L-bracket on the 4×4 legs with it before fastening the bench to a wall. D) When building a simple workbench. It is mentioned as one of the handheld implements that can be stored behind lockable doors on the middle shelf to shield it from the weather. E) When cutting carpet. With stubborn staples or adhesive are encountered, most experts advise melting the bond with a heat gun instead of trying to loosen it with it alone. F) When cutting carpet. A slit about four to five inches long is generally considered wide enough for it to slip through and begin lifting the carpet from the floor.; Passage UIDs: ['aal235d1743d41d6ae5518e5e6789cd1', 'f9917d05531740f7972bcd708e934e42', 'f9917d05531740f7972bcd708e934e42', '0983fd3d87c04d44b41ce94c397fc654', '0983fd3d87c04d44b41ce94c397fc654']; Passage Titles: ['how to paint oak cabinets', 'how to paint oak cabinets', 'how to build a simple workbench', 'how to build a simple workbench', 'how to cut carpet 3', 'how to cut carpet 3']

Query ID: query_00058; Query Text: In a video that is repairing a car chassis in a busy garage, what is the object that the camera wearer puts onto the tractor? A) toolbox B) car battery C) grease gun D) seat cushion E) jack stand F) hose pipe; Passage UIDs: None; Passage Titles: None

Query ID: query_00058_2hop; Query Text: In a video that is repairing a car chassis in a busy garage, which statement is true about the object that the camera wearer puts onto the tractor? A) When replacing brakes on a john deere 5105 tractor. It should be placed directly under the vehicle's final drives to provide the most stable support during brake servicing. B) When replacing brakes on a john deere 5105 tractor. It is best situated beneath the transmission to support the tractor after the floor jack is removed. C) When making a desk chair from a car seat. In the dismantling phase, it is normally fastened by six Torx bolts, all of which have to be removed before any steel cutting can begin. D) When making a desk chair from a car seat. Before the builder marks out the C-lengths of box-section steel, the front-to-back span of the steel plate that it previously sat on is measured to supply the necessary dimension. E) When detailing model car engines on the cheap. Floral wire, pre-painted green and available in multiple gauges, is proposed as an excellent substitute for radiator or heater versions of it.; Passage UIDs: ['86d1949d78e4045949b41c0d5095fddb', '86d1949d78e4045949b41c0d5095fddb', '2f2cd28839d94d82b6c78576c7321987', '2f2cd28839d94d82b6c78576c7321987', '271b70db44ce4ba9bc17643734a5', '271b70db44ce4ba9bc17643734a5']; Passage Titles: ['how to replace brakes on a john deere 5105 tractor', 'how to replace brakes on a john deere 5105 tractor', 'how to make a desk chair from a car seat', 'how to make a desk chair from a car seat', 'how to detail model car engines on the cheap', 'how to detail model car engines on the cheap']

Query ID: query_00059; Query Text: In a video that organizing workshop tools and plugging in a soldering iron, what is the object that camera wearer sticks onto the workbench? A) magnet B) small vise C) power strip D) sheet of sandpaper E) metal clamp F) tape; Passage UIDs: None; Passage Titles: None

Query ID: query_00059_2hop; Query Text: In a video that organizing workshop tools and plugging in a soldering iron, which statement is true about the object that camera wearer sticks onto the workbench? A) When rebuilding an engine. Being molded from flexible nylon, it can be slipped off a hose by hand without the need for pliers or screwdrivers. B) When rebuilding an engine. While detaching coolant hoses, it is actually more difficult to replace than the rubber hoses themselves, so great care is taken not to damage it. C) When painting brake calipers. Automotive technicians recommend an ultra-fine grade of 400 grit or higher for it to avoid scoring the metal of a caliper. D) When painting brake calipers. For proper surface scuffing before the first coat of paint, it is selected in a medium range of about 150 to 200 grit when used on brake calipers. E) When taping handlebars. To strip old material from bicycle handlebars, it should always be peeled off with a razor blade to ensure no residues remain. F) When taping handlebars. Before the main spiral begins, two short pieces of this are usually positioned on the back of each brake lever to provide extra coverage.; Passage UIDs: ['5024a272e085b4b55a9d91f368ca152f8', '5024a272e085b4b55a9d91f368ca152f8', '186bfd680de548e59e2bf832060ea2400', '186bfd680de548e59e2bf832060ea2400', 'f86a07d89c2f4244a8d8fad398ab69f9', 'f86a07d89c2f4244a8d8fad398ab69f9']; Passage Titles: ['how to rebuild an engine', 'how to rebuild an engine', 'how to paint brake calipers', 'how to paint brake calipers', 'how to tape handlebars', 'how to tape handlebars']

Query ID: query_00060; Query Text: In a video that wiping a wall corner baseboard with a purple cloth, what is the object that camera wearer put onto the floor? A) screwdriver B) smartphone C) coffee mug D) paintbrush E) roll of tape F) towel; Passage UIDs: None; Passage Titles: None

Query ID: query_00060_2hop; Query Text: In a video that wiping a wall corner baseboard with a purple cloth, which statement is true about the object that camera wearer put onto the floor? A) When cleaning windows without chemicals. It is singled out as the primary tool for evenly painting the vinegar-and-water solution across every window pane. B) When cleaning windows without chemicals. It is described as a soft-bristled brush that should be kept on hand to knock dust loose while cleaning windows without chemicals. C) When coping with homesickness while traveling. Travel guides emphasize that bringing it adds needless weight, so anyone committed to packing light should definitely exclude it. D) When coping with homesickness while traveling. It is presented as a familiar everyday item that travelers can pack and use to recreate a comforting sense of home whenever homesickness appears. E) When cleaning and organize your tack room. It is suggested as the preferred cloth for wiping clean grooming kits, non-leather stirrups, and other small pieces of synthetic tack. F) When cleaning and organize your tack room. It is not mentioned among the small items advised to be stored in a box, bag, or bucket during reorganization.; Passage UIDs: ['12738felf5el4e199lb87c164b682314', '0fdb8e3a78084b2bbda0f459f02b96c3', '0fdb8e3a78084b2bbda0f459f02b96c3', '68990dbb7daa41df876d852a3eca7345', '68990dbb7daa41df876d852a3eca7345']; Passage Titles: ['how to clean windows without chemicals', 'how to clean windows without chemicals', 'how to cope with homesickness while traveling', 'how to cope with homesickness while traveling', 'how to clean and organize your tack room', 'how to clean and organize your tack room']

Query ID: query_00061; Query Text: In a video that preparing for a nighttime motorcycle ride by grabbing helmet, what is the object that the woman opens from the bike? A) fuel tank cap B) side saddlebag C) top trunk box D) front headlight casing E) battery compartment panel F) bike seat storage; Passage UIDs: None; Passage Titles: None

Query ID: query_00061_2hop; Query Text: In a video that preparing for a nighttime motorcycle ride by grabbing helmet, which statement is true about the object that the woman opens from the bike? A) When carrying cargo on a bike. This attaches directly beneath the saddle on the seat post, functioning mainly as a discreet container for repair tools. B) When carrying cargo on a bike. It is a small bicycle bag that depends on the handlebars and is favored for easy access to items like maps or phones while riding. C) When starting a carbureted , electric start outboard motor. It is meant to stay fully tightened during operation so that no outside air can reach the fuel supply. D) When starting a carbureted , electric start outboard motor. Opening it allows the fuel tank to vent, ensuring adequate fuel flow when the outboard engine is started. E) When making a motorcycle out of old watches. It relies on two hollowed cylinders made from stiff watchbands that can later be snapped over it with a razor blade to ensure no residues remain. F) When making a motorcycle out of old watches. It is finished by placing a G-shaped piece on the rear fender so that the element serves as a backrest directly behind the seat.; Passage UIDs: ['7b1f4e8df40c406fafbe8a9e44ff655', '7b1f4e8df40c406fafbe8a9e44ff655', 'd628cf54df724934b61f146b777c462', 'a2af44590a2d4e558955b0d915a30dbe', 'a2af44590a2d4e558955b0d915a30dbe']; Passage Titles: ['how to carry cargo on a bike 1', 'how to carry cargo on a bike 1', 'how to start a carbureted , electric start outboard motor', 'how to start a carbureted , electric start outboard motor', 'how to make a motorcycle out of old watches', 'how to make a motorcycle out of old watches']

Query ID: query_00062; Query Text: In a video that fixing household items in a cluttered indoor workspace, what is the object that man opens? A) toolbox B) drawer C) paint can D) laptop E) cardboard box F) water bottle; Passage UIDs: None; Passage Titles: None

Query ID: query_00062_2hop; Query Text: In a video that fixing household items in a cluttered indoor workspace, which statement is true about the object that man opens? A) When making an hourglass clock out of light bulbs. It is sliced into narrow strips that are baked at 350 °F alongside the rinsed beach sand to remove any lingering moisture before the sand is added to the bulbs. B) When making an hourglass clock out of light bulbs. It is placed beneath the light bulb during the metal-tip removal so that any falling shards of broken glass are safely caught. C) When swapping out your laptops video card. It usually houses its replaceable video card in an AGP expansion bay that requires no safety latches to be released. D) When swapping out your laptops video card. It secures the video card in a PCI x16 slot from which the card can be slid out once the side latches are undone. E) When making a doll house into a hamster cage. Instead of drilling through the wooden wall of the dollhouse, it can simply be slid into a cardboard tube with a hole cut so that its nozzle protrudes for easy drink access. F) When making a doll house into a hamster cage. The instructions require that this be strictly circular and small-mouthed, that no placed inside the wall-mounted basket to make their more inviting.; Passage UIDs: ['926a1121e94d4e358d2a6fcc062213 78', '926a1121e94d4e358d2a6fcc06221378', '499cbccffd814dfc89a1cb2e7a6d6e39', '499cbccffd814dfc89a1cb2e7a6d6e39', '5bb50ee5135748 24b48a5a7fa16fad50', '5bb50ee5135748 24b48a5a7fa16fad50']; Passage Titles: ['how to make an hourglass clock out of light bulbs', 'how to make an hourglass clock out of light bulbs', 'how to swap out your laptops video card 2', 'how to swap out your laptops video card 2', 'how to make a doll house into a hamster cage', 'how to make a doll house into a hamster cage']

Query ID: query_00063; Query Text: In a video that fixing household items in a cluttered indoor workspace, what is the object that man checks? A) microwave B) washing machine C) vacuum cleaner D) coffee maker E) table lamp F) refrigerator; Passage UIDs: None; Passage Titles: None

Query ID: query_00063_2hop; Query Text: In a video that fixing household items in a cluttered indoor workspace, which statement is true about the object that man checks? A) When decorating. This is singled out as the key accessory in the sophisticated décor style, recommended for creating an upscale ambience. B) When decorating. It is conspicuously missing from the passage's lighting suggestions, which instead direct readers to use a floor lamp for mood illumination. C) When living in your bedroom. Because it is regarded as a serious fire risk, this should never be included among the few appliances selected for a bedroom setup. D) When living in your bedroom. When living in a small room, it is prized for heating quick foods like noodles, steamed vegetables, and canned soups in addition to coffee. E) When getting an elderly person to bathe or shower. In addition to shower temperature, the text advises relocating it into the bathroom as an older adult can easily reach cold beverages without stepping into the kitchen. F) When getting an elderly person to bathe or shower. It is singled out as a highly visible kitchen appliance on which a calendar marking twice-weekly bathing sessions should be affixed to help an older remember to bathe.; Passage UIDs: ['ac85bd4d1d00462b597dfd33e154ffad', '3ae1877e27bf4344b3cd437c9dc3de42', '3ae1877e27bf4344b3cd437c9dc3de42', '9cd4787d5afb481b0e5ba4a6b8571771', '9cd4787d5afb481b0e5ba4a6b8571771']; Passage Titles: ['how to decorate', 'how to decorate', 'how to live in your bedroom', 'how to live in your bedroom', 'how to get an elderly person to bathe or shower 2', 'how to get an elderly person to bathe or shower 2']

Query ID: query_00064; Query Text: In a video that placing a wooden plank on the wall, what is the object that the man picks from the table? A) hammer B) paintbrush C) screwdriver D) metal pipe E) ceramic tile F) wood; Passage UIDs: None; Passage Titles: None

Query ID: query_00064_2hop; Query Text: In a video that placing a wooden plank on the wall, which statement is true about the object that the man picks from the table? A) When painting ceramic. Latex paint is regarded as the most durable option for high-traffic areas when one decides to repaint this. B) When painting ceramic. A thorough repainting process for this involves laying down two coats of oil-based high-adhesion primer and then refining that prominent with 1500- to 2000-grit sandpaper before adding the paint layers. C) When building a seawall. The seawall guide recommends that it measure roughly 2.4 metres in length so that about half of its remains exposed above grade after installation. D) When building a seawall. The plan specifies a single row of them set at fixed intervals before setting the post in the ground. E) When making a squirrel feeder. The project requires only three sections cut from it—a base, a back, and a lid—because the feeder design does not call for separate side panels. F) When making a squirrel feeder. The recommended material for the feeder is it in the form of a cedar fence slat measuring roughly 5 × 12 × 4 feet, which is then divided into the base, back, lid, and side pieces.; Passage UIDs: ['1dda56421cc4c3389aa5cee5254cb78', '1dda56421cc4c3389aa5cee5254cb78', 'b7db0f233d114ed5ae581ca86a35544', 'b7db0f233d114ed5ae581ca86a3554a1', '2b30e5b6cd745f783acd7d805cbf867', '2b30e5b6cd745f783acd7d805cbf867']; Passage Titles: ['how to paint ceramic 2', 'how to paint ceramic 2', 'how to build a seawall', 'how to build a seawall', 'how to make a squirrel feeder', 'how to make a squirrel feeder']

Query ID: query_00065; Query Text: In a video that shows scraping a metal pipe with sandpaper and scraper, what is the object that the camera wearer touches from the floor? A) sandpaper B) wrench C) screwdriver D) paintbrush E) hammer F) scraper; Passage UIDs: None; Passage Titles: None

Query ID: query_00065_2hop; Query Text: In a video that shows scraping a metal pipe with sandpaper and scraper, which statement is true about the object that the camera wearer touches from the floor? A) When making a guinea pig cage. It is described as too hazardous for novice builders, so the instructions advise choosing pre-made frames instead of wielding one. B) When making a guinea pig cage. It is included in the recommended collection of basic tools, alongside an electric drill and screwdriver, for assembling a custom guinea pig hutch. C) When making harry potter stuff. It is paired with black paint to outline a large triangle, a central vertical line, and an inner circle that together create the Deathly Hallows symbol on a white t-shirt. D) When painting an exterior door. This doubles as the frame that holds a small foam roller for evenly applying primer. F) When painting an exterior door. Before the filler is left to dry, this is used to press and smooth bonds or wood filler into the door surface.; Passage UIDs: ['723f36d2f29965491898dea11e5df8450', '723f36d2f29965491898dea11e5df8450', '6cec9b2407c9423182283678373960d6', '6cec9b2407c9423182283678373960d6', '435b3e98d47b4d2ea38c2815ec86d6f3', '435b3e98d47b4d2ea38c2815ec86d6f3']; Passage Titles: ['how to make a guinea pig cage 1', 'how to make a guinea pig cage 1', 'how to make harry potter stuff 2', 'how to make harry potter stuff 2', 'how to paint an exterior door', 'how to paint an exterior door']

Query ID: query_00066; Query Text: In a video that performing home maintenance : vacuuming bedroom and fixing furniture underside, what is the object that woman drops onto the bed? A) phone B) book C) blanket D) laptop E) teddy bear F) pillow; Passage UIDs: None; Passage Titles: None

Query ID: query_00066_2hop; Query Text: In a video that performing home maintenance : vacuuming bedroom and fixing furniture underside, which statement is true about the object that woman drops onto the bed? A) When cleaning your room as teens. Readers are advised to store it inside the wardrobe so the bed remains uncluttered after tidying up. B) When cleaning your room as teens. This is offered as a small decorative object that can be placed on a desk or on the bed once the room has been cleaned and organized. C) When swapping out your laptops video card. It can be dismantled as soon as the power is switched off because its components remain cool enough to handle immediately. D) When swapping out your laptops video card. It secures the video card in a PCI x16 slot from which the card can be slid out once the side latches are undone. E) When making a cat wall bed from a basket. The instructions require that this be strictly circular and small-mouthed, that no placed inside the wall-mounted basket to make their more inviting. F) When making a cat wall bed from a basket. The two interchangeable soft liners—along with a blanket—that can be placed inside the wall-mounted basket to swap out the basket's visual theme.; Passage UIDs: ['6328e5ea5c164bdcb6ca48bd52dc5e98', '499cbccffd814dfc89a1cb2e7a6d6e39', '499cbccffd814dfc89a1cb2e7a6d6e39', '7e0d65b261084a3ba8b957d6b9f161ld', '7e0d65b261084a3ba8b957d6b9f161ld']; Passage Titles: ['how to clean your room teens', 'how to clean your room teens', 'how to swap out your laptops video card 2', 'how to swap out your laptops video card 2', 'how to make a cat wall bed from a basket', 'how to make a cat wall bed from a basket']

Query ID: query_00067; Query Text: In a video that shows organizing scattered artwork and papers on the floor, what is the object that the camera wearer opens from the book? A) bookmark B) photograph C) small envelope D) dried flower E) DVD disc F) paper; Passage UIDs: None; Passage Titles: None

Query ID: query_00067_2hop; Query Text: In a video that shows organizing scattered artwork and papers on the floor, which statement is true about the object that the camera wearer opens from the book? A) When personalizing your locker. To keep your style flexible, you can stick magnetic letters directly on it and rearrange them whenever you locker theme. B) When personalizing your locker. The suggested use is to slide it beneath the locker's ventilation slots, allowing friends to discreetly drop written notes inside. C) When writing a personal history. The advice suggests excluding it from your research spreadsheet unless it was professionally shot and carries precise details of information. D) When writing a personal history. You must have playing any kind of background music, even a quiet sleep-music station, will inevitably alert your parents to your activity. This warns you to gather your books, electronics, a snack, water, and a flashlight in advance so you can work through the night without needing to leave your room.; Passage UIDs: ['f6b80a2a3184b6286223b2d6876b157', 'f6b80a2a3184b6286223b2d6876b157', '1280434a449f4cb48f7c406ad441f90c', '1280434a449f4cb48f7c406ad441f90c', '3648e380e01 4010ba4177cdfaacec91', '3648e380e014010ba4177cdfaacec91']; Passage Titles: ['how to personalize your locker', 'how to personalize your locker', 'how to write a personal history 2', 'how to write a personal history 2', 'how to stay up late at night writing a paper without your parents knowing']

Query ID: query_00068; Query Text: In a video that shows organizing scattered artwork and papers on the floor, what is the object that the camera wearer takes from the book on the lap? A) pen B) bookmark C) smartphone D) pair of scissors E) small ruler F) paper; Passage UIDs: None; Passage Titles: None

Query ID: query_00068_2hop; Query Text: In a video that shows organizing scattered artwork and papers on the floor, which statement is true about the object that the camera wearer takes from the book on the lap? A) When coping in school. It is suggested that students leave it in their locker with their PE kit and extra stationery so their locker stays tidy during random checks. B) When coping in school. It is listed as an essential tool to keep in a pencil case alongside pens, pencils, a ruler, an eraser, a sharpener, a glue stick and sticky notes. C) When preventing confusion. The material advises people to keep their daily errands and appointment lists on it so they can consult them wherever they go. E) When staying up late at night writing a paper without your parents knowing. This warns that playing any kind of background music, even a quiet sleep-music station, will inevitably alert your parents to your activity. F) When staying up late at night writing a paper without your parents knowing. This advises you to gather your books, electronics, a snack, water, and a flashlight in advance so you can work through the night without needing to leave your room.; Passage UIDs: ['30449140247740305a0f8cca30 98057', '30449140247740305a0f8cca3098057', '0f281ab4aa8c4e9ab9338453aef406b', '0f281ab4aa8c4e9ab9338453aef406b', '3648e380e014010ba4177cdfaacec91', '3648e380e014010ba4177cdfaacec91']; Passage Titles: ['how to cope in school', 'how to cope in school', 'how to prevent confusion', 'how to prevent confusion', 'how to stay up late at night writing a paper without your parents knowing', 'how to stay up late at night writing a paper without your parents knowing']

Query ID: query_00069; Query Text: In a video that assembling and organizing picture frames on cluttered floor, what is the object that the camera wearer removes from the photo frame? A) glass pane B) cardboard backing C) picture nail D) price tag E) hanging wire F) photo; Passage UIDs: None; Passage Titles: None

Query ID: query_00069_2hop; Query Text: In a video that assembling and organizing picture frames on cluttered floor, which statement is true about the object that the camera wearer removes from the photo frame? A) When arranging pictures on a wall. When adhesive hanging strips are used, it is normally removed entirely because the strips take over all weight-bearing duties. B) When arranging pictures on a wall. During installation, the nail is passed underneath it and looped through, letting the frame rest on that support while final adjustments are made. C) When hanging things from parents. People sometimes stitch it into the lining of a coat

sleeve, turning the stiff strip into a hidden slide-in sheath for money or notes. E) When hiding things from parents. Taping it all the way around a personal safe can make the safe look like an ordinary cardboard storage box and keep parents from realizing what the container really is.; E) When starting a best friends club for kids. It should never be placed on public posters because doing so would give away confidential information about the club. F) When starting a best friends club for kids. It may be displayed on school flyers or stapled posters, accompanied by the person's full name and other simple details, when seeking new members.; Passage UIDs: ['2492b4cba109420380a9c2afa91baa41', '2492b4cba109420380a9c2afa91baa41', '5901d57f17f74e33a9847e7c2f43b77', '5901d57f17f74e33a9847e7c2f43b77', 'a84928b047ae04bc09702a613d9d049ac']; Passage Titles: ['how to arrange pictures on a wall', 'how to arrange pictures on a wall', 'how to hide things from parents', 'how to hide things from parents', 'how to start a best friends club for kids', 'how to start a best friends club for kids']

Query ID: query_00070; Query Text: In a video that storing groceries in a cabinet and picking fruit, what is the object that the camera wearer washes from the sink in the sink? A) apple B) plate C) spoon D) lettuce E) cup F) hand; Passage UIDs: None; Passage Titles: None

Query ID: query_00070_2hop; Query Text: In a video that storing groceries in a cabinet and picking fruit, which statement is true about the object that the camera wearer washes from the sink in the sink? A) When cooking frozen lobster tails. It must be preheated so the Maine lobster tails can rest inside it before they are lowered into the butter bath. B) When cooking frozen lobster tails. It is employed to measure the amount of water that had been holding the lobster tails as an equal volume of butter can be melted for the beurre monté. C) When cooking italian food. After bringing it home, an Italian who values freshness will leave the unwashed head on the kitchen counter, confident that, if used within a few days, it will remain in better condition than if it had been refrigerated. D) When preventing kitchen burns. Standard kitchen safety guidelines mandate that it always be enclosed in an oven mitt whenever removing any container from a microwave oven. F) When preventing kitchen burns. Because hot oil can shoot several feet out of a pan, it is susceptible to burns unless the cook stands several feet away while frying.; Passage UIDs: ['7747b589dfe2b4f01b07dd3aaa444f77a', '7747b589dfe2b4f01b07dd3aaa444f77a', 'e031cf90a197456b0b9ab62d5466bfa', 'e031cf90a197456b0b9ab62d5466bfa', 'fe5b6a1a9bde453ab1aa5a3daf663ea4', 'fe5b6a1a9bde453ab1aa5a3daf663ea4']; Passage Titles: ['how to cook frozen lobster tails', 'how to cook frozen lobster tails', 'how to cook italian food', 'how to cook italian food', 'how to prevent kitchen burns 3', 'how to prevent kitchen burns 3']

Query ID: query_00071; Query Text: In a video that storing groceries in a cabinet and picking fruit, what is the object that camera wearer pulls from the cooker? A) saucepan B) pot lid C) spatula D) baking tray E) oven mitt F) hand towel; Passage UIDs: None; Passage Titles: None

Query ID: query_00071_2hop; Query Text: In a video that storing groceries in a cabinet and picking fruit, which statement is true about the object that camera wearer pulls from the cooker? A) When safelying cook chicken from frozen. Covering the bird with it during the initial ninety minutes of roasting helps keep the heat moist and shortens the overall cook time. B) When safelying cook chicken from frozen. After the chicken has baked for about forty-five minutes, it is advised to use tongs in one hand and this glove on the other to pull out the giblets before adding stuffing. C) When making easy homemade biscuits. The recipe instructs bakers to preheat it in the oven so that the biscuits start cooking the instant they touch the metal. D) When making easy homemade biscuits. For this biscuit recipe, it does not require greasing or any other preparation, although the cook may choose to do so if desired. E) When decorating a kitchen. It is advised to be kept exclusively in plain white to preserve a clean, hospital-like atmosphere in the kitchen. F) When decorating a kitchen. This is recommended to carry the same pattern as the curtains and tablecloth to give the kitchen a cohesive theme.; Passage UIDs: ['727f4ed80932a4656910958d568e89c1', '727f4ed80932a4656910958d568e89c1', 'f1fac3e60a734dab97f8b379fd300bf8', 'f1fac3e60a734dab97f8b379fd300bf8', 'e44e552f7d2c4d479cfa19216f4f51b0', 'e44e552f7d2c4d479cfa19216f4f51b0']; Passage Titles: ['how to safely cook chicken from frozen 1', 'how to safely cook chicken from frozen 1', 'how to make easy homemade biscuits 2', 'how to make easy homemade biscuits 2', 'how to decorate a kitchen 1', 'how to decorate a kitchen 1']

Query ID: query_00072; Query Text: In a video that storing groceries in a cabinet and picking fruit, what is the object that camera wearer wipes? A) cabinet door B) kitchen counter C) shelf surface D) apple E) shopping bag F) hand; Passage UIDs: None; Passage Titles: None

Query ID: query_00072_2hop; Query Text: In a video that storing groceries in a cabinet and picking fruit, which statement is true about the object that camera wearer wipes? A) When cooking food in sims. The level-10 baked angel food cake relies on two eggs and it as its only required fruit component. B) When cooking food in sims. French toast at cooking level 9 is prepared with two of it plus a single egg and is served during breakfast or brunch. C) When making johnnycake. It is preheated in the oven so the underside of each johnnycake begins browning the instant the rounds are set upon it. D) When making johnnycake. It is lightly dusted with flour solely to hold the portioned balls of dough until they are later stretched and fried rather than baked. E) When making quick stuffed peppers turkey. Prior to baking, the chef is instructed to dust it lightly with shredded cheese so melted strands will cling to the stencil during serving. F) When making quick stuffed peppers turkey. The cook is cautioned to protect it with oven mitts or pot holders because severe burns would make holding a fork painful.; Passage UIDs: ['df63fe1bbf0d4f61bc913d2bcb5b1aa0', 'df63fe1bbf0d4f61bc913d2bcb5b1aa0', '5b9b78a857484364aa499c140961b0d50', '5b9b78a857484364aa499c140961b0d50', '5530a150781864f4b25740597539125f', '5530a150781864f4b25740597539125f']; Passage Titles: ['how to cook food in sims 3', 'how to cook food in sims 3', 'how to make johnnycake 3', 'how to make johnnycake 3', 'how to make quick stuffed peppers turkey', 'how to make quick stuffed peppers turkey']

Query ID: query_00073; Query Text: In a video that packing a bag with a gaming console, what is the object that the camera wearer holds? A) game controller B) laptop C) water bottle D) headphones E) pair of sneakers F) bag; Passage UIDs: None; Passage Titles: None

Query ID: query_00073_2hop; Query Text: In a video that packing a bag with a gaming console, which statement is true about the object that the camera wearer holds? A) When going back to school shopping. It is deemed unnecessary if a student already possesses ballet flats, which are said to provide equivalent support for physical education classes. B) When going back to school shopping. Students in middle and high school are advised to purchase an extra one specifically for gym class, particularly when their regular shoes are Converse. C) When staying up late without waking your parents. It is described as too noisy to use when attempting to stay up late without waking your parents. D) When staying up late without waking your parents. Using it in only one ear is suggested so the other ear can remain alert for parents moving around at night. E) When getting ready for a first date in middle school. It is recommended that selecting one made of waterproof fabric is sensible if the plan is to meet at the beach. F) When getting ready for a first date in middle school. It is referred to solely as the puffy area under your eyes that you should make sure not to have before a first middle-school date.; Passage UIDs: ['6120815722a21489f850f81a5747249a3', '6120815722a21489f850f81a5747249a3', '49db7cf8f4d7487fa6a808c753295f5e', '49db7cf8f4d7487fa6a808c753295f5e', '055f61c36e5e4a6c96d902fa76a4e321', '055f61c36e5e4a6c96d902fa76a4e321']; Passage Titles: ['how to go back to school shopping', 'how to go back to school shopping', 'how to stay up late without waking your parents', 'how to stay up late without waking your parents', 'how to get ready for a first date in middle school', 'how to get ready for a first date in middle school']

Query ID: query_00074; Query Text: In a video that welding a car's metal part in dim garage, what is the object that the camera wearer opens? A) toolbox B) car hood C) garage door D) vice grip E) safety goggles F) welding mask; Passage UIDs: None; Passage Titles: None

Query ID: query_00075; Query Text: In a video that repairing and cleaning a bicycle wheel with tools, what is the object that the camera wearer folds? A) microfiber cloth B) sandpaper sheet C) instruction manual D) rag towel E) plastic bag F) tissue paper; Passage UIDs: None; Passage Titles: None

Query ID: query_00076; Query Text: In a video that repairing a wooden fence by hammering and pulling nails, what is the object that the camera wearer picks from the ground? A) hammer B) screwdriver C) metal bracket D) paintbrush E) measuring tape F) wooden picket; Passage UIDs: None; Passage Titles: None

Query ID: query_00077; Query Text: In a video that wearing, removing, and handling masks around a cluttered workshop, what is the object that the camera wearer removes from his face? A) pair of safety goggles B) blue respirator C) yellow hard hat D) black face shield E) red bandana F) white mask; Passage UIDs: None; Passage Titles: None

Query ID: query_00078; Query Text: In a video that removing a bicycle tire from using tire levers, what is the object that the camera wearer puts onto the iron plate? A) screwdriver B) bicycle chain C) hand pump D) water bottle E) helmet F) bike tire remover; Passage UIDs: None; Passage Titles: None

Query ID: query_00079; Query Text: In a video that shows someone sanding a wooden furniture piece at home, what is the object that camera wearer sands from wooden rack? A) wooden rack leg B) wooden tabletop C) metal chair frame D) wooden drawer front E) plastic storage bin F) wooden cork edge; Passage UIDs: None; Passage Titles: None

Query ID: query_00080; Query Text: In a video that repairing a mountain bike in a workshop, what is the object that the camera wearer straightens? A) bicycle chain B) bicycle seat C) bicycle pedal D) bicycle brake cable E) bicycle handlebar F) bicycle tire; Passage UIDs: None; Passage Titles: None

Query ID: query_00081; Query Text: In a video that drilling a wooden shoe rack with a cordless drill, what is the object that camera wearer lifts from the right hand? A) Screwdriver B) Measuring tape C) Bottle of wood glue D) Wooden dowel E) Small bag of screws F) Emery paper; Passage UIDs: None; Passage Titles: None

Query ID: query_00082; Query Text: In a video that vacuuming a bedroom floor with a cordless vacuum, what is the object that the woman opens? A) door B) drawer C) closet D) cabinet E) wardrobe F) window; Passage UIDs: None; Passage Titles: None

Query ID: query_00083; Query Text: In a video that is repairing an electronic component on a cluttered workbench, what is the object that the camera wearer flips? A) soldering iron B) rectangular circuit board C) roll of electrical tape D) small screwdriver E) digital multimeter F) cylindrical can; Passage UIDs: None; Passage Titles: None

Query ID: query_00084; Query Text: In a video that shows repairing furniture underneath a desk with hand tools, what is the object that the camera wearer places onto the wooden slab? A) screwdriver B) hammer C) measuring tape D) glue bottle E) wooden block F) left hand; Passage UIDs: None; Passage Titles: None

Query ID: query_00085; Query Text: In a video that is entering a car and preparing for basketball outing, what is the object that person touch? A) car keys B) basketball C) water bottle D) steering wheel E) sunglasses F) phone; Passage UIDs: None; Passage Titles: None

Query ID: query_00086; Query Text: In a video that organizing a whiteboard with magnets erasing and writing, what is the object that the camera wearer put into the bag on a chair? A) laptop computer B) Tripod stand C) Rolled-up yoga mat D) Water bottle E) Folded umbrella F) Folding stool; Passage UIDs: None; Passage Titles: None

Query ID: query_00087; Query Text: In a video that arranging jackets and bags inside an elevator, what is the object that the girl lifts from the left hand? A) scarf B) water bottle C) cell phone D) clipboard E) sunglasses F) ziplock; Passage UIDs: None; Passage Titles: None

Query ID: query_00088; Query Text: In a video that microwaving coffee in the kitchen microwave, what is the object that the camera wearer kicks from the floor? A) kitchen towel B) coffee mug C) soccer ball D) pair of slippers E) cardboard box F) painting; Passage UIDs: None; Passage Titles: None

Query ID: query_00089; Query Text: In a video that loading baseball equipment into a car in a garage, what is the object that the camera wearer searches? A) baseball bat B) toolbox C) helmet D) duffel bag E) glove F) locker; Passage UIDs: None; Passage Titles: None

Query ID: query_00090; Query Text: In a video that loading baseball equipment into a car in a garage, what is the object that camera wearer opens from the garage in the garage? A) the car's trunk B) the car hood C) the glove compartment D) the front passenger door E) the garage door F) the car driver door; Passage UIDs: None; Passage Titles: None

Query ID: query_00091; Query Text: In a video that organizing belongings around the house, what is the object that the camera wearer touches from the side of the board? A) piece of chalk B) dry-erase marker C) sticky note pad D) small magnet E) pair of scissors F) eraser; Passage UIDs: None; Passage Titles: None

Query ID: query_00092; Query Text: In a video that shows repairing a car engine under dim garage lighting, what is the object that camera wearer uncovers? A) spark plug B) oil filter C) car battery D) air filter E) radiator cap F) fuse; Passage UIDs: None; Passage Titles: None

Query ID: query_00093; Query Text: In a video that wiping a knife and closing a plastic container, what is the object that the camera wearer removes? A) paper towel B) plastic lid C) cutting board D) latex glove E) dish cloth F) knife cover; Passage UIDs: None; Passage Titles: None

Query ID: query_00094; Query Text: In a video that wiping a knife and closing a plastic container, what is the object that the camera wearer places into the box? A) sponge B) cutting board C) apple slice D) paper towel E) wooden spoon F) knife cover; Passage UIDs: None; Passage Titles: None

Query ID: query_00095; Query Text: In a video that sorting household items into a large storage bin, what is the object that the man picks? A) ceramic vase B) winter coat C) garden hose D) board game box E) electric kettle F) wedding album; Passage UIDs: None; Passage Titles: None

Query ID: query_00096; Query Text: In a video that fixing a worn kitchen sink with a wrench, what is the object that camera wearer put into the sink? A) metal wrench B) yellow sponge C) blue coffee mug D) stainless steel pot lid E) green dishcloth F) hand wash; Passage UIDs: None; Passage Titles: None

Query ID: query_00097; Query Text: In a video that repairing an automotive starter motor on a dirt floor, what is the object that the camera wearer touches? A) wrench B) spark plug C) car battery D) oil filter E) pliers F) screw driver; Passage UIDs: None; Passage Titles: None

Query ID: query_00098; Query Text: In a video that organizing workshop tools and plugging in a soldering iron, what is the object that the camera wearer puts into the drawer? A) screwdriver B) hammer C) paintbrush D) pliers E) hot glue gun F) masking tape; Passage UIDs: None; Passage Titles: None

Query ID: query_00099; Query Text: In a video that organizing workshop tools and plugging in a soldering iron, what is the object that the camera wearer picks up from the floor? A) hammer B) tape measure C) safety goggles D) extension cord E) power drill F) electric soldering iron; Passage UIDs: None; Passage Titles: None

Query ID: query_00100; Query Text: In a video that organizing workshop tools and plugging in a soldering iron, what the camera wearer untangle? A) Tape measure B) Green garden hose C) Orange extension cord for power tools D) Black headphone earbuds right E) Electric soldering iron wire; Passage UIDs: None; Passage Titles: None

Query ID: query_00101; Query Text: In a video that shows measuring and marking metal pieces on a workbench, what is the object that the camera wearer adjusts? A) tape measure B) digital caliper C) C-clamp D) metal protractor E) wooden ruler F) 1 square, metal; Passage UIDs: None; Passage Titles: None

Query ID: query_00102; Query Text: In a video that preparing a car and packing sports gear for travel, what is the object that the camera wearer closes? A) car door B) garage door C) cooler lid D) backpack E) drawer F) shelf; Passage UIDs: None; Passage Titles: None

Query ID: query_00103; Query Text: In a video that cleaning a living room table with spray cleaner, what is the object that the camera wearer places onto the table? A) coffee mug B) remote control C) decorative vase D) hardcover book E) smartphone F) tumbler; Passage UIDs: None; Passage Titles: None

Query ID: query_00104; Query Text: In a video that carrying and loading wooden chairs into a car trunk, what is the object that camera wearer talks? A) woman B) dog C) wooden chair D) car trunk E) tree F) man; Passage UIDs: None; Passage Titles: None

Query ID: query_00105; Query Text: In a video that installing a lampshade onto a lamp fixture, what is the object that the camera wearer places onto the tape? A) screwdriver B) hammer C) wrench D) tape measure E) paintbrush F) plier; Passage UIDs: None; Passage Titles: None

Query ID: query_00106; Query Text: In a video that repairing equipment and rummaging through boxes in cluttered workshop, what is the object that camera wearer disconnects? A) electric drill B) soldering iron station C) air compressor D) table saw E) 3D printer F) knife sharpener machine; Passage UIDs: None; Passage Titles: None

Query ID: query_00107; Query Text: In a video that assembling a baby playard on the living room floor, what is the object that the camera wearer moves from the floor? A) white pillow B) blue blanket C) cardboard box D) baby stroller E) stuffed teddy bear F) black bag; Passage UIDs: None; Passage Titles: None

Query ID: query_00108; Query Text: In a video that cutting a bike cable and adjusting the rear derailleur, what is the object that camera wearer put into the tool kit? A) steel bolt B) rubber grommet C) plastic zip tie D) aluminum spoke E) copper washer F) brass nipple; Passage UIDs: None; Passage Titles: None

Query ID: query_00109; Query Text: In a video that organizing scattered artwork and papers on the floor, what is the object that the camera wearer turns from the lap? A) tablet screen B) laptop keyboard C) sketchpad cover D) clipboard E) magazine page F) book page; Passage UIDs: None; Passage Titles: None

Query ID: query_00110; Query Text: In a video that retrieving bananas and preparing a blender on kitchen counter, what is the object that the camera wearer passes from her left hand to her right hand? A) banana B) knife C) blender lid D) measuring cup E) dish towel F) cutting board; Passage UIDs: None; Passage Titles: None

Query ID: query_00111; Query Text: In a video that tightening a bicycle pedal with a wrench, what is the object that the camera wearer dusts off? A) bicycle seat B) handlebar grip C) front tire D) water bottle E) brake lever F) bicycle gear; Passage UIDs: None; Passage Titles: None

Query ID: query_00112; Query Text: In a video that is folding a cable and placing it in nightstand, what is the object that the camera wearer drops onto the shelf in the bedroom? A) cell phone B) hairbrush C) water bottle D) pair of sunglasses E) hardcover book F) wire cord; Passage UIDs: None; Passage Titles: None

Query ID: query_00113; Query Text: In a video that selecting tools and repairing a household device in workshop. After the camera wearer moves a tool rack with his hand, what is the object that camera wearer moves? A) power drill B) wooden board C) ladder D) paint bucket E) tool rack; Passage UIDs: None; Passage Titles: None

Query ID: query_00113_2hop; Query Text: In a video that selecting tools and repairing a household device in workshop. After the camera wearer moves a tool rack with his hand, which statement is true about the object that camera wearer moves? A) When choosing neutral paint colors to prepare a house for sale. To keep the paint fresh, it should be stored completely full and left open in an occupied bedroom after use. B) When choosing neutral paint colors to prepare a house for sale. During the painting process, only about one-tenth of it is poured into a paint tray before loading the roller. C) When building a fort in 7 days to die. When stretched across a moat, it acts as a damaging barrier that injures any creature attempting to traverse it. D) When building a fort in 7 days to die. Suspending it beneath a bridge offers players a personal route into a stronghold while remaining unclimbable by zombies. E) When organizing kitchen cabinets. It is recommended as a wall-mounted fixture placed beside the kitchen's entrance to help frequently-used pots and pans within easy reach. F) When organizing kitchen cabinets. It is not specifically included among the organizational aids that encompass utensil trays, cup hooks, dish-drying racks, and dry-goods canisters.; Passage UIDs: ['121463865004c44d983e1f7ecc0a784c8', '121463865004c44d983e1f7ecc0a784c8', '5a23ead39bba24607799ef6101a18c8497', '5a23ead39bba24607799ef6101a18c8497', '808dc1f9cf44e5e58f0e55d60bdc1720', '808dc1f9cf44e5e58f0e55d60bdc1720']; Passage Titles: ['how to choose neutral paint colors to prepare a house for sale', 'how to choose neutral paint colors to prepare a house for sale', 'how to build a fort in 7 days to die', 'how to build a fort in 7 days to die', 'how to organize kitchen cabinets', 'how to organize kitchen cabinets']

Query ID: query_00114; Query Text: In a video that selecting tools and repairing a household device in workshop. Before the camera wearer moves a tool rack with his hand, what is the object that camera wearer moves? A) hammer B) power drill C) screwdriver set D) pliers E) toolbox F) tool rack; Passage UIDs: None; Passage Titles: None

Query ID: query_00114_2hop; Query Text: In a video that selecting tools and repairing a household device in workshop. Before the camera wearer moves a tool rack with his hand, which statement is true about the object that camera wearer moves? A) When maximizing workspace in a small garage. A fishing tackle box can serve in place of it, providing an organized container for small power tools and their accessory bits. C) When easilying modify the house reveals. B) When maximizing workspace in a small garage. Installing extra-long air hoses along every wall will largely make this unnecessary, as most small attachments can simply be hung directly from the hose reels. D) When maximizing workspace in a small garage. A fishing tackle box can serve in place of it, providing an organized container for small power tools and their accessory bits. C) When easilying modify the house reveals. It is chiefly employed in the instructions to remove the assorted screws that hold the layout plug through the dish rack while it drips alongside other utensils. so that the user can reach inside the plunger tube and clip through the three prongs that secure an internal plastic insert. E) When heavily modify a part firmly? D) When maximizing workspace in a small garage', 'how to easily modify a nerf longshot 2', 'how to hang a pot rack 2']; Passage Titles: ['how to maximize workspace in a small garage', 'how to easily modify a nerf longshot 2', 'how to hang a pot rack 2']

Query ID: query_00115; Query Text: In a video that selecting tools and repairing a household device in workshop. After the camera wearer moves a tool rack with his hand, what is the object that camera wearer moves? A) toolbox B) ladder C) cardboard box D) power drill E) workbench F) tool rack; Passage UIDs: None; Passage Titles: None

Query ID: query_00115_2hop; Query Text: In a video that selecting tools and repairing a household device in workshop. After the camera wearer moves a tool rack with his hand, which statement is true about the object that camera wearer moves? A) When building a simple chicken coop. The instructions advise constructing it with a sand-infused paint layer so that lumber doesn't slide while being cut. B) When building a simple chicken coop. It is explicitly listed among the essential tools—along with a measuring tape, sander, power saw, hammer, and drill—required for constructing the simple chicken coop. C) When making a nunchaku. It is used to hollow out the interior of the pipes so the finished weapons remain light and safe. D) When making a nunchaku. It is used during the necessary tools, which instead include various saws, screw eyes, pliers, cement, and electrical tape. E) When organizing kitchen cabinets. It is described as a fabric pouch filled with baking soda or eucalyptus that can be tucked into cabinet corners to absorb odors and deter insects. F) When organizing kitchen cabinets. It is not specifically included among the organizational aids that encompass utensil trays, cup hooks, dish-drying racks, and dry-goods canisters.; Passage UIDs: ['d7a4e8c9c6e54586af0c0f354cebdc', '7d7a4e8c9c6e54586af0c0f355c4cebdc', 'e464566fb4344f399a107b2b32ce5e77', 'e464566fb4344f399a107b2b32ce5e77', '808dc1f9cf44e5e58f0e55d60bdc1720']; Passage Titles: ['how to build a simple chicken coop', 'how to build a simple chicken coop', 'how to make a nunchaku 2', 'how to make a nunchaku 2', 'how to organize kitchen cabinets', 'how to organize kitchen cabinets']

Query ID: query_00116; Query Text: In a video that fixing a device on a living room floor. Before a woman plays with A baby, what is the object that woman places onto the neck? A) scarf B) headphones C) necklace D) baby bottle E) remote control F) hand; Passage UIDs: None; Passage Titles: None

Query ID: query_00116_2hop; Query Text: In a video that fixing a device on a living room floor. Before a woman plays with A baby, which statement is true about the object that woman places onto the neck? A) When childing proof a living room. Because its components are considered too large to swallow, safety experts rarely list it as a choking danger for infants. B) When childing proof a living room. Its small buttons can detach and end up under furniture, creating a choking hazard for crawling babies who might find them and put them in their mouth. C) When reusing household objects for cleaning. Spritzing the interior of it with common spray cleaner before cloth for reaching inside of sticking surfaces for close. D) When making household objects for cleaning. It is placed inside the sock, with the fingers holding the catnip dangling at the bottom, before the fabric is twisted closed.; Passage UIDs: ['52cc9c1d77714cd29b21b910ba35fbe6d', '0781b70b6d02483298383f5bf904aecd', '0781b70b6d02483298383f5bf904aecd', '5b54eddb5cee42a3acf80c521e62dda6', '5b54eddb5cee42a3acf80c521e62dda6']; Passage Titles: ['how to child proof a living room', 'how to child proof a living room', 'how to reuse household objects for cleaning', 'how to reuse household objects for cleaning', 'how to make cat toys out of common household items 4', 'how to make cat toys out of common household items 4']

Query ID: query_00117; Query Text: In a video that playing baseball by self-tossing and batting the ball. Before the a man stands, what is the the man picks from the ground? A) baseball bat B) glove C) cap D) water bottle E) smartphone F) ball; Passage UIDs: None; Passage Titles: None

Query ID: query_00117_2hop; Query Text: In a video that playing baseball by self-tossing and batting the ball. Before the a man stands, which statement is true about the the man picks from the ground? A) When throwing a curveball. Pitchers learning the knuckle curve are often told to remove it from their non-throwing hand before forming the grip so the batter cannot read the pitch. B) When throwing a curveball. In mastering the knuckle curve, the ball should first be seated deep inside it before the fingers assume their specialized grip. C) When hitting a slowpitch softball. To maximize leverage for a left-handed swing, the correct technique is to place the right hand above the left when gripping it. D) When hitting a slowpitch softball. A batter seeking firmer control should grip it close to the handle and may choke up slightly during the swing. E) When playing basketball. During basketball games, it is customarily left resting on the chalk-marked home base between innings to show which team is up to shoot. F) When playing basketball. At the very beginning of a basketball match, the competitor who first picks it up—before the final line is given it to start play.; Passage UIDs: ['96ce59d25a41f4009e54cef4a69b4a4a', '96ce59d25a41f4009e54cef4a69b4a4a', '2cf85ba90c094a01a139b0d50c8428e', '2cf85ba90c094a01a139b0d50c8428e', 'e0561fe5cd0d4dd8915b94984f2acd26a']; Passage Titles: ['how to throw a curveball 3', 'how to throw a curveball 3', 'how to hit a slowpitch softball', 'how to hit a slowpitch softball', 'how to play basketball']

Query ID: query_00118; Query Text: In a video that repairing and cleaning a bicycle wheel with tools. After the camera wearer Looks around a garage, what is the object that the camera wearer picks up from the shelf? A) wrench B) bicycle tire C) rag D) helmet E) chain F) bottle; Passage UIDs: None; Passage Titles: None

Query ID: query_00118_2hop; Query Text: In a video that repairing and cleaning a bicycle wheel with tools. After the camera wearer Looks around a garage, which statement is true about the object that the camera wearer picks up from the shelf? A) When teaching your toddler to pedal a bike. When fitting it on a young rider, the front rim should sit roughly two finger-widths above the child's eyebrows to provide optimal sightlines. B) When teaching your toddler to pedal a bike. For a child's safe ride, it needs to rest perfectly level on the forehead and be tight enough to barely wiggle yet loose enough not to feel glued to the head. C) When replacing brakes on a john deere 5105 tractor. During brake service on a John Deere 5105 tractor, it must be wedged between the transmission and the final drive to keep components from shifting before the cherry picker is attached. D) When replacing brakes on a john deere 5105 tractor. Throughout the detailed procedure for replacing the brakes on a John Deere 5105 tractor, it is never included among the tools or supplies the mechanic is told to have on hand. E) When checking the fluids in a car. It exclusively holds windshield washer fluid, making its maintenance unimportant for the operation of the engine or transmission. F) When checking the fluids in a car. It often carries a label explaining whether the windshield washer fluid inside must be diluted with water or poured in at full strength.; Passage UIDs: ['650cf1865245d4eba70c742f4af1caaaf', '8d1949d78e404594fb41fcd0509fdb', '8d1949d78e404594fb41fcd0509fdb', '015dbb412b374d6f959245a17bd69fc', '015dbb412b374d6f959245a17bd69fc']; Passage Titles: ['how to teach your toddler to pedal a bike', 'how to teach your toddler to pedal a bike', 'how to replace brakes on a john deere 5105 tractor', 'how to replace brakes on a john deere 5105 tractor', 'how to check the fluids in a car']

Query ID: query_00119; Query Text: In a video that repairing a car wheel and brakes using tools. Before the camera wearer bends besides the car, what is the object that the camera wearer picks from the bonnet? A) screwdriver B) jack handle C) pair of pliers D) flashlight E) oil can F) wheel spanner; Passage UIDs: None; Passage Titles: None

Query ID: query_00119_2hop; Query Text: In a video that repairing a car wheel and brakes using tools. Before the camera wearer bends besides the car, which statement is true about the object that the camera wearer picks from the bonnet? A) When checking oil level in car. It is commonly molded onto the dipstick so owners can distinguish it from the engine-oil dipstick. B) When checking oil level in car. Most cars display it as an old-style graphic on or near the engine-oil dipstick, marking the correct stick for checking oil level. C) When repairing an iphone from water damage. To accelerate drying of a soaked iPhone, some repair techniques call for leaving it switched on and positioned close to the charging port so that the warmth of its lamp drives moisture out. D) When repairing an iphone from water damage. The only role it plays in diagnosing water damage is illuminating the SIM-tray or charging port so the liquid-contact marker that turns red can be seen clearly. E) When checking your battery on your iphone. It is conspicuously absent from the recommended emergency-tool list that includes a flashlight, screwdrivers, pliers and an adjustable spanner for road trips.; Passage UIDs: ['9b1c43ce0f54e028231e0b85321ea4e', '9b1c43ce0f54e028231e0b85321ea4e', '8d912a198b5844d6af6f57af8f1bb9a5', '640c06aa084849b8875859a925934399']; Passage Titles: ['how to check oil level in car', 'how to check oil level in car', 'how to repair an iphone from water damage', 'how to repair an iphone from water damage', 'how to check your car before a road trip']

Query ID: query_00120; Query Text: In a video that repairing household items on a cluttered workbench. After the camera wearer places the glue on the table with his right hand, what is the object that camera wearer passes from the left hand to the right hand? A) silver screwdriver B) red utility knife C) blue tape roll D) small wooden dowel E) white plastic cap F) black cardboard cut-out; Passage UIDs: None; Passage Titles: None

Query ID: query_00120_2hop; Query Text: In a video that repairing household items on a cluttered workbench. After the camera wearer places the glue on the table with his right hand, which statement is true about the object that the camera wearer passes from the left hand to the right hand? A) When making a fairy wand. It must be soaked overnight in warm water before any decorations can be attached to it. B) When making a fairy wand. It is first covered with hot-glue squiggles and vines so that, after painting, these raised lines give it an embossed appearance. C) When making a cool bookmark. It is meant to be laid in overlapping wide strips that cover the entire bookmark so the finished piece becomes resistant to both water and oil. D) When making a gauntlet from soda bottles. It is folded into segmented pockets that are stitched onto the hand portion of the finished improvised armor. E) When making a gauntlet from soda bottles. It is not required at any stage of constructing the soda-bottle gauntlet and is never mentioned in the process, so omitting it has no effect on the final result.; Passage UIDs: ['46f4a324b8a24b81a1199cd0baec1089', '46f4a324b8a24b81a1199cd0baec1089', 'd3fa966c733841bca71917b525c0eca6', 'd3fa966c733841bca71917b525c0eca6', 'f51698fa3a3c44b0aebc3e762919ddc', 'f51698fa3a3c44b0aebc3e762919ddc']; Passage Titles: ['how to make a fairy wand 3', 'how to make a fairy wand 3', 'how to make a cool bookmark 1', 'how to make a cool bookmark 1', 'how to make a gauntlet from soda bottles', 'how to make a gauntlet from soda bottles']

1188

1189 Query ID: query_0121; Query Text: In a video that arranging containers during dimly lit home repair. Before the camera wearer holds the keg with his left hand, what is the object that the camera wearer drops into the basket? A) screwdriver B) paint brush C) rag D) hammer E) measuring tape F) bottle; Passage UIDs: None; Passage Titles: None
Query ID: query_0121_2hop; Query Text: In a video that arranging containers during dimly lit home repair. Before the camera wearer holds the keg with his left hand, which statement is true about the object that the camera wearer drops into the basket? A) When building your own computer case. It is a practical alternative to masking tape when shielding surfaces during spray-painting the custom case. B) When building your own computer case. It is explicitly included in the passage's checklist of essential tools, appearing between masking tape and a rotary tool or grinder. C) When making a guinea pig cage. The instructions caution against using it on the wire mesh walls since striking the chicken wire could cause it to snap. D) When making a guinea pig cage. It is included in the recommended collection of basic tools, alongside an electric drill and screwdriver, for assembling a custom guinea pig hutch. E) When cleaning your room as teens. It is recommended in the room-cleaning advice as a makeshift flower vase to brighten up shelves and cabinets. F) When cleaning your room as teens. The guide explicitly tells you to put it in the trash bin whenever you come across it while tidying up.; Passage UIDs: ['1bb6656f1748d40c6ac4faa9191472ddd', '23f36f2296549189bdae118cdf84503', '23f36f2296549189bdae118cdf84503', '6328e5aa5c148bdcb6ca48bd52dc5e98f', '6328e5aa5c148bdcb6ca48bd52dc5e98f', '6328e5aa5c148bdcb6ca48bd52dc5e98f']; Passage Titles: ['how to build your own computer case', 'how to build your own computer case', 'how to make a guinea pig cage 1', 'how to make a guinea pig cage 1', 'how to clean your room teens', 'how to clean your room teens']

1190 [dense query text continues]

1191

1192 Query ID: query_0122; Query Text: In a video that fixing a car undercarriage, tightening bolts with a wrench. Before the camera wearer moves the pipe wrench on the floor, what is the object that the camera wearer removes? A) screwdriver B) tire iron C) oil filter D) car jack E) lug nut F) pipe wrench; Passage UIDs: None; Passage Titles: None
Query ID: query_0122_2hop; Query Text: In a video that fixing a car undercarriage, tightening bolts with a wrench. Before the camera wearer moves the pipe wrench on the floor, which statement is true about the object that the camera wearer removes? [options A–F continue]

1193

1194 Query ID: query_0123; Query Text: In a video that fixing a car undercarriage, tightening bolts with a wrench. After the camera wearer removes the pipe wrench, what is the object that the camera wearer moves from the floor? A) screwdriver B) hammer C) flashlight D) oil filter E) jack stand F) pipe wrench; Passage UIDs: None; Passage Titles: None

1195 Query ID: query_0123_2hop; Query Text: In a video that fixing a car undercarriage, tightening bolts with a wrench. After the camera wearer removes the pipe wrench, which statement is true about the object that the camera wearer moves from the floor? [options continue]

1196

1197 Query ID: query_0124; Query Text: In a video that fixing a car undercarriage, tightening bolts with a wrench. Before the camera wearer moves beside the vehicle, what is the object that the camera wearer moves from the floor? A) screwdriver B) flashlight C) oil filter D) tire jack E) pliers F) pipe wrench; Passage UIDs: None; Passage Titles: None

1198 Query ID: query_0124_2hop; Query Text: In a video that fixing a car undercarriage, tightening bolts with a wrench. Before the camera wearer moves beside the vehicle, which statement is true about the object that the camera wearer moves from the floor? [options continue]

1199 Query ID: query_0125; Query Text: In a video that shows manipulating metal strips on a cluttered workbench. Before the camera wearer drops the second metal in his hand, what is the object that the camera wearer drops from his hands onto the table? A) screwdriver B) wooden block C) plastic ruler D) roll of tape E) sheet of sandpaper F) metal; Passage UIDs: None; Passage Titles: None

1200 Query ID: query_0125_2hop; Query Text: In a video that shows manipulating metal strips on a cluttered workbench. Before the camera wearer drops the second metal in his hand, which statement is true about the object that the camera wearer drops from his hands onto the table? [options continue]

1201

1202 Query ID: query_0126; Query Text: In a video that shooting basketballs into a hoop on a sunny day. Before the camera wearer throws the ball in the basket ball net, what is the object that the man throws into the basket ball hoop? A) Frisbee B) water bottle C) hat D) shoe E) apple F) ball; Passage UIDs: None; Passage Titles: None
Query ID: query_0126_2hop; Query Text: In a video that shooting basketballs into a hoop on a sunny day. [options continue]

1203

1204 Query ID: query_0127; Query Text: In a video that shooting basketballs into a hoop on a sunny day. Before the camera wearer takes the ball, what is the object that the man throws into the basket ball hoop? A) Frisbee B) Water bottle C) Sunglasses D) Backpack E) Towel F) Ball; Passage UIDs: None; Passage Titles: None

1205 Query ID: query_0127_2hop; Query Text: In a video that shooting basketballs into a hoop on a sunny day. [options continue]

1206

1207 Query ID: query_0128; Query Text: In a video that shooting basketballs into a hoop on a sunny day. Before the camera wearer takes the ball, what is the object that the man throws into the basket ball hoop? A) water bottle B) frisbee C) hat D) skateboard E) umbrella F) ball; Passage UIDs: None; Passage Titles: None

1208 Query ID: query_0128_2hop; Query Text: [options continue]

1209

1210 Query ID: query_0129; Query Text: In a video that shooting basketballs into a hoop on a sunny day. Before the camera wearer takes the ball, what is the object that the man throws into the basket ball hoop? A) Water bottle B) Frisbee C) Backpack D) Towel E) Sunglasses F) Ball; Passage UIDs: None; Passage Titles: None

1211 Query ID: query_0129_2hop; Query Text: [options continue]

1212 Query ID: query_0130; Query Text: In a video that unlocking and opening a door with a key. Before the camera wearer inserts a key in door, what is the object that the camera wearer holds? A) Coin B) Smartphone C) Wallet D) Screwdriver E) Credit card F) Key; Passage UIDs: None; Passage Titles: None

1213 Query ID: query_0130_2hop; Query Text: In a video that unlocking and opening a door with a key. [options continue]

1214

1215 Query ID: query_0131; Query Text: In a video that organizing caps on a wall rack and vacuuming shelf. Before the camera wearer drops the piece of cloth on the bed in his right hand, what is the object that the camera wearer picks from the bag on the bed into his right hand? A) baseball cap B) sneaker C) water bottle D) smartphone E) book F) cloth; Passage UIDs: None; Passage Titles: None

1216 Query ID: query_0131_2hop; Query Text: [options continue]

1217

1218 Query ID: query_0132; Query Text: In a video that organizing caps on a wall rack and vacuuming shelf. Before a man picks the drill guideline material with her right hand, what is the object that the woman brushes? A) dust on the wall rack B) the wooden shelf surface C) a baseball cap hanging nearby D) her own shirt sleeve E) the drill guideline material itself F) hair; Passage UIDs: None; Passage Titles: None
Query ID: query_0132_2hop; Query Text: In a video that repairing a table underside with a drill. Before a man sits under the table, what is the object that the camera wearer releases? A) chair B) drill C) toolbox D) door E) wooden board F) table; Passage UIDs: None; Passage Titles: None

1219 Query ID: query_0133; Query Text: In a video that repairing a table underside with a drill. Before a man sits under the table, which statement is true about the object that the camera wearer releases? [options continue]

1220

1221 Query ID: query_0134; Query Text: In a video that driving a car on a quiet tree-lined road. After the camera wearer drives past a tricycle, what is the object that camera wearer drives? A) bicycle B) pickup truck C) electric scooter D) sedan car E) horse cart F) motorcycle; Passage UIDs: None; Passage Titles: None

1222 Query ID: query_0134_2hop; Query Text: [options continue]

1223

1224 Query ID: query_0135; Query Text: In a video that carrying a ladder from garage to driveway. Before the camera wearer walks in the garage, what is the object that camera wearer takes? A) shovel B) garden hose C) bicycle D) toolbox E) rake F) ladder; Passage UIDs: None; Passage Titles: None

1225 Query ID: query_0135_2hop; Query Text: [options continue]

1226 Query ID: query_0136; Query Text: In a video that walking indoors, climbing stairs, and entering an apartment kitchen. After the camera wearer walks towards the a man, what is the object that man open? A) window B) refrigerator C) cupboard D) microwave E) desk drawer F) door; Passage UIDs: None; Passage Titles: None

1227 Query ID: query_0136_2hop; Query Text: [options continue]

1228

1229 Query ID: query_0137; Query Text: In a video that repairing a mountain bike in a workshop. Before the camera wearer straightens the bicycle tire, what is the object that camera wearer pick? A) metal wrench B) tire air pump C) bicycle helmet D) chain lubricant bottle E) water bottle F) foam wrap; Passage UIDs: None; Passage Titles: None

1230 Query ID: query_0137_2hop; Query Text: In a video that repairing a mountain bike in a workshop. Before the camera wearer straightens the bicycle tire, which statement is true about the object that camera wearer pick? [options continue]

1231

1232 Query ID: query_0138; Query Text: In a video that drilling a wooden shoe rack with a cordless drill. Before the camera wearer removes the drill driver from the shelf with his right hand, what is the object that the camera wearer uses from shelf? A) Hammer B) Tape measure C) Paintbrush D) Screwdriver E) Wood glue bottle F) Drill driver; Passage UIDs: None; Passage Titles: None

1233 Query ID: query_0138_2hop; Query Text: [options continue]

1234

1235 Query ID: query_0139; Query Text: In a video that drilling a wooden shoe rack with a cordless drill. After the camera wearer uses the drill driver on the shelf with his right hand, what is the object that the camera wearer removes from the shelf? A) hammer B) measuring tape C) paintbrush D) wooden plank E) level tool F) drill driver; Passage UIDs: None; Passage Titles: None

1236 Query ID: query_0140; Query Text: In a video that drilling a wooden shoe rack with a cordless drill. After the camera wearer removes the drill driver from the shelf with his right hand, what is the object that camera wearer uses from shelf? A) hammer B) tape measure C) paintbrush D) wrench E) spirit level F) drill driver; Passage UIDs: None; Passage Titles: None

1237

1238 Query ID: query_0141; Query Text: In a video that writing measurements on paper with a pencil in workshop. Before the camera wearer writes on the paper on the locker with the pencil in his right hand, what is the object that the camera wearer places into the locker? A) screwdriver B) measuring tape C) glove D) metal ruler E) wooden block F) paper; Passage UIDs: None; Passage Titles: None

1239 Query ID: query_0141_2hop; Query Text: In a video that writing measurements on paper with a pencil in workshop. [options continue]

1240

1241 Query ID: query_0142; Query Text: In a video that writing measurements on paper with a pencil in workshop. After the camera wearer places the paper on the locker with his left hand, what is the object that camera wearer writes from the locker? A) tape measure B) wood plank C) clipboard D) metal ruler F) paper; Passage UIDs: None; Passage Titles: None
Query ID: query_0142_2hop; Query Text: [options continue]

1242

1243 Passage Titles: ['how to build a garage work bench 2', 'how to build a garage work bench 2', 'how to make a cozy cabin facial tissue box cover', 'how to make a cozy cabin facial tissue box cover', 'how to make a chess board 3', 'how to make a chess board 3']
Query ID: query_00143; Query Text: In a video that sorting books and cassette cases on floor. After the camera wearer Adjusts books on the floor, what is the object that the camera wearer picks up from the floor? A) cassette case B) DVD case C) smartphone D) remote control E) pencil F) book; Passage UIDs: None; Passage Titles: None

1244 Query ID: query_00143_2hop; Query Text: In a video that sorting books and cassette cases on floor. After the camera wearer Adjusts books on the floor, which statement is true about the object that the camera wearer picks up from the floor? A) When building a squirrel house. It is proposed as an unconventional bedding material for squirrels once it has been shredded into soft fragments. B) When building a squirrel house. It is listed as one of the basic items, alongside measuring tape and paper, that builders should keep on hand during the project. C) When making a room for your american girl doll. It is recommended to finish it with a layer of clear nail polish to create a durable, glossy surface. D) When making a room for your american girl doll. This can be fashioned by cutting a piece of cardboard to shape and gluing a magazine photograph of one onto the front. E) When caring for a chicken in an apartment. It asserts that roost bars must be installed at least four feet above the coop floor to promote healthy jumping exercises. F) When caring for a chicken in an apartment. It directs the caretaker to mix oyster shell and grit into the feed so the chickens receive essential minerals and can properly grind their food.; Passage UIDs: ['fb71c95bb1bc4a348b640967caf25b7e', 'fb71c95bb1bc4a348b640967caf25b7e', '1835e4dad3d541a591cc2a3058b42328', '1835e4dad3d541a591cc2a3058b42328', 'how to build a squirrel house 1', 'how to build a squirrel house 1', 'how to make a room for your american girl doll 3', 'how to make a room for your american girl doll 3', 'how to care for a chicken in an apartment', 'how to care for a chicken in an apartment']

1245 Query ID: query_00144; Query Text: In a video that sorting books and cassette cases on floor. Before the camera wearer Puts down books on the floor, what is the object that the camera wearer picks up from the floor? A) cassette case B) DVD disc C) remote control D) pair of headphones E) water bottle F) book; Passage UIDs: None; Passage Titles: None

1246 Query ID: query_00144_2hop; Query Text: In a video that sorting books and cassette cases on floor. Before the camera wearer Puts down books on the floor, which statement is true about the object that the camera wearer picks up from the floor? A) When disinfecting a hamsters cage. It should be sterilized every week with the same bleach solution applied to a hamster's toys before being refilled. B) When disinfecting a hamsters cage. It is washed only with soapy water, not with any disinfectant, before being dried and returned to the hamster's cage. C) When making a room for your american girl doll. It is suggested to wrap a slim block of Styrofoam in aluminum foil so it looks like a sleek miniature device. D) When making a room for your american girl doll. This can be fashioned by cutting a piece of cardboard to shape and gluing a magazine photograph of one onto the front. E) When caring for a chicken in an apartment. It directs the caretaker to mix oyster shell and grit into the feed so the chickens receive essential minerals and can properly grind their food.; Passage Titles: ['62b719caf6b24f0b8c7bd6eebef0bce7', '62b719caf6b24f0b8c7bd6eebef0bce7', '1835e4dad3d541a591cc2a3058b42328', '1835e4dad3d541a591cc2a3058b42328', 'how to disinfect a hamsters cage 2', 'how to make a room for your american girl doll 3', 'how to care for a chicken in an apartment', 'how to care for a chicken in an apartment']

1248 Query ID: query_00145; Query Text: In a video that sorting books and cassette cases on floor. After the camera wearer Picks up books from the floor, what is the object that the camera wearer puts onto the floor? A) cassette case B) shoe C) water bottle D) mobile phone E) pillow F) book; Passage UIDs: None; Passage Titles: None

1249 Query ID: query_00145_2hop; Query Text: In a video that sorting books and cassette cases on floor. After the camera wearer Picks up books from the floor, which statement is true about the object that the camera wearer puts onto the floor? A) When making a cat wall bed from a basket. The instructions require that this be strictly circular and stud-supported to match the shape of the mounted basket. B) When making a cat wall bed from a basket. It is suggested as one of the two interchangeable soft liners—along with a blanket—that can be placed inside the wall-mounted basket to make the cat's new bed more inviting. C) When hiring a netball umpire.

1251 Query ID: query_00146; Query Text: In a video that sorting books and cassette cases on floor. Before the camera wearer Walks around a room, what is the object that the camera wearer puts onto the floor? A) cassette case B) laptop computer C) throw pillow D) coffee mug E) remote control F) book; Passage UIDs: None; Passage Titles: None

1254 Query ID: query_00147; Query Text: In a video that sorting books and cassette cases on floor. Before the camera wearer Puts down books on the floor, what is the object that the camera wearer picks up from the shelf? A) cassette case B) DVD case C) ceramic mug D) remote control E) stuffed toy F) book; Passage UIDs: None; Passage Titles: None

1257 Query ID: query_00148; Query Text: In a video that sorting books and cassette cases on floor. Before the camera wearer Adjusts books on the floor, what is the object that the camera wearer puts onto the floor? A) cassette case B) CD F) jewel case C) mobile phone D) pair of scissors E) backpack F) book; Passage UIDs: None; Passage Titles: None

1260 Query ID: query_00149; Query Text: In a video that sorting books and cassette cases on floor. After the camera wearer Puts down books on the floor, what is the object that the camera wearer adjusts from the floor? A) cassette case B) smartphone C) pair of glasses D) coffee mug E) laptop charger F) book; Passage UIDs: None; Passage Titles: None

1263 Query ID: query_00150; Query Text: In a video that dribbling and shooting a basketball on an outdoor court. Before a face catches a ball, what is the object that camera wearer shoots? A) basketball hoop B) scoreboard C) water bottle D) orange traffic cone E) trash can F) ball; Passage UIDs: None; Passage Titles: None

1265 Query ID: query_00151; Query Text: In a video that dribbling and shooting a basketball on an outdoor court. Before a man catches a ball, what is the object that the man runs? A) bicycle B) dog C) water bottle D) cone E) skateboard F) ball; Passage UIDs: None; Passage Titles: None

1268 Query ID: query_00152; Query Text: In a video that repairing furniture underneath a desk with hand tools. Before the camera wearer places her left hand on the wooden slab, what is the object that A man place? A) screw B) bolt C) washer D) wooden dowel E) measuring tape F) nail; Passage UIDs: None; Passage Titles: None

1271 Query ID: query_00153; Query Text: In a video that shows repairing furniture underneath a desk with hand tools. After a man holds the edge of the wooden slab with his left hand, what is the object that man drives onto the wooden slab? A) screw B) metal bracket C) wooden clamp D) adhesive tape E) rubber washer F) nail; Passage UIDs: None; Passage Titles: None

1273 Query ID: query_00154; Query Text: In a video that measuring a table using a tape measure and level. After the a man picks a tape measure, what is the object that man throws onto the table? A) notebook B) smartphone C) screwdriver F) tape measure; Passage UIDs: None; Passage Titles: None

1276 Query ID: query_00155; Query Text: In a video that tidying video game cases on a wooden shelf. Before the camera wearer holds an envelope, what statement is true about the object that the camera wearer puts into the cabinet? A) When ring list your bedroom.

1279 Query ID: query_00156; Query Text: In a video that organizing tools and materials on a workshop table. Before the camera wearer drops the ceramic ware on the table with his right hand, what is the object that the camera wearer puts onto the table? A) When making clocks. It is necessary for fastening the clock hands onto the spindle during assembly of the mechanism. B) When making clocks.

1281 Query ID: query_00157; Query Text: In a video that connecting a wire to earphones indoors. Before the camera wearer drops the ceramic ware on the table with his right hand, which statement is true about the object that the camera wearer picks from the table in the bedroom? A) smartphone B) notebook C) pair of sunglasses D) hairbrush E) glass bottle F) ceramic ware; Passage UIDs: None; Passage Titles: None

1283 Query ID: query_00158; Query Text: In a video that connecting a wire to earphones indoors. Before the camera wearer walks to the living room from the bedroom, what is the object that the camera wearer drops onto the table? A) mobile phone B) key necklace C) paper notebook D) television remote E) plastic water bottle F) ceramic ware; Passage UIDs: None; Passage Titles: None

1286 Query ID: query_00159; Query Text: In a video that fixing an electric scooter wheel in a workshop. After the camera wearer Picks up a stapler from a wall, what is the object that the camera wearer spins? A) screwdriver B) bicycle wheel C) power drill E) tape measure F) metal rod; Passage UIDs: None; Passage Titles: None

1289 Query ID: query_00160; Query Text: In a video that shooting basketball into hoop on outdoor park court. Before a person holds a basketball, what is the object that the camera wearer throws into the basketball net? A) Frisbee B) Water bottle C) Tennis ball D) Football E) Basketball; Passage UIDs: None; Passage Titles: None

1291 Query ID: query_00161; Query Text: In a video that shooting basketball into hoop on outdoor park court. Before a person bends, what is the object that person moves? A) shoulder B) foot C) knee D) waist E) basketball F) hand; Passage UIDs: None; Passage Titles: None

1294 Query ID: query_00162; Query Text: In a video that working on a cluttered workshop repair task. Before the camera wearer walks around the garage, what is the object that camera wearer zips? A) tool bag B) toolbox C) duffel bag D) backpack E) boots F) jacket; Passage UIDs: None; Passage Titles: None

1295 Query ID: query_00162_2hop; Query Text: In a video that working with homesickness while traveling. It is described as essential footwear for the morning strolls that help maintain your daily routine in an unfamiliar city. B) When coping with homesickness while traveling. It is presented as a handy purchase you may happily splurge on while traveling to shift attention from missing home to enjoying the destination. C) When stopping being a nerd over summer. It is labeled "dorky" and the suggestion is to swap it out for a pretty tote to carry school supplies. D) When living within your means. The guide specifically urges readers to try it used from thrift stores, implying a full retail price for a new one. E) When living within your means. The advice recommends delaying the decision to purchase it for 48 hours to reduce impulsive spending.; Passage UIDs: ['34a3c3d57c96412cb1ff75a277b798e4', '34a3c3d57c96412cb1ff75a277b798e4', '15c05e6a920249938c72258432de9ec6', '15c05e6a920249938c72258432de9ec6']; Passage Titles: ['how to cope with homesickness while traveling', 'how to cope with homesickness while traveling', 'how to stop being a nerd over summer', 'how to stop being a nerd over summer', 'how to live within your means', 'how to live within your means']

Query ID: query_00163; Query Text: In a video that working on a cluttered workshop repair task. After the camera wearer walks around the garage, what is the object that the camera wearer adjusts? A) wrench B) hose C) screwdriver D) belt E) spark plug F) cable; Passage UIDs: None; Passage Titles: None

Query ID: query_00163_2hop; Query Text: In a video that working on a cluttered workshop repair task. After the camera wearer walks around the garage, which statement is true about the object that the camera wearer adjusts? A) When making cat toys out of common household items. It is listed as a suitable substitute for ribbon so owners can manipulate the toy in ways that mimic real animal movements. B) When making cat toys out of common household items. It is never actually mentioned among the suggested materials—like string or ribbon—when describing how to assemble the homemade cat toy. C) When changing a kids room to a tweens room. It is said to be discouraged for tween redecorating projects because its sharp point poses an unacceptable safety risk. D) When changing a kids room to a tweens room. It is paired with a power drill, is mentioned as sufficient for installing most simple additions during the room makeover. E) When finding the perfect off campus student apartment. It is singled out as an electronic accessory that requires tenants to verify the presence of sufficient wall outlets for computers, video games, and chargers. F) When finding the perfect off campus student apartment. It is never brought up at all, since apartment-hunting involves checking outlets, ventilation, water pressure, window locks, noise levels, or nearby laundromat access, which are explicitly discussed.; Passage UIDs: ['53603216d4af47e5b3660c9c4e7ac362', '53603216d4af47e5b3660c9c4e7ac362', 'bc1afd61a3b54911a46ec1069990789a', '1e1bafd929d241ffb0fb89f281caa4cb', '1e1bafd929d241ffb0fb89f281caa4cb']; Passage Titles: ['how to make cat toys out of common household items 8', 'how to make cat toys out of common household items 8', 'how to change a kids room to a tweens room', 'how to change a kids room to a tweens room', 'how to find the perfect off campus student apartment', 'how to find the perfect off campus student apartment']

Query ID: query_00164; Query Text: In a video that working on a cluttered workshop repair task. Before the camera wearer walks around the garage, what is the object that the camera wearer adjusts? A) wrench B) hose C) screwdriver D) belt E) glove F) cable; Passage UIDs: None; Passage Titles: None

Query ID: query_00164_2hop; Query Text: In a video that working on a cluttered workshop repair task. Before the camera wearer walks around the garage, which statement is true about the object that the camera wearer adjusts? A) When covering wood paneling. It should be taken off during the sanding stage because its rubber surface can grind dust back into the wood grooves and spoil the finish. B) When covering wood paneling. It is recommended to wear it along with a face mask when tackling extensive paneling projects in spaces with limited airflow. C) When painting concrete siding. This is specified as the preferred apparatus for forcing polyurethane-fortified paint evenly into every groove of fiber-cement siding. D) When painting concrete siding. It is absent from the listed tools and materials deemed necessary for cleaning, priming, or painting fiber-cement siding according to the detailed procedure. E) When finding the perfect off campus student apartment. It is singled out as an electronic accessory that requires tenants to verify the presence of sufficient wall outlets for computers, video games, and chargers. F) When finding the perfect off campus student apartment. It is never brought up at all, since apartment-hunting involves checking outlets, ventilation, water pressure, window locks, noise levels, or nearby laundromat access, which are explicitly discussed.; Passage UIDs: ['cfe2e3e7bf454a20895la353f8857bda', 'cfe2e3e7bf454a20895la353f8857bda', 'a9421 6aae5a145868fe39612 4d27af03', 'a9421 6aae5a145868fe39612 4d27af03', '1e1bafd929d241ffb0fb89f281caa4cb', '1e1bafd929d241ffb0fb89f281caa4cb']; Passage Titles: ['how to cover wood paneling', 'how to cover wood paneling', 'how to paint concrete siding', 'how to paint concrete siding', 'how to find the perfect off campus student apartment', 'how to find the perfect off campus student apartment']

Query ID: query_00165; Query Text: In a video that playing baseball by pitching, batting, and catching. After the camera wearer walks towards the mark on the ground, what is the object that the camera wearer raises? A) baseball glove B) baseball C) helmet D) water bottle E) scorecard F) bat; Passage UIDs: None; Passage Titles: None

Query ID: query_00165_2hop; Query Text: In a video that playing baseball by pitching, batting, and catching. After the camera wearer walks towards the mark on the ground, which statement is true about the object that camera wearer raises? A) When playing golf in gta v. It shows your total driving distance and average swing speed for each of the nine holes you play in GTA V La Santos golf activity. B) When playing golf in gta v. It appears only once you have completed the ninth hole at the Los Santos Golf Club, presenting a summary of how well you performed in the GTA V golf mini-game. C) When playing a catcher in baseball. To help sell borderline strikes, a catcher is coached to nudge it forward over the brow as the glove frames the pitch. D) When playing a catcher in baseball. When tracking pop-ups or foul tips, a catcher is expected to remove it immediately to gain unobstructed sight and agility. E) When playing wall bat. It is kept five yards from the wall so the pitcher can grab it immediately before every pitch. F) When playing wall bat. It is swung by the batter to hit the tennis ball after it bounces off the wall, sending the ball high up the wall.; Passage UIDs: ['61647da030ff4dc2b78e0c4c7566f16d', '61647da030ff4dc2b78e0c4c7566f16d', '68dcc699a0454f06a3bd33783d42d1c6', '68dcc699a0454f06a3bd33783d42d1c6', 'afa513dc1a5b4041bef49757c7c90c6c', 'afa513dc1a5b4041bef49757c7c90c6c']; Passage Titles: ['how to play golf in gta v', 'how to play golf in gta v', 'how to be a catcher in baseball', 'how to be a catcher in baseball', 'how to play wall bat', 'how to play wall bat']

Query ID: query_00166; Query Text: In a video that playing baseball by pitching, batting, and catching. After the camera wearer throws the ball up, what is the object that the camera wearer plays with the camera wearer? A) baseball glove B) baseball C) frisbee D) plastic bottle E) cap F) ball; Passage UIDs: None; Passage Titles: None

Query ID: query_00166_2hop; Query Text: In a video that playing baseball by pitching, batting, and catching. After the camera wearer throws the ball up, which statement is true about the object that the camera wearer plays with the camera wearer? A) When playing plastic bottle toss. A successful throw only counts when it rebounds off the wall once before landing in the bin, compelling players to use bank shots. B) When playing plastic bottle toss. Competitors stand behind a marked tape line and attempt to toss it directly into an otherwise empty trash can to earn points. C) When getting into baseball career. It is specifically required for students pursuing a bachelor of science in sports administration if they intend to work in baseball career. Although the guidance covers throwing, catching, batting, and scouting, it is never once identified as necessary gear for advancing in a baseball career. D) When throwing a baseball. In a recommended grip, the index and middle fingers run lengthwise between two seams while the thumb presses down on the leather above them. E) When throwing a baseball. A proper grip rests it on the fingertips with the index and middle fingers just across a seam and the thumb placed underneath as a third support point, enabling contact with all four seams.; Passage UIDs: ['968ed0a71454a4894366f7b0cf9bdc51', '968ed0a71454a4894366f7b0cf9bdc51', '57669e38a39da489488428094349d23', '57669e38a39da489488428094349d23', 'abf651705c1b442f803ab3ae4eefbad6', 'abf651705c1b442f803ab3ae4eefbad6']; Passage Titles: ['how to play plastic bottle toss', 'how to play plastic bottle toss', 'how to get into baseball career', 'how to get into baseball career', 'how to throw a baseball', 'how to throw a baseball 1']

Query ID: query_00167; Query Text: In a video that placing napkins and utensils on a dining table. Before a person walks around the room, what is the object that the camera wearer operates? A) television remote B) tablet computer C) light switch D) door handle E) drinking glass F) phone; Passage UIDs: None; Passage Titles: None

Query ID: query_00167_2hop; Query Text: In a video that placing napkins and utensils on a dining table. Before a person walks around the room, which statement is true about the object the camera wearer operates? A) When painting a stained door. Lightly snuffing it with a detergent is recommended so the finishing paint bonds well to its metallic skin. If the door cannot be taken off its hinges, painters tape should be run around it and the adjoining frame to protect them while brushing on cabinet lighting. For electrical safety, it must always operate on a completely dedicated circuit that serves no other receptacles or fixtures. D) When installing under cabinet lighting. When this is added to a kitchen outlet, its yellow wire is joined to a black wire that supplies power to the light fixture cable. E) When keeping a parakeet safe out of its cage. Because its electromagnetic emissions may interfere with a bird's navigation abilities, users are told to leave it in a separate room wherever the parakeet is flying. F) When keeping a parakeet safe out of its cage. Caregivers are explicitly advised to avoid using it while supervising the parakeet's daily free-flight period to prevent distracting themselves from the bird.; Passage UIDs: ['a3d5aa404e204db5850ca06d053bc8ab', 'a3d5aa404e204db5850ca06d053bc8ab', '8cf99dd634ab4a90958bc0c9924bc4e7', '8cf99dd634ab4a90958bc0c9924bc4e7', '6ba706fc5fa9465dbeacf100edab4857']; Passage Titles: ['how to paint a stained door', 'how to paint a stained door', 'how to install under cabinet lighting', 'how to install under cabinet lighting', 'how to keep a parakeet safe out of its cage', 'how to keep a parakeet safe out of its cage']

Query ID: query_00168; Query Text: In a video that placing napkins and utensils on a dining table. After a person opens the drawer, what is the object that person pick up? A) spoon B) fork C) napkin D) plate E) salt shaker F) table cloth; Passage UIDs: None; Passage Titles: None

Query ID: query_00168_2hop; Query Text: In a video that placing napkins and utensils on a dining table. After a person opens the drawer, which statement is true about the object person pick up? A) When making your own garden table. It is supposed to be fashioned from a single solid board measuring 186 cm by 92 cm to guarantee a seamless surface. B) When making your own garden table. It is built from separate 92 cm (36.2 in) planks that are screwed down at four points each, leaving a 1 cm gap between adjacent planks for appearance. C) When helping around the house. Readers are told that this should be thrown away immediately after clearing the table so crumbs do not attract bugs. D) When helping around the house. The instructions state that folding it in decorative ways can make a table setting look nicer and more creative. E) When cleaning up after a dinner party. It is highlighted as an item that must be thrown straight into the washing machine before refitting it flight. When homemade snack bars are vacuum-sealed, it can keep them at room temperature for only about five days before their safety and quality. F) When cleaning up after a dinner party. It is not mentioned at all in the long checklist of cleanup chores suggested after the dinner party.; Passage UIDs: ['9ef444840a5140bf9dc373c01cb03c6bf', '9ef444840a5140bf9dc373c01cb03c6bf', 'fee6042709e649c98505986681d0025e', 'fee6042709e649c98505986681d0025e']; Passage Titles: ['how to make your own garden table', 'how to make your own garden table', 'how to help around the house', 'how to clean up after a dinner party', 'how to clean up after a dinner party']

Query ID: query_00169; Query Text: In a video that jotting measurements on paper at a cluttered workbench, what is the object that the camera wearer holds? A) screwdriver B) Tablet computer C) Wooden plank D) Tape measure E) Coffee mug F) Paper; Passage UIDs: None; Passage Titles: None

Query ID: query_00169_2hop; Query Text: In a video that jotting measurements on paper at a cluttered workbench. Before the camera wearer walks towards a table in the workshop, which statement is true about the object that the camera wearer holds? A) When making a doll house. It verifies that the dowel holes run at consistent two-inch intervals along the rising sides of every board prior to gluing. B) When making a doll house. It is laid against the narrow edge of each 24-inch board to mark points precisely three and six inches from both ends so the drilled holes will align. C) When surviving in survival mode in minecraft. It is impossible to turn it into a cheat or any other form of in-game storage container. D) When surviving in survival mode in minecraft. Crafting a bed in the game requires six of it placed under three blocks of wool on a crafting grid. E) When building a cookie tree. Builders are instructed to cut this into three lengths—30 cm, 40 cm, and 50 cm—to create the graduated branches of the cookie tree. F) When building a cookie tree. The directions mention that crumpling it is an optional way to fill the pot and stabilize the completed cookie tree.; Passage UIDs: ['db9e8f22abb448b8ae82ef4632768f47', 'db9e8f22abb448b8ae82ef4632768f47', '651815a87fb941cbaad07ffa2951d28d', '651815a87fb941cbaad07ffa2951d28d', 'c02f89a6d4e48298c24ae5f4bed52d8'] Passage Titles: ['how to make a doll house 3', 'how to make a doll house 3', 'how to survive in survival mode in minecraft', 'how to survive in survival mode in minecraft', 'how to build a cookie tree', 'how to build a cookie tree']

Query ID: query_00170; Query Text: In a video that taking a knife and preparing food in kitchen. Before the camera wearer walks around in a house, what is the object that the camera wearer throws? A) refrigerator B) drawer C) microwave D) oven door E) pantry door F) cabinet; Passage UIDs: None; Passage Titles: None

Query ID: query_00170_2hop; Query Text: In a video that taking a knife and preparing food in kitchen. Before the camera wearer walks around in a house, what is the object that the camera wearer closes? A) When losing weight in 3 weeks. It should guard a pantry that has been cleared of sugary snacks and junk food to minimize temptation during a weight-loss program. C) When using baking soda for stubborn oven stains. Some cleaning guides recommend lifting it off its hinges entirely before treatment so paste cannot drip into the oven cavity. D) When using baking soda to clean it with a damp baking-soda paste stay on the stained inner glass for an extended overnight soak. E) When making a healthy snack bar. It provides the safest and most effective environment for very long-term storage of homemade snack bars, surpassing both refrigeration and freezing in inhibiting bacterial growth. F) When making a healthy snack bar. When homemade snack bars are vacuum-sealed, it can keep them at room temperature for only about five days before their safety and quality begin to deteriorate.; Passage UIDs: ['f98eea7873d84c8bc429f6a523a54c', 'f98eea7873d84c8bc429f6a523a54c', '5ffcfc4e91bf4a5897e97 3ec502a30e7', '5ffcfc4e91bf4a5897e97 3ec502a30e7', '34dca52fc3284 3c09ac417a578aa3a8d']; Passage Titles: ['how to lose weight in 3 weeks', 'how to lose weight in 3 weeks', 'how to use baking soda for stubborn oven stains 3', 'how to use baking soda for stubborn oven stains 3', 'how to make a healthy snack bar 3']

Query ID: query_00171; Query Text: In a video that tightening a bolt beneath a desk using a wrench. Before the camera wearer adjusts the bolt, what is the object that adjusts? A) knob B) lever C) quill D) drawer E) lamp F) bolt; Passage UIDs: None; Passage Titles: None

Query ID: query_00171_2hop; Query Text: In a video that tightening a bolt beneath a desk using a wrench. Before the camera wearer adjusts the bolt, which statement is true about the object that adjusts? A) When getting a room like zoey 10is. It is recommended to pick a vivid orange one and hang it from the ceiling to replicate the Zoey 101 dorm vibe. B) When getting a room like zoey 10is. It is flattened and used as a bedside table next to a pink clock in the suggested room layout. C) When creating a home inventory for insurance purposes. It is judged unnecessary to record in a household inventory because its miscellaneous contents are presumed inexpensive. D) When creating a home inventory for insurance purposes. It is singled out as the compartment to examine immediately after catologing the objects on a desk when assembling a detailed home office inventory. E) When replacing missing rivets in hockey skates. To reduce unwanted height inside the boot, technicians pair it with the smallest available washer, which also distributes pressure more evenly on the skate sole. F) When replacing missing rivets in hockey skates. When a broken rivet stem remains in the hole, it can be shoved upward with it from the bottom of the skate to knock the leftover piece into the boot.; Passage UIDs: ['61155c75f891430bb23a0f20fea525a54c', '7df547b7493043afb0014f3289la18d4f', '2fd547b7493043afb0014f3289la18d4f', 'e1eb02effce64b96b3cb6bf5fd74f033']; Passage Titles: ['how to get a room like zoey 10is', 'how to get a room like zoey 10is', 'how to create a home inventory for insurance purposes', 'how to create a home inventory for insurance purposes', 'how to replace missing rivets in hockey skates']

Query ID: query_00172; Query Text: In a video that carrying a wooden chair through a house. After the camera wearer Walks to a living room, what is the object that the man carries from the floor? A) lamp B) coffee table C) floor rug D) guitar E) suitcase F) chair; Passage UIDs: None; Passage Titles: None

Query ID: query_00172_2hop; Query Text: In a video that carrying a wooden chair through a house. After the camera wearer Walks to a living room, which statement is true about the object that the man carries from the floor? A) When bing a drifter. Readers are told to fold their suitcase into it before locking it away so that interview clothes stay pristine. B) When bing a drifter. It is mentioned only to note that, compared with a backpack, it is a less convenient item for a drifter to carry. C) When earning pocket money. It is presented as an effective way to improve computer programming skills by learning its chord patterns. D) When earning pocket money. It is suggested as an instrument you can perform at weddings for a fee to earn extra money. E) When cleaning your house for a sleepover. Sprinkling baking soda or cornstarch on its seat and then vacuuming is recommended to remove lingering odors from the living room. F) When cleaning the fully closed and pushed in to avoid blocking entrances and to keep the living room looking tidy.; Passage UIDs: ['90529925151d4803 9cca6013b4c4680c', '90529925151d4803 9cca6013b4c4680c', '43046356ae5f4e7daecad9826100b0d8', '43046356ae5f4e7daecad9826100b0d8', '181b6436a2824073b5908192 8be63777']; Passage Titles: ['how to be a drifter', 'how to be a drifter', 'how to earn pocket money 3', 'how to earn pocket money 3', 'how to clean your house for a sleepover', 'how to clean your house for a sleepover']

Query ID: query_00173; Query Text: In a video that carrying a wooden chair through a house. Before a man carries the chair from the floor with both hands, what is the object that man drops onto the floor? A) lamp B) guitar C) vacuum cleaner D) painting E) suitcase F) chair; Passage UIDs: None; Passage Titles: None

Query ID: query_00173_2hop; Query Text: In a video that carrying a wooden chair through a house. Before a man carries the chair from the floor with both hands, which statement is true about the object that man drops onto the floor? A) When bing a drifter. It is presented as the ideal piece of luggage when living on the road, praised for its light weight and portability. B) When bing a drifter. It is mentioned only to note that, compared with a backpack, it is a less convenient item for a drifter to carry. C) When living alone happily. Because of its significant weight, the text recommends dismantling it or hiring assistance before moving it to prevent personal injury. D) When living alone happily. The text presents it as a quirky item you can hang on your wall to personalize your space and lift your mood when you live alone. E) When cleaning your house for a sleepover. Placing it beside the hallway entrance is suggested to give arriving guests an immediate place to sit while they take off their shoes. F) When cleaning your house for a sleepover. It is a reclining piece of furniture, it should be fully closed and pushed in to avoid blocking entrances and to keep the living room looking tidy.; Passage UIDs: ['90529925151d4803 9cca6013b4c4680c', 'ad1be36bf8c4b85a862le04a06fab97', '181b6436a2824073b5908192 8be63777', '181b6436a2824073b5908192 8be63777']; Passage Titles: ['how to be a drifter', 'how to be a drifter', 'how to live alone happily', 'how to live alone happily', 'how to clean your house for a sleepover', 'how to clean your house for a sleepover']

Query ID: query_00174; Query Text: In a video that carrying a wooden chair through a house. After the a man drops the chair on the floor with both hands, what is the object that man carries from the floor? A) table B) suitcase C) lamp D) cardboard box E) guitar F) chair; Passage UIDs: None; Passage Titles: None

Query ID: query_00174_2hop; Query Text: In a video that carrying a wooden chair through a house. After the a man drops the chair on the floor with both hands, which statement is true about the object that man carries from the floor? A) When earning pocket money. It is presented as an effective way to improve computer programming skills by learning its chord patterns. B) When earning pocket money. It is suggested as an instrument you can perform at weddings for a fee to earn extra money. C) When making an hourglass clock out of light bulbs. It is flattened and used as a makeshift sanding board to smooth the sharp rim left after its black glass cap has been shattered. D) When making an hourglass clock out of light bulbs. It is placed beneath the light bulb during the metal-tip removal so that any falling shards of broken glass are safely caught. E) When cleaning your house for a sleepover. It should be relocated to the kitchen an hour before guests arrive so the floor beneath it can be thoroughly aired and dried. F) When cleaning your house for a sleepover. It is a reclining piece of furniture, it should be fully closed and pushed in to avoid blocking entrances and to keep the living room looking tidy.; Passage UIDs: ['43046356ae5f4e7daecad9826100b0d8', '926a1121e94d4e358d2a66c006221378', '926a1121e94d4e358d2a66c006221378', '181b6436a2824073b5908192 8be63777']; Passage Titles: ['how to earn pocket money 3', 'how to earn pocket money 3', 'how to make an hourglass clock out of light bulbs', 'how to make an hourglass clock out of light bulbs', 'how to clean your house for a sleepover', 'how to clean your house for a sleepover']

Query ID: query_00175; Query Text: In a video that carrying a wooden chair through a house. Before the a man walks out of the apartment With The camera wearer, what is the object that man carries from the floor? A) table B) lamp C) suitcase D) box E) guitar F) chair; Passage UIDs: None; Passage Titles: None

Query ID: query_00175_2hop; Query Text: In a video that carrying a wooden chair through a house. Before the a man walks out of the apartment with the camera wearer, which statement is true about the object that man carries from the floor? A) When starting a job where you make your own schedule. It is cited in the passage as an essential possession for anyone hoping to qualify for work as a ride-share driver with Uber or Lyft. B) When getting a job where you make your own schedule. It is mentioned as a skill that someone could teach through one-on-one lessons while choosing their own working hours. C) When cleaning and organize your tack room. It should be positioned beneath the sink to keep grooming liquids readily accessible. D) When cleaning and organize your tack room. It needs to be replaced with a new one if mold or fungus is detected on it. E) When cleaning your house for a sleepover. Sprinkling baking soda or cornstarch on its seat and then vacuuming is recommended to remove lingering odors from the living room. F) When cleaning your house for a sleepover. It is a reclining piece of furniture, it should be fully closed and pushed in to avoid blocking entrances and to keep the living room looking tidy.; Passage UIDs: ['ddb7267b0e07 4a469e0912c1c91f61b8', '88990dbb7baa41df87f6d852a3eca7345', '88990dbb7baa41df87f6d852a3eca7345', '181b6436a2824073b5908192 8be63777']; Passage Titles: ['how to get a job where you make your own schedule 2', 'how to get a job where you make your own schedule 2', 'how to clean and organize your tack room', 'how to clean and organize your tack room', 'how to clean your house for a sleepover', 'how to clean your house for a sleepover']

Query ID: query_00176; Query Text: In a video that servicing and cleaning bicycle wheel components on a workbench. Before the camera wearer picks the can of lubricant from the table with his right hand, what is the object that the camera wearer drops onto the table? A) Bicycle chain B) Pedal C) Gear cassette D) Spoke wrench E) Wheel hub; Passage UIDs: None; Passage Titles: None

Query ID: query_00176_2hop; Query Text: In a video that servicing and cleaning bicycle wheel components on a workbench. Before the camera wearer picks the can of lubricant from the table with his right hand, which statement is true about the object that the camera wearer drops onto the table? A) When changing a tire on a 1999-2004 Jeep grand cherokee wj. Before using it to pry the table, must be firmly engaged to keep the vehicle from rolling. B) When changing a tire on a 1999-2004 Jeep grand cherokee wj. Throughout the detailed tire-changing procedure for a 1999-2004 Jeep Cherokee WJ, it is never mentioned or used at any point. C) When riding and mount a unicycle. Because it frequently bumps against the rider's knees, knee pads are deemed essential protective equipment when practicing with it. D) When riding and mount a unicycle. For beginners, it is best if it is made of rubber rather than metal to make mastering the idea difficult. E) When changing a cv axle. It is attached to the strut tower by two 17 mm bolts that must be removed so the axle slides freely through the center hole.; Passage UIDs: ['72faf9c0742a4828b50e5d3eb0993bcc', 'dca275a4191c48502 4fc33b88b123f9', '91e990dfla749f28b62a2bbaa7a4f38'] Passage Titles: ['how to change a tire on a 1999-2004 Jeep grand cherokee wj', 'how to change a tire on a 1999-2004 Jeep grand cherokee wj', 'how to ride and mount a unicycle', 'how to ride and mount a unicycle', 'how to change a cv axle']

Query ID: query_00177; Query Text: In a video that exiting a car then walking toward a sunny playground. After the camera wearer opens the car door, what is the object that the camera wearer touches? A) Smartphone B) Car keys C) Umbrella D) Backpack E) Sandwich F) Water bottle; Passage UIDs: None; Passage Titles: None

Query ID: query_00177_2hop; Query Text: In a video that exiting a car then walking toward a sunny playground. After the camera wearer opens the car door, which statement is true about the object that the camera wearer holds? A) When having fun in the summer holidays. It is suggested as a refreshing ingredient to blend with lemonade when experimenting with mixed summer drinks. B) When having fun in the summer holidays. It is listed among the small foods, alongside potato chips, recommended for staying awake through an all-nighter. C) When making friends in a new place. It is portrayed as a required equipment for anyone enrolling in a karate class through the community center. D) When making friends in a new place. It is suggested as a handy way to carry a ball and a frisbee to the park so you can start a game if necessary. E) When caring for campbells russian hamsters. Experienced keepers note that their use is actively discouraged because these without risk of territorial conflict. F) When caring for campbells russian hamsters. Preparing a habitat for a Campbell's Russian hamster, owners are instructed to position it in a corner of the aquarium on an appropriate stand.; Passage UIDs: ['a7ac0c3d180846ea4lfe41 8ela78b15f', '791411c07f634f07a1c0b0805e86f226', '791411c07f634f07a1c0b0805e86f226']; Passage Titles: ['how to have fun in the summer holidays', 'how to have fun in the summer holidays', 'how to make friends in a new place', 'how to make friends in a new place', 'how to care for campbells russian hamsters', 'how to care for campbells russian hamsters']

Query ID: query_00178; Query Text: In a video that sorting household items into a large storage bin. Before the camera wearer lifts up a calendar in the storage box with her left hand, what is the object that man holds? A) broom B) pillow C) vacuum cleaner D) lamp E) laundry basket F) storage box; Passage UIDs: None; Passage Titles: None

Query ID: query_00178_2hop; Query Text: In a video that sorting household items into a large storage bin. Before the camera wearer lifts up a calendar in the storage box with her left hand, which statement is true about the object that man holds? A) When building your toddlers wardrobe. It should ideally be lined with breathable mesh so that any toddler clothing inside does not develop mildew. B) When building your toddlers wardrobe. It can be set aside exclusively for a toddler's clothes over the course of a week in order to determine how many items the child actually wears. C) When hiding diary keys. The guide explains that concealing keys requires cutting a rectangular cavity into its central support column. D) When hiding diary keys. The instructions advise pushing a tack through the tiny slots of cloth shade and hanging the keys on that tack to hide them. E) When selecting quality kitchen knives. It generally features a built-in knife edge is honed automatically whenever it is taken out or put back. F) When selecting quality kitchen knives. It is sometimes supplied with a single knife as a dedicated container, although many chefs still opt to wrap their blades in an individual.; Passage UIDs: ['eb33a2a2d32a4beaf1d03c032d7a49c77', '4b48bfdddc54974b7261931l2a78908', '4b48bfdddc54974b7261931l2a78908', '32277c94e2e74d83b84f24a4897ed8f'] Passage Titles: ['how to build your toddlers wardrobe 1', 'how to build your toddlers wardrobe 1', 'how to hide diary keys', 'how to hide diary keys', 'how to select quality kitchen knives', 'how to select quality kitchen knives']

Query ID: query_00179; Query Text: In a video that driving a car down a sunlit, tree-lined road. After the camera wearer drives past a grey car, which statement is true about the object that camera wearer drives? A) motorcycle B) skateboard C) electric scooter D) convertible car E) segway F) bicycle; Passage UIDs: None; Passage Titles: None

Query ID: query_00179_2hop; Query Text: In a video that driving a car down a sunlit, tree-lined road. After the camera wearer drives past a grey car, which statement is true about the object that camera wearer drives? A) When driving a golf cart. The majority of insurers exclude it from the recreational-vehicle policies that typically cover golf carts and snowmobiles. B) When driving a golf cart. Many insurance providers include it under broad vehicle policies that also cover golf carts and snowmobiles. C) When riding a scooter. It can cruise at 25 miles per hour without any need for kick-off assistance. D) When riding a scooter. Using its rear foot brake can send power back to the battery because it is equipped with a regenerative braking system. E) When staying fit during your vacation. The passage claims that bringing it into a hotel gym typically requires a special permit because its terrain restrictions. F) When staying fit during your vacation. It is highlighted as an item that most hotels offer for rent alongside kayaks, providing vacationers an enjoyable way to stay active without feeling like they are exercising.; Passage UIDs: ['f3c31c292b504282aa95d4e72ffa5ebf', 'bf311c292b504282aa95d4e72ffa5ebf', '0e97522l1b8b414cba1324b7b0a51672', '0e97522l1b8b414cba1324b7b0a51672', '5ba35183d8fd451f2b46c6ea384567 3e7']; Passage Titles: ['how to drive a golf cart 2', 'how to drive a golf cart 2', 'how to ride a scooter 2', 'how to ride a scooter 2', 'how to stay fit during your vacation 1']

Query ID: query_00180; Query Text: In a video that assembling a bicycle wheel with disc brake components. Before the camera wearer Moves a bicycle wheel on the floor, what is the object that camera wearer puts onto the floor? A) disc brake rotor B) chain lubricant bottle D) hand wrench E) bicycle wheel; Passage UIDs: None; Passage Titles: None

Query ID: query_00180_2hop; Query Text: In a video that assembling a bicycle wheel with disc brake components. Before the camera wearer Moves a bicycle wheel on the floor, which statement is true about the object that camera wearer puts onto the floor? A) When hanging a bike on the wall. For ceiling storage, it is best positioned horizontally with the wheels pointing sideways to reduce the effort of retrieval. B) When hanging a bike on the wall. In perpendicular wall mounting, it must remain vertical so the bike can be rolled upward and hooked by its front wheel. C) When greasing trailer bearings. It is not included among the necessary implements for greasing trailer bearings, a list that instead specifies a screwdriver, rubber mallet, and grease gun. D) When biking through sand. Many mechanics claim that increasing spoke tension on it directly multiplies pedal torque, making it less likely to stall in soft sand. F) When biking through sand. The diameter of its rim sets an upper limit on the width of tire that can be mounted, so riders seeking very broad sand tires may need a different wheel size.; Passage UIDs: ['1aeced6714d84629989e77f13bc1d45a3', '1aeced6714d84629989e77f13bc1d45a3', '60dbb5b19e8a45019 2b31569b0daaf49', '60dbb5b19e8a45019 2b31569b0daaf49', '627664a63dd85768 6e087']; Passage Titles: ['how to hang a bike on the wall 1', 'how to hang a bike on the wall 1', 'how to grease trailer bearings', 'how to grease trailer bearings', 'how to bike through sand', 'how to bike through sand']

Query ID: query_00181; Query Text: In a video that repairing a car chassis in a busy garage. Before the camera wearer walks around, what is the object that camera wearer pulls? A) wrench B) protective gloves C) toolbox drawer D) car jack E) tire iron F) face shield; Passage UIDs: None; Passage Titles: None

Query ID: query_00181_2hop; Query Text: In a video that repairing a car chassis in a busy garage. Before the camera wearer walks around, which statement is true about the object that camera wearer pulls? A) When patching a flat tire. For the adhesive to bond correctly, this must itself be wiped down thoroughly with denatured alcohol before any glue is applied. B) When patching a flat tire. When freeing a tire from its rim in order to fix a flat, this acts as one of the two indispensable prying tools used together with a large screwdriver. C) When getting a good deal on a used car. This is treated as an optional add-on that, when bundled into a dealer's warranty plan, can inflate the overall purchase cost by roughly $1,500 to $3,000. D) When getting a good deal on a used car. When negotiating a prospective buyer needs to examine assessing the condition of a used vehicle. E) When replacing spark plug wires. It is usually mounted on the lower left side of the driver's dashboard near the hood latch for easy access. F) When replacing spark plug wires. It needs to be replaced whenever its outer covering becomes cracked, frayed, or burned, since such damage can let high-voltage sparks jump to the engine.; Passage UIDs: ['8428f073fccc0 4b4c86e0a2cb74df12d8', '8428f073fccc0 4b4c86e0a2cb74df12d8', '049ac37b1333431 6bd3d30dcfc91084', '6e2766a7ab38b4d2ebd343d857686e087', '627664a7ab38b4d2ebd343d857686e087']; Passage Titles: ['how to patch a flat tire', 'how to patch a flat tire', 'how to get a good deal on a used car', 'how to replace spark plug wires 1', 'how to replace spark plug wires 1']

Query ID: query_00182; Query Text: In a video that repairing a car chassis in a busy garage. After the camera wearer walks around, what is the object that camera wearer puts onto the tractor? A) metal toolbox B) spare tire C) red fuel can D) tire chain E) yellow safety helmet F) hose pipe; Passage UIDs: None; Passage Titles: None

Query ID: query_00182_2hop; Query Text: In a video that repairing a car chassis in a busy garage. After the camera wearer walks around, which statement is true about the object that camera wearer pulls? A) toolbox B) air hose C) welding torch D) shop rag E) jack stand F) face shield; Passage UIDs: None; Passage Titles: None

Query ID: query_00183; Query Text: In a video that repairing a car chassis in a busy garage. Before the camera wearer walks around, which statement is true about the object that camera wearer pulls? A) When replacing brakes on a john deere 5105 tractor. It is best situated beneath the tractor after the front jack is removed. C) When installing a camshaft. Mechanics twist this around the camshaft journals to apply oil conditioner evenly before sliding the cam back into the block. D) When installing a camshaft. After the camshaft and related valve-gear parts are cleaned with solvent, they are set in so it so they can dry completely before reassembly. E) When replacing spark plug wires. It is usually mounted on the lower left side of the driver's dashboard near the hood latch for easy access. F) When replacing spark plug wires. It needs to be replaced whenever its outer covering becomes cracked, frayed, or burned, since such damage can let high-voltage sparks jump to the engine.; Passage UIDs: ['8d1949d78e40459 4b91fcd05099fddb', '354b4 0eaaf6a488ca00bf4082dc62e9', '354b40eaaf6a488ca00bf4082dc82e9', '627664a7ab38b4d2ebd343d857686e087', '627664a7ab38b4d2ebd343d857686e087']; Passage Titles: ['how to replace brakes on a john deere 5105 tractor', 'how to install a camshaft', 'how to install a camshaft', 'how to replace spark plug wires 1', 'how to replace spark plug wires 1']

Query ID: query_00184; Query Text: In a video that shows someone walking toward an empty outdoor basketball court. Before the camera wearer walks around, what is the object that the camera wearer touches? A) basketball B) metal fence C) concrete bench D) smartphone E) baseball cap F) camera; Passage UIDs: None; Passage Titles: None

1350

1351 Query ID: query_00184_2hop; Query Text: In a video that shows someone walking toward an empty outdoor basketball court. Before the camera wearer walks around, which statement is true about the object that the camera wearer touches? A) When pumping a spalding neverflat basketball. For a refund or replacement, customers must ship it to Spalding's service center in Akron, Ohio together with their sales receipt. B) When pumping a spalding neverflat basketball. Pumping it at any time during its one-year limited warranty period will void that warranty. C) When pumping a spalding neverflat basketball. Opening the little black rubber plug that covers its valve voids the standard one-year Neverflat warranty supplied with most modern versions of this headgear. D) When improving your effectiveness at running back in football. Strength coaches recommend using it to record every weight-training session so running backs can evaluate their lifting technique throughout the three-to-five workouts each week. F) When improving your effectiveness at running back in football. Running backs are told to master catching the ball with both hands before attempting flashy one-handed grabs intended only to impress it.; Passage UIDs: ['87c7c58b2f764e1c825d402824dec016', '87c7c58b2f764e1c825d402824dec016', '80302f84acdf48119418ce0acb320e2?', '80302f84acdf48119418ce0acb320e2?', 'a8790d3adee24f94bab1d959f1a6a398', 'a8790d3adee24f94bab1d959f1a6a398']; Passage Titles: ['how to pump a spalding neverflat basketball 1', 'how to pump a spalding neverflat basketball 1', 'how to pump a spalding neverflat basketball 2', 'how to pump a spalding neverflat basketball 2', 'how to improve your effectiveness at running back in football', 'how to improve your effectiveness at running back in football']

1352

1353 Query ID: query_00185; Query Text: In a video that driving a car along a tree-lined highway. Before the camera wearer drives past a yellow truck, what is the object that camera wearer drives? A) car B) bus C) bicycle D) scooter E) pickup truck F) motorcycle; Passage UIDs: None; Passage Titles: None

1354 Query ID: query_00185_2hop; Query Text: In a video that driving a car along a tree-lined highway. Before the camera wearer drives past a yellow truck, which statement is true about the object that camera wearer drives? A) When choosing the perfect first car. It is highlighted as the vehicle type that consistently delivers the best fuel economy for novice drivers. B) When choosing the perfect first car. It is listed as one of three body-style choices—alongside compact cars and midsize sedans—when deciding on a first vehicle. C) When barspinning. To master a barspin on it, riders are instructed to lower the seat to its minimum height so that the seat-pinch can be executed more readily. D) When barspinning. For stationary barspin practice, it can be held motionless by pressing its wheels firmly against a chain-link fence while the front wheel is lifted.

1355 E) When transporting a motorcycle. When securing it for trailer transport, using bungee cords instead of ratchet straps is recommended because the elastic material protects painted parts from abrasion. F) When transporting a motorcycle. For optimal stability, a companion should press downward on the handlebars with their body weight while the ratchet straps are tightened so that its front suspension is fully compressed.; Passage UIDs: ['?ce09abc081c34c3b9b54c15221555a8', '?ce09abc081c34c3b9b54c15221555a8', '7f2875bee77d404aa0f151b4d940e866', '7f2875bee77d404aa0f151b4d940e866', '5922e904447044060b03ae7c86518773', '5922e904447044060b03ae7c86518773']; Passage Titles: ['how to choose the perfect first car', 'how to choose the perfect first car', 'how to barspin 2', 'how to barspin 2', 'how to transport a motorcycle 2', 'how to transport a motorcycle 2']

1356 Query ID: query_00186; Query Text: In a video that repeatedly shooting basketballs into an outdoor hoop. Before the camera wearer takes the ball, what is the object that the man throws into the basket ball hoop? A) water bottle B) frisbee C) backpack D) cell phone E) sunglasses F) ball; Passage UIDs: None; Passage Titles: None

1357 Query ID: query_00187; Query Text: In a video that repeatedly shooting basketballs into an outdoor hoop. After a man takes the ball, what is the object that the man throws into the basket ball hoop? A) frisbee B) water bottle C) backpack D) shoe E) hat F) ball; Passage UIDs: None; Passage Titles: None

1358 Query ID: query_00187_2hop; Query Text: In a video that repeatedly shooting basketballs into an outdoor hoop. After a man takes the ball, which statement is true about the object the man throws into the basket ball hoop? A) When determining if you are a tall girl. The text advises tall girls to remove it while slow-dancing so that shorter partners will not feel uncomfortable about their own stature. B) When determining if you are a tall girl. It is mentioned only metaphorically, appearing solely within an empty-based idiom and never as an actual article of clothing. C) When getting in good basketball shape. The regimen recommends selecting it in an insulated stainless-steel design to prevent condensation from making other gear wet. D) When getting in good basketball shape. The gear list allows filling it with either plain water or a sports drink such as Gatorade for hydration breaks. E) When shooting like kevin durant. Kevin Durant seldom passes it because his offensive style relies mainly on isolation moves and step-jukes. F) When shooting like kevin durant. Even from point-blank range, Kevin Durant typically hops before sending it toward the hoop.; Passage UIDs: ['7c9e88e6ceb940d0bc1c1d198e3ca058', '7c9e88e6ceb940d0bc1c1d198e3ca058', '4fead56e17a1401aaf8b43b234507366f', '4fead56e17a1401aaf8b43b234507366f', 'cf440adec2f747aaa720332b5a304cdc', 'cf440adec2f747aaa720332b5a304cdc']; Passage Titles: ['how to determine if you are a tall girl', 'how to determine if you are a tall girl', 'how to get in good basketball shape', 'how to get in good basketball shape', 'how to shoot like kevin durant 1', 'how to shoot like kevin durant 1']

1359 Query ID: query_00188; Query Text: In a video that repeatedly shooting basketballs into an outdoor hoop. Before a man takes the ball, what is the object that camera wearer takes? A) water bottle B) towel C) cap D) mobile phone E) jump rope F) ball; Passage UIDs: None; Passage Titles: None

1360 Query ID: query_00188_2hop; Query Text: In a video that repeatedly shooting basketballs into an outdoor hoop. Before a man takes the ball, which statement is true about the object that camera wearer takes? A) When trying olympics inspired fitness for kids. This is described as something to be rolled up into a tight cylinder so children can use it as an improvised relay baton during family track races. B) When trying olympics inspired fitness for kids. It is suggested that families lay it on the grass so children can try to jump the entire length of it as part of Olympic-inspired games. C) When getting in good basketball shape. During the drills, the plan suggests placing it on the court to serve as a visual target for crossover moves. D) When getting in good basketball shape. The gear list allows filling it with either plain water or a sports drink such as Gatorade for hydration breaks. E) When shooting like kevin durant. Kevin Durant typically holds onto it, using quick step-jukes rather than passing to create the space for his jump shot. F) When shooting like kevin durant. Kevin Durant often finds room by moving into space and briefly passing it to a teammate before receiving it back for the shot.; Passage UIDs: ['50d0c3b14af84444a63a2d743442bf0f', '50d0c3b14af84444a63a2d743442bf0f', '4fead56e17a1401aaf8b43b234507366f', '4fead56e17a1401aaf8b43b234507366f', 'cf440adec2f747aaa720332b5a304cdc', 'cf440adec2f747aaa720332b5a304cdc']; Passage Titles: ['how to try olympics inspired fitness for kids 1', 'how to try olympics inspired fitness for kids 1', 'how to get in good basketball shape', 'how to get in good basketball shape', 'how to shoot like kevin durant 1', 'how to shoot like kevin durant 1']

1361

1362 Query ID: query_00189; Query Text: In a video that sorting screws and nails on a cluttered workbench. Before the camera wearer walks around in the garage, what is the object that the camera wearer puts onto the workbench? A) screw B) bolt C) washer D) nut E) drill bit F) nail; Passage UIDs: None; Passage Titles: None

1363 Query ID: query_00189_2hop; Query Text: In a video that sorting screws and nails on a cluttered workbench. Before the camera wearer walks around in the garage, which statement is true about the object that the camera wearer puts onto the workbench? A) When making a chimney starter charcoal starter. It is used only once during construction because every drilling step requires the exact same bit with an the bolt head, eliminating any need to change sizes. B) When making a chimney starter charcoal starter. It is first chosen wider than the bolt's head to bore halfway through the handle and is then exchanged for a bit that matches the 4-inch bolt's diameter to finish the opening. C) When making a chimney starter charcoal starter. During construction of the chimney starter, it is flattened with a hammer after drilling to remove sharp edges along its rim. D) When making a chimney starter charcoal starter. Each bolt is secured by using it three times—once against the handle, once outside the can, and once inside the can. E) When building a raccoon trap. Assembly directions require bending this fastener into a hook shape so it can prop the sliding door permanently open until the bait is taken. F) When building a raccoon trap. A length of string or wire is secured just behind the head of this fastener so that, when the mousetrap's hammer bar snaps, it yanks the fastener free and allows the door to drop shut.; Passage UIDs: ['73a79396490d47bc95d0a1532d75f48b', '73a79396490d47bc95d0a1532d75f48b', '961124b195264a5f987d695670e085fb', '961124b195264a5f987d695670e085fb']; Passage Titles: ['how to make a chimney starter charcoal starter', 'how to make a chimney starter charcoal starter', 'how to build a raccoon trap', 'how to build a raccoon trap']

1364

1365 Query ID: query_00190; Query Text: In a video that fixing household items at a cluttered indoor workspace. Before the a man opens the water bottle, what is the object that the man picks up from the table? A) screwdriver B) coffee mug C) pair of scissors D) cell phone E) notebook F) water bottle; Passage UIDs: None; Passage Titles: None

1366 Query ID: query_00190_2hop; Query Text: In a video that fixing household items at a cluttered indoor workspace. Before the a man opens the water bottle, which statement is true about the object that the man picks up from the table? A) When making space shoes. It is praised as the perfect blotting material for softening paint details like clouds and nebulas on the footwear design. B) When making space shoes. It is kept nearby expressly to record how many compliments the finished galaxy shoes attract. C) When being safe when staying home alone girls. It is most secure when clipped to the outside of your backpack, where it is plainly visible and easily grabbed in a hurry. D) When being safe when staying home alone girls. It ought to be connected to a charger as soon as the battery indicator reaches roughly twenty-five percent to ensure it is ready for any emergency. E) When making a doll house into a hamster cage. A common technique is to dangle it from the dollhouse ceiling on fishing line so the hamster has to stand erect to access the spout, encouraging exercise. F) When making a doll house into a hamster cage. Instead of drilling through the wooden wall of the dollhouse, it can simply be slid into a cardboard tube with a hole cut so that its nozzle protrudes for the hamster to drink.; Passage UIDs: ['73a79396490d47bc95d0a1532d75f48b', '73a79396490d47bc95d0a1532d75f48b', '2bf8c84d2790408691806a2dacd2068f', '2bf8c84d2790408691806a2dacd2068f', '5bb50ea51357482484b5a7fe16fdd50?', '5bb50ea51357482484b5a7fe16fdd50?']; Passage Titles: ['how to make space shoes 2', 'how to make space shoes 2', 'how to be safe when staying home alone girls', 'how to be safe when staying home alone girls', 'how to make a doll house into a hamster cage', 'how to make a doll house into a hamster cage']

1367

1368 Query ID: query_00191; Query Text: In a video that fixing household items at a cluttered indoor workspace. Before the a man walks to the passage, what is the object that man puts onto the table? A) hammer B) coffee mug C) screwdriver D) roll of duct tape E) pair of safety goggles F) water bottle; Passage UIDs: None; Passage Titles: None

1369 Query ID: query_00191_2hop; Query Text: In a video that fixing household items at a cluttered indoor workspace. Before the a man walks to the passage, which statement is true about the object that man puts onto the table? A) When making a punching bag. It is initially placed over the pipe's drilled holes to block debris from entering while the PVC is being cut. B) When making a punching bag. It is first pressed onto the carpet padding nearest the base of the PVC pipe and then wound upward in overlapping layers to cover the padding along the entire length of the punching bag. C) When changing a kids room to a tweens room. It is highlighted as the preferred tool for applying detail paint in hard-to-reach corners of the redesigned bedroom. D) When changing a kids room to a tweens room. It is mentioned as sufficient for installing most simple additions during the room makeover. E) When making a doll house into a hamster cage. Some keepers recommend filling it with a diluted dish-soap solution along with water to keep algae from forming in the cage's drinking supply. F) When making a doll house into a hamster cage. Instead of drilling through the wooden wall of the dollhouse, it can simply be slid into a cardboard tube with a hole cut so that its nozzle protrudes for the hamster to drink.; Passage UIDs: ['6d30695768db4897f8f244b1dad953a1', '6d30695768db4897f8f244b1dad953a1', 'bc1afd61a3b5491a4e6c169990789a', 'bc1afd61a3b5491a4e6c169990789a', '5bb50ea51357482484b5a7fe16fdd50?', '5bb50ea51357482484b5a7fe16fdd50?']; Passage Titles: ['how to make a punching bag 1', 'how to make a punching bag 1', 'how to change a kids room to a tweens room', 'how to change a kids room to a tweens room', 'how to make a doll house into a hamster cage', 'how to make a doll house into a hamster cage']

1370 Query ID: query_00192; Query Text: In a video that fixing household items at a cluttered indoor workspace. After the a man walks to the passage, what is the object that camera wearer adjusts? A) lamp B) chair C) toolbox D) curtain E) painting F) shoe; Passage UIDs: None; Passage Titles: None

1371 Query ID: query_00192_2hop; Query Text: In a video that fixing household items at a cluttered indoor workspace. After the a man walks to the passage, which statement is true about the object that camera wearer adjusts? A) When living alone happily. Because of its significant weight, the text recommends dismantling it or hiring assistance before moving it to prevent personal injury. B) When living alone happily. The text presents it as a quirky item you can hang on your wall to personalize your space and lift your mood when you live alone. C) When making a simple cat bed. The guide proposes taping a soft piece of fabric over the doorway and using it to grant the cat more privacy inside the box. D) When cleaning your house for a sleepover. Sprinkling baking soda on it before vacuuming is recommended to eliminate any lingering odors before friends arrive. E) When cleaning your house for a sleepover. This is characterized as the most common piece of clutter in the hall and should be stored neatly in a wardrobe or on a dedicated rack.; Passage UIDs: ['e6lbcb5bf8c4b85a862lad4ca06feb97', 'e6lbcb5bf8c4b85a862lad4ca06feb97', '3669dd1fbb0418c8de3d080bdf57ea?', '3669dd1fbb0418c8de3d080bdf57ea?', '181b6436a282407b3b59081828be63777', '181b6436a282407b3b59081828be63777']; Passage Titles: ['how to live alone happily', 'how to live alone happily', 'how to clean your house for a sleepover', 'how to clean your house for a sleepover']

1372

1373 Query ID: query_00193; Query Text: In a video that placing a wooden plank onto the wall. After a man walks in the room, what is the object that the man puts onto the table? A) metal pipe B) paint can C) toolbox D) folded blanket E) laptop computer F) wood; Passage UIDs: None; Passage Titles: None

1374 Query ID: query_00193_2hop; Query Text: In a video that placing a wooden plank onto the wall. After a man walks in the room, which statement is true about the object that the man puts onto the table? A) When replacing the lcd inverter in a hp dv6000 series laptop. To reach its LCD inverter, it requires removing four perimeter screws that are concealed beneath rubber bumpers on the screen bezel. B) When replacing the lcd inverter in a hp dv6000 series laptop. Hidden beneath small rubber caps, it houses six screws around the display's perimeter that must be removed before the LCD bezel can be separated. C) When redoing your room for a hundred or less. It is promoted as the preferred cover for sliding windows that travelers can pack and use to recreate a comforting sense of home wherever journeys take them. D) When redoing your room for a hundred or less. For this project, it is wedged between a replacement steel beam and the floor joists to absorb vibrations after a load-bearing wall has been taken down. E) When making a squirrel feeder. The project requires only three sections cut from it—a base, a back, and a lid—because the feeder design does not call for separate side panels. F) When making a squirrel feeder. The recommended material for the feeder is it in the form of a cedar fence slat measuring roughly 5 × 12 × 4 feet, which is then divided into the base, back, lid, and side pieces.; Passage UIDs: ['?cf0ba9a78088b2bbdadf459f02b96c3', '?cf0ba9a78088b2bbdadf459f02b96c3', '1faa2e6a90db4167a5c2251c83f6f43b', '1faa2e6a90db4167a5c2251c83f6f43b', '2b30e5b4bcd745f783acd7d805cbf867', '2b30e5b4bcd745f783acd7d805cbf867']; Passage Titles: ['how to replace the lcd inverter in a hp dv6000 series laptop', 'how to replace the lcd inverter in a hp dv6000 series laptop', 'how to redo your room for a hundred or less', 'how to redo your room for a hundred or less', 'how to make a squirrel feeder 3', 'how to make a squirrel feeder 3']

1375 Query ID: query_00194; Query Text: In a video that placing a wooden plank onto the wall. After a man stands in the room, what is the object that the man picks from the table? A) hammer B) tape measure C) screwdriver D) paintbrush E) coffee mug F) wood; Passage UIDs: None; Passage Titles: None

1376 Query ID: query_00194_2hop; Query Text: In a video that placing a wooden plank onto the wall. After a man stands in the room, which statement is true about the object that the man picks from the table? A) When coping with homesickness while traveling. Travel guides emphasize that bringing it adds needless weight, so anyone committed to packing light should definitely exclude it. B) When coping with homesickness while traveling. It is presented as a familiar everyday item that travelers can pack and use to recreate a comforting sense of home wherever journeys take them. C) When making a squirrel feeder. To apply a coat of standard white paint to every cut piece of lumber, after which the boards are left to dry for several hours. D) When making a squirrel feeder. The project requires only three sections cut from it—a base, a back, and a lid—because the feeder design does not call for separate side panels. E) When making a squirrel feeder. The recommended material for the feeder is it in the form of a cedar fence slat measuring roughly 5 × 12 × 4 feet, which is then divided into the base, back, lid, and side pieces.; Passage UIDs: ['?cf0ba9a78088b2bbdadf459f02b96c3', '?cf0ba9a78088b2bbdadf459f02b96c3', '1faa2e6a90db4167a5c2251c83f6f43b', '1faa2e6a90db4167a5c2251c83f6f43b', '2b30e5b4bcd745f783acd7d805cbf867', '2b30e5b4bcd745f783acd7d805cbf867']; Passage Titles: ['how to cope with homesickness while traveling 1', 'how to cope with homesickness while traveling 1', 'how to make a squirrel feeder 3', 'how to make a squirrel feeder 3', 'how to make a squirrel feeder 3', 'how to make a squirrel feeder 3']

1377

1378 Query ID: query_00195; Query Text: In a video that handing wooden planks to a woman. Before the camera wearer picks another wood, which statement is true about the object that camera wearer picks? A) Hammer B) Metal pipe C) Brick D) Plastic sheet E) Cardboard box F) Wood plank; Passage UIDs: None; Passage Titles: None

1379 Query ID: query_00195_2hop; Query Text: In a video that handing wooden planks to a woman. Before the camera wearer picks another wood, which statement is true about the object that camera wearer picks? A) When making a cat tree. It must first be sliced lengthwise into a concave shape with a utility knife before any carpet or rope is added. B) When making a cat tree. It can serve as a vertical support in a DIY cat tree, functioning as an alternative to PVC pipe. C) When removing interior walls. It is wedged between a replacement steel beam and the floor joists to absorb vibrations after a load-bearing wall has been taken down. D) When removing interior walls. It can be spread on the floor as a drop sheet to catch dust and debris while an interior wall is being demolished. E) When making a duct tape bookmark. For decorative detailing, this is trimmed into slender 0.5-centimetre-wide strips that form the frames around the cabin's doors and windows.; Passage UIDs: ['72314cbca08a44a91982607409e8bec1e', '2314cbca08a44a91982607409e8bec1e', 'ad6e8592led3499aa356310d1ccc2583f', 'ad6e8592led3499aa356310d1ccc2583f', '5939ef72a29c440ab6aab2c5fd0bdd39', '5939ef72a29c440ab6aab2c5fd0bdd39']; Passage Titles: ['how to make a cat tree 1', 'how to make a cat tree 1', 'how to remove interior walls', 'how to remove interior walls', 'how to make a copy facial tissue box cover', 'how to make a copy facial tissue box cover']

1380

1381 Query ID: query_00196; Query Text: In a video that shows assembling a household item on a cluttered table. After the camera wearer folds them together with both hands, what is the object that the camera wearer drops onto the table? A) small metal screw B) rubber band C) paper clip D) plastic bottle cap E) coin F) cellotape piece; Passage UIDs: None; Passage Titles: None

1382 Query ID: query_00196_2hop; Query Text: In a video that shows assembling a household item on a cluttered table. After the camera wearer folds them together with both hands, which statement is true about the object that the camera wearer drops onto the table? A) When making a mini crossbow. It is bored with only one pair of holes and holds a single wooden stick that doubles as both the handle and the limbs of the miniature crossbow. B) When making a mini crossbow. It is given three distinct pairs of side holes—one near the lower rim, a second halfway up, and a third close to the flat top—so that two popsicle sticks and a cotton swab can be threaded through to form the frame of a mini crossbow. C) When making a bracelet holder. The crafting instructions advocate carving a vertical channel down its length instead of simply drawing a side-to-side center line for alignment. D) When making a bracelet holder. The tutorial notes that, if unavailable, it can be replaced by a rolled-up magazine stuffed inside a paper-towel tube. E) When making a duct tape bookmark. It is wrapped around the very end of the tail to create a loop that keeps the tassel from pulling back through the hole in the bookmark.; Passage UIDs: ['?c75ad9a74e14a2798225e1f44d28f72', '?c75ad9a74e14a2798225e1f44d28f72', '30f83cdc05524daab2456f2e89bc14c9', 'a97f59401f4e4f1888864b766b849e46', 'a97f59401f4e4f1888864b766b849e46']; Passage Titles: ['how to make a mini crossbow 1', 'how to make a mini crossbow 1', 'how to make a bracelet holder 1', 'how to make a bracelet holder 1']

1383

1384 Query ID: query_00197; Query Text: In a video that shows organizing scattered artwork and papers on the floor. Before the camera wearer turns the pages of the book pages on her lap with her left hand, what is the object that the camera wearer closes from her hand? A) notebook B) paintbrush C) scissors D) mobile phone E) glue stick F) paper; Passage UIDs: None; Passage Titles: None

1385 Query ID: query_00197_2hop; Query Text: In a video that shows organizing scattered artwork and papers on the floor. Before the camera wearer turns the pages of the book pages on her lap with her left hand, which statement is true about the object that the camera wearer closes from her hand? A) When organizing your backpack after a break. The guide claims that this should always be kept in a zippered pocket to avoid sticking to papers. B) When organizing your backpack after a break. The organizer's advice notes that it should avoid fully restocking all of your classroom supplies. C) When having fun alone in your bedroom girls only. It is recommended that you keep it switched off and out of sight during solo creative makeovers to avoid distractions. D) When having fun alone in your bedroom girls only. A device you can turn on to stream music through the hole in your bedroom, even a quiet sleep-music station, will inevitably alert your parents to your activity. E) When staying up late at night writing a paper without your parents knowing. This advises you to gather your books, a snack, water, and a flashlight in advance so you can work through the night without needing to leave your room.; Passage UIDs: ['b0fa76f726bf42fbb056ef13d419994l', 'b0fa76f726bf42fbb056ef13d419994l', '36daa380e01401b0a4177cdfaacec91', '36daa380e01401b0a4177cdfaacec91']; Passage Titles: ['how to organize your backpack after a break', 'how to organize your backpack after a break', 'how to have fun alone in your bedroom girls only', 'how to have fun alone in your bedroom girls only', 'how to stay up late at night writing a paper without your parents knowing', 'how to stay up late at night writing a paper without your parents knowing']

1386

1387 Query ID: query_00198; Query Text: In a video that scraping paint from a window frame with putty knife. Before the camera wearer adjusts the scraper, what is the object that the camera wearer taps? A) When making an advent calendar. It is recommended that scraping paint off a window frame with putty knife. B) When making an advent calendar; Passage UIDs: None; Passage Titles: None

1388 Query ID: query_00198_2hop; Query Text: In a video that scraping paint from a window frame with putty knife. Before the camera wearer adjusts the scraper, which statement is true about the object that the camera wearer taps? A) When making an advent calendar. It is utilized to paint the numerical labels onto the fronts of the matchbox drawers for the advent calendar. B) When painting an exterior door. It is the tool that you angle upward at 45 degrees and tap with a hammer to pry the hinge pin out of a door hinge. D) When building an anglo saxon house. When building and cracks on the door surface.; Passage UIDs: None; Passage Titles: ['how to make an advent calendar 2', 'how to build an anglo saxon house', 'how to paint an exterior door']

1389

1390 Query ID: query_00199; Query Text: In a video that scraping paint from a window frame with putty knife. Before the camera wearer taps on the scraper, which statement is true about the object that camera wearer adjusts? A) When spraying paint a nerf gun. It is described as the side abrasive tool utilized for scuffing brand-new plastic components and color coats. B) When painting an exterior door. Before the filler is left to dry, this is used to press and smooth bondo or wood filler into nail holes and cracks on the door surface.; Passage UIDs: None; Passage Titles: ['how to spray paint a nerf gun', 'how to paint an exterior door']

1391

1392 Query ID: query_00200; Query Text: In a video that scraping paint from a window frame with putty knife. After the camera wearer adjusts the scraper, which statement is true about the object that the camera wearer taps? A) When making a guinea pig cage. It is included in the recommended collection of basic tools, alongside an electric drill and screwdriver. B) When painting an exterior door. Before the filler is left to dry, this is used to press and smooth bondo or wood filler into nail holes and cracks on the door surface.; Passage UIDs: None; Passage Titles: ['how to make a guinea pig cage', 'how to paint an exterior door']

1393

1394 Query ID: query_00201; Query Text: In a video that scraping paint from a window frame with putty knife. Before the camera wearer adjusts the scraper, what is the object that the camera wearer taps? A) paintbrush B) hammer C) screwdriver D) window frame E) paint can F) scraper; Passage UIDs: None; Passage Titles: None

1395 Query ID: query_00201_2hop; Query Text: In a video that scraping paint from a window frame with putty knife. Before the camera wearer adjusts the scraper, which statement is true about the object that the camera wearer taps? A) When making an advent calendar. It is required to paint the exterior shells of every matchbox before any decorative paper or washi tape is added. B) When making the outside of your home. It should be inspected for blockages and cracks because it channels rainwater away from the roof edges. D) When painting an exterior door. Before the filler is left to dry, this is used to press and smooth bondo or wood filler into nail holes and cracks on the door surface.; Passage UIDs: None; Passage Titles: ['how to make an advent calendar 2', 'how to improve the outside of your home', 'how to paint an exterior door']

1396

1397 Query ID: query_00202; Query Text: In a video that shows assembling furniture with screws and tools on carpet. Before the camera wearer changes the position of some objects on the floor with both hands, what is the object that camera wearer turns from the room? A) door handle B) thermostat C) ceiling fan D) window blind E) water faucet F) light; Passage UIDs: None; Passage Titles: None

1398 Query ID: query_00202_2hop; Query Text: In a video that shows assembling furniture with screws and tools on carpet. Before the camera wearer changes the position of some objects on the floor with both hands, which statement is true about the object that the camera wearer turns from the room? A) When making a chlorine dosifier for a cabin. In the system diagram, this sits above the tee, drain pipe, and drain valve, which are depicted beneath it. C) When setting up a home office in your garage. The usual guideline is to have it span exactly three-fourths of the home workspace's perimeter when converting a garage corner into an office. D) When cleaning your house for a sleepover. You should be welcomed in by leaving the curtains open.; Passage UIDs: None; Passage Titles: ['how to make a chlorine dosifier for a cabin', 'how to set up a home office in your garage 2', 'how to clean your house for a sleepover']

1399

1400 Query ID: query_00203; Query Text: In a video that shows attaching a metal pipe fitting onto a pipe. Before the camera wearer attaches the metal pipe fitting to the metal pipe with his hands, what is the object that woman drops onto the floor? A) screwdriver B) tape measure C) rag D) pair of work gloves E) small plastic bag F) wrench; Passage UIDs: None; Passage Titles: None

1401 Query ID: query_00203_2hop; Query Text: In a video that shows attaching a metal pipe fitting onto a pipe. After the camera wearer attaches the metal pipe fitting to the metal pipe with his hands, which statement is true about the object that woman drops onto the floor? A) When removing interior walls. It is expressly recommended as protective handwear when using the reciprocating saw to sever studs from the wall plates. B) When removing interior walls. The demolition procedure never mentions or recommends using this at any point in the wall-removal process. C) When building a coffee table. It is employed to polish the self-adhesive veneer edging until it sits perfectly flush with the plywood top. D) When building a coffee table. It can substitute it for a brush to lay on each of the three coats of varnish. E) When making a windmill. According to the plans, it should be applied only to the hexagonal top piece before any other parts are assembled. F) When making a windmill. During the project, it is used to sand every wooden part except the solid dowel, ensuring a smooth, even surface before painting.; Passage UIDs: None; Passage Titles: ['how to remove interior walls', 'how to build a coffee table', 'how to make a windmill']

1402

1403 Query ID: query_00204; Query Text: In a video that unfolding and positioning an aluminum ladder on grass. Before the camera wearer walks beside the ladder, what is the object that the camera wearer puts down? A) toolbox B) backpack C) garden hose D) bag E) paint can F) ladder; Passage UIDs: None; Passage Titles: None

Query ID: query_00204_2hop; Query Text: In a video that unfolding and positioning an aluminum ladder on grass. Before the camera wearer walks beside the ladder, which statement is true about the object that the camera wearer puts down? A) When making a flagpole. The instructions direct that the same one used to mix the concrete will later serve as the permanent base for the flagpole. B) When making a flagpole. Its flagpole, its height dictates how far up the PVC pole needs to be wrapped in plastic sheeting before the concrete is poured. D) When building a garage work bench. Because of its significant weight, builders are instructed to use exclusively heavy-duty standalone units rather than models that attach to a workbench. Legroom can still be preserved beneath the unit so that it can be relocated whenever needed. F) When getting up to the top bunk of a bunk bed. Safety instructions specify that it must be firmly bolted to the bunk bed before anyone uses it to climb to the top bunk.; Passage UIDs: ['fe434a616899453fa6e201faaefad595', 'fe434a616899453fa6e201faaefad595', '32c7ae0b063943ee81bf0a96ed3f08e4', '32c7ae0b063943ee81bf0a96ed3f08e4', 'bf8ce0508c8d4be78956abd8cd3329aa', 'bf8ce0508c8d4be78956abd8cd3329aa']

Passage Titles: ['how to make a flagpole', 'how to make a flagpole', 'how to build a garage work bench 2', 'how to build a garage work bench 2', 'how to get up to the top bunk of a bunk bed 1', 'how to get up to the top bunk of a bunk bed 1']

Query ID: query_00205; Query Text: In a video that adjusting metal rods on welding table near vintage truck. Before the camera wearer adjusts the metal rod on the welding table with his right hand, what is the object that the camera wearer drops from the left hand onto the welding table? A) copper wire B) welding clamp C) small hammer D) tape measure E) pair of pliers F) metal rod; Passage UIDs: None

Query ID: query_00205_2hop; Query Text: In a video that adjusting metal rods on welding table near vintage truck. Before the camera wearer adjusts the metal rod on the welding table with his right hand, which statement is true about the object that the camera wearer drops from the left hand onto the welding table? ...

up slightly during the swing. E) When playing basketball. During basketball games, it is customarily left resting on the chalk-marked home base between innings to show which team is up to shoot. F) When playing basketball. At the very beginning of a basketball match, the competitor who first scores from the foul line is given it to start play.; Passage UIDs: ['96ce59d25a414fd09e54cef4a06bba4a', '96ce59d25a414fd09e54cef4a06bba4a', '2cf85ba90c094a010a1390d050c84285e', '2cf85ba90c094a010a1390d050c84285e', 'e0561fe5cd0d4dd8915694848fad26a']; Passage Titles: ['how to throw a curveball 3', 'how to hit a slowpitch softball', 'how to hit a slowpitch softball', 'how to play basketball', 'how to play basketball']?

[... dense query log content ...]

Query ID: query_00229_2hop_audio; Query Text: In a video that fixing a bicycle wheel inside a living room. Before the event related to this sound, what is the object that the camera wearer place? A) pump B) helmet C) screwdriver D) basketball E) coffee mug F) bicycle tire; Passage UIDs: None; Passage Titles: None

# D EXPERIMENT DETAILS

## D.1   VIDEO UNDERSTANDING EVALUATION DETAILS

For video understanding evaluation, we use *GPT-4o-mini* with the following prompt to evaluate the correctness of caption of each *correctly retrieved* video segment:

```
# Role
You are an information sufficiency checker.

# Goal
Given a video description and a multiple-choice question stem (without
    answer options), decide whether the description provides enough
    information to reasonably answer the question. Output ONLY `Y` or `N`.

# Decision Rules
Return `Y` if the description contains the main action or event
    mentioned in the question (the "query anchor") and provides at least
    some details about the subject, object, or context involved. The
    description does not need to include every fine-grained attribute,
    but it must mention the relevant action. Return `N` if the
    description omits the anchor event or is too vague to connect to the
    question.

# Inputs
- description: string -textual description of the video
- question_stem: string -the multiple-choice question prompt (WITHOUT
    options)

# Output
Return ONLY a single uppercase character: `Y` or `N` (no extra text,
    punctuation, or explanation).

# Examples
## Question 1
Which statement is true about the object that the camera wearer plugs
    into the socket?"

## Description 1
1. The camera wearer walks into a room with a workbench and various
    tools.
2. ...
...
8. The camera wearer takes a power drill and plugs it into the wall
    socket before continuing.

## Output 1
Y

## Question 2
In a video that repairing a car wheel and brakes using tools, what is
    the object that the camera wearer picks from the bonnet?

## Description 2
1. The camera wearer is working on a car in a garage.
2. ...
... (no info about the bonnet)
8. The camera wearer cleans dust off several tools.

## Output 2
N
```

After determining binary correctness for each retrieved segment, we calculate video understanding accuracy as the proportion of queries—within each query type—that are deemed correct according to the above evaluation.

## D.2 BASELINE IMPLEMENTATION DETAILS

In this section, we provide additional implementation details for the baselines evaluated in section 4.

### D.2.1 GRAPHRAG

For evaluation, we adopt the *local* configuration of *GraphRAG*. We use *GPT-4o-mini* as the chat model and *text-embedding-3-small* as the embedding model. Documents are segmented into chunks of 1200 tokens with an overlap of 100 tokens. This setup ensures that evaluation focuses on the local search capability of GraphRAG.

To perform multiple-choice QA, we slightly modify the prompt file *local_search_system_prompt.txt* to make it answer multiple-choice questions. Specifically, we append the following instructions to the original prompt:

```
---Role---

You are a helpful assistant responding to multiple-choice questions
    about data in the tables provided.

---Goal---

Generate a response of the target length and format that responds to the
    user's question, selecting **one letter choice (A, B, C, ...)** as
    the final answer, and providing a brief explanation.

- You must always output a single letter answer, even if uncertain.
- If unsure, choose the most likely/certain answer based on the data and
    general knowledge.
- Do not output "I don't know" or "No answer."

Points supported by data should list their data references as follows:

"This is an example sentence supported by multiple data references
    [Data: <dataset name> (record ids); <dataset name> (record ids)]."

Do not list more than 5 record ids in a single reference. Instead, list
    the top 5 most relevant record ids and add "+more" to indicate that
    there are more.

For example:

"Person X is the owner of Company Y and subject to many allegations of
    wrongdoing [Data: Sources (15, 16), Reports (1), Entities (5, 7);
    Relationships (23); Claims (2, 7, 34, 46, 64, +more)]."

where 15, 16, 1, 5, 7, 23, 2, 7, 34, 46, and 64 represent the id (not
    the index) of the relevant data record.

Do not include information where the supporting evidence for it is not
    provided.

---Target response length and format---

Output format should be:

**Answer: X**
<one short paragraph explanation>

---Data tables---
```

```
{context_data}

---Goal---

Generate a response of the target length and format that responds to the
    user's question, selecting one multiple-choice option (letter) with
    explanation. Always provide a single letter answer, supported by
    evidence if available, or the most plausible choice if uncertain.

---Target response length and format---

Add sections and commentary to the response as appropriate for the
    length and format. Style the response in markdown.
```

### D.2.2 VIDEORAG (JEONG)

In this variant, we use the *Qwen2.5-VL-3B-Instruct* model for generation. Same with *VideoRAG (Ren)*, we set number of video clips to retrieve as 5 and number of passages to retrieve as 15. To incorporate passage retrieval, we add an additional step to retrieve relevant *WikiHow* articles using the same text embedding model as query embedding in the original paper. And set the number of passages to retrieve as 5. We also slightly modify the original prompt to include passages and make it answer multiple-choice questions. The prompt is as follows:

```
Reference passages: {passages_text}. Considering the given videos,
    explain {query}. Please only give an capitalized option (A, B, C, D,
    E, or F) without any additional text.
```

### D.2.3 VIDEORAG (REN) (WITH OR WITHOUT LOCALIZATION)

**Fallback strategy** Sometimes the prompt triggered safety policies in *GPT-4o-mini*, resulting in no answer. In such cases, we fallback to original *MiniCPM* model to inference locally.

For the main model, we use *GPT-4o-mini* as the chat model and *text-embedding-3-small* as the embedding model. We set number of video clips to retrieve as 5 where each clip is represented by 15 uniformly sampled frames when generating video description. For passage retrieval, we retrieve 6 relevant *WikiHow* articles (one passage per option) using the same text embedding model as query embedding in the original paper. We also slightly modify the original prompt to include passages. We detail the prompts we use below.

Our prompt for query rewriting for visual retrieval is as follows:

```
-Goal-
Rewrite the question as a single short sentence starting with "A video
    that ...".

-Examples-
Question: In a video where a group of hikers are crossing a narrow
    bridge, what is the landscape surrounding them?
Output: A video that depicts hikers crossing a narrow bridge.

Question: In a video showing a crowded marketplace, what is the vendor
    placing on the display table?
Output: A video that shows a crowded marketplace.

Question: In a video where a person is cooking in a small kitchen, which
    statement is true about the object that is being stirred in the pot?
Output: A video that shows a person cooking in a small kitchen.

-Real Data-
Question: {input_text}
Output:
```

Our prompt for query rewriting for passage retrieval is as follows:

```
-Goal-
Given a multiple-choice question (MCQ), produce a JSON array of
    declarative search queries, one per option, that could retrieve text
    passages relevant to judging the options' correctness.

######################
-Examples-
######################

Question:
In a video showing a chef preparing a meal in a kitchen, which statement
    is true about the object that the camera wearer uses to stir soup?
(A) When baking cakes. The object is recommended for folding whipped
    cream into sponge batter to avoid deflating the mixture.
(B) When baking cakes. The object should be avoided for mixing flour
    because its flexible edges cannot break up clumps.
(C) When making scrambled eggs. The object is considered best for
    preventing sticking and producing soft curds in a nonstick pan.
(D) When making scrambled eggs. The object may melt when exposed to high
    stovetop temperatures.
(E) When crafting clay models. The object is often used to scoop and
    shape soft clay before firing.
(F) When crafting clay models. The object should not be used because it
    absorbs water and weakens the clay structure.
Output:
[
  "A passage about when baking cakes, some object is recommended for
      folding whipped cream into sponge batter.",
  "A passage about when baking cakes, some object should be avoided for
      mixing flour because it cannot break up clumps.",
  "A passage about when making scrambled eggs, some object is best for
      preventing sticking and creating soft curds.",
  "A passage about when making scrambled eggs, some object may melt when
      exposed to high heat.",
  "A passage about when crafting clay models, some object is used to
      scoop and shape soft clay.",
  "A passage about when crafting clay models, some object should not be
      used because it absorbs water and weakens the structure."
]

Question:
In a science lab video, which statement is true about the object that
    the man places on the lab bench?
(A) When measuring chemicals for titration. The object should not be
    used for precise volumes since it lacks calibration marks.
(B) When measuring chemicals for titration. The object is recommended
    for exact milliliter measurement to ensure accuracy.
(C) When heating liquids over a Bunsen burner. The object is safe
    because borosilicate material resists cracking from sudden
    temperature changes.
(D) When heating liquids over a Bunsen burner. The object should not be
    placed directly on the flame because it may shatter without
    protective gauze.
(E) When holding biological samples. The object is commonly sterilized
    in an autoclave before introducing living cultures.
(F) When holding biological samples. The object cannot be autoclaved
    since heat and pressure would deform the glass permanently.
Output:
[
  "A passage about when measuring chemicals for titration, some object
      should not be used for precise volumes.",
  "A passage about when measuring chemicals for titration, some object is
      recommended for exact milliliter measurement.",
```

```
  "A passage about when heating liquids, some object is safe because
      borosilicate resists cracking.",
  "A passage about when heating liquids, some object should not be placed
      directly on a flame.",
  "A passage about when holding biological samples, some object is
      commonly sterilized in an autoclave.",
  "A passage about when holding biological samples, some object cannot be
      autoclaved because heat deforms glass."
]

##############################
-Real Data-
#####################
Question:
{input_text}
#####################
Output:
```

Our prompt for keyword extraction for video retrieval is as follows:

```
- Goal -
Given a query, extract the most relevant and concrete keywords that can
    help locate the answer in a video.

Requirements:
1. Output one line of a numbered list of the extracted keywords,
    separated by commas and ending with a period.
2. Do **not** include overly generic or meaningless terms such as
    "object".

#####################
- Examples -
#####################

Question: Which animal does the protagonist encounter in the forest
    scene?
################
Output:
1. animal, 2. protagonist, 3. forest.

Question: What is the weather like during the opening scene of the film?
    (A) Sunny (B) Rainy (C) Snowy (D) Windy
################
Output:
1. weather, 2. opening scene, 3. film, 4. Sunny, 5. Rainy, 6. Snowy, 7.
    Windy.

Question: In a video showing a chef preparing a meal in a kitchen, what
    is the object that the camera wearer uses to stir soup?
#################
Output:
1. chef, 2. meal, 3. kitchen, 4. camera wearer, 5. soup.

Question: In a video showing a chef preparing a meal in a kitchen, which
    statement is true about the object that the camera wearer uses to
    stir soup?
#################
Output:
1. chef, 2. meal, 3. kitchen, 4. camera wearer, 5. soup.

##############################
- Real Data -
#####################
Question: {input_text}
#####################
```

```
Output:
```

Our prompt for generating multiple-choice answers is as follows for *1-hop* queries:

```
---Role---

You are a helpful assistant responding to a multiple-choice question
    with retrieved knowledge.

---Goal---

Generate a concise response that addresses the user's question by
    summarizing relevant information derived from the retrieved text and
    video content. Ensure the response aligns with the specified format
    and length.
Please note that there is only one choice is correct.

---Retrieved Information From Videos---

{video_data}

---Retrieved Text Chunks---

{chunk_data}

---Goal---

Generate a concise response that addresses the user's question by
    summarizing relevant information derived from the retrieved text and
    video content. Ensure the response aligns with the specified format
    and length.
Please note that there is only one choice is correct.

---Notice---
Please provide your answer in JSON format as follows:
{{
    "Answer": "The label of the answer, like A/B/C/D or 1/2/3/4 or
        others, depending on the given query"
    "Explanation": "Provide explanations for your choice. Use sections
        and commentary as needed to ensure clarity and depth. Format the
        response in Markdown."
}}
Key points:
1. Ensure that the "Answer" reflects the correct label format.
2. Structure the "Explanation" for clarity, using Markdown for any
    necessary formatting.
```

Our prompt for generating multiple-choice answers is as follows for *2-hop* queries:

```
---Role---

You are a helpful assistant responding to a multiple-choice question
    with retrieved knowledge.

---Goal---

Generate a concise response that addresses the user's question by
    summarizing relevant information derived from the retrieved text,
    video content, and retrieved passage from a passage corpus. Ensure
    the response aligns with the specified format and length.
Please note that there is only one choice is correct.

---Retrieved Information From Videos---

{video_data}
```

```
---Retrieved Text Chunks---

{chunk_data}

---Retrieved Passages---

{passage_data}

---Goal---

Generate a concise response that addresses the user's question by
    summarizing relevant information derived from the retrieved text and
    video content. Ensure the response aligns with the specified format
    and length.
Please note that there is only one choice is correct.

---Notice---
Please provide your answer in JSON format as follows:
{{
    "Answer": "The label of the answer, like A/B/C/D or 1/2/3/4 or
        others, depending on the given query"
    "Explanation": "Provide explanations for your choice. Use sections
        and commentary as needed to ensure clarity and depth. Format the
        response in Markdown."
}}
Key points:
1. Ensure that the "Answer" reflects the correct label format.
2. Structure the "Explanation" for clarity, using Markdown for any
    necessary formatting.
```

Our prompt for video captioning is as follows:

```
List only the main actions shown across the provided screenshots,
    presented step by step as a numbered list (1., 2., 3., ...).
If any intermediate steps appear missing between screenshots, reasonably
    infer and include them to ensure continuity.
Make each action as specific and detailed as possible, focusing on the
    key elements of the query -if some elements in it are not relevant
    or visible, you may ignore them.
Avoid describing the environment, visual appearance, or any irrelevant
    details.
Output only the ordered sequence of actions.
```

All other settings are the same as default.

### D.2.4 Additional Details on Localization

We use the following prompt to generate text for video localization for each query:

```
-Goal-
Given a query about a video, output a JSON object with two fields:

1. "description": the anchor event in the video, used to localize the
    relevant part.
  - If the query explicitly contains a temporal cue ("before ..." or
      "after ..."),
    then "description" must be that anchor event.
    Example: "before they reach the summit" →"description": "they reach
        the summit"
    Example: "after the camera wearer chops vegetables" →"description":
        "the camera wearer chops vegetables"
  - If the query does not contain such cues, then "description" must be
      the event itself
```

```
    that needs to be observed to answer the query.
    Example: "what does the goalkeeper do when the ball is kicked toward
        the goal?"
    →"description": "the goalkeeper kicks the ball toward the goal"

2. "position": where the part that must be watched lies relative to this
    anchor.
   One of: "before", "after", or "within".

-Examples-
Question: In a video that shows children playing soccer in a park, what
    is the object that goalkeeper kicks toward the goal?
Output: {{
 "description": "the goalkeeper kicks something toward the goal",
 "position": "within"
}}

Question: In a video that shows children playing soccer in a park, which
    statement is true about the object that goalkeeper kicks toward the
    goal?
Output: {{
 "description": "the goalkeeper kicks something toward the goal",
 "position": "within"
}}

Question: In a video that shows a chef preparing food in a restaurant
    kitchen, after the camera wearer chops vegetables, what is the
    object that the camera wearer place into the pan?
Output: {{
 "description": "the camera wearer chops vegetables",
 "position": "after"
}}

Question: In a video that shows a chef preparing food in a restaurant
    kitchen, after the camera wearer chops vegetables, which statement
    is true about the camera wearer place into the pan?
Output: {{
 "description": "the camera wearer chops vegetables",
 "position": "after"
}}

Question: In a video that shows a group of people hiking up a mountain
    trail, before people reach the summit, what is the object that one
    of the hikers adjust on their backpack?
Output: {{
 "description": "people reach the summit",
 "position": "before"
}}

Question: In a video that shows a group of people hiking up a mountain
    trail, before people reach the summit, which statement is true about
    the object that one of the hikers adjust on their backpack?
Output: {{
 "description": "people reach the summit",
 "position": "before"
}}

-Real Data-
Question: {input_text}

Output a parseable JSON object with "description" and "position" fields,
    without any additional text.
```

If the *position* turns to be *before* or *after*, which is a temporal cue, we shift the localization window by 1 second accordingly.

