# OpenReview forum: "EgoFact: A Benchmark for Multi-Hop Multimodal Retrieval-Augmented Generation"
_ICLR.cc/2026/Conference — ICLR 2026 Conference Withdrawn Submission_

### Official Review · Reviewer_Xtjw · 2025-10-28

**Soundness:** 3
**Presentation:** 3
**Contribution:** 3
**Rating:** 6
**Confidence:** 2

**Summary:**

The core contribution of this work is the introduction of a new benchmark, EgoFact, which is designed to evaluate the capabilities of multi-hop, multi-modal Retrieval-Augmented Generation (RAG) models. EgoFact specifically focuses on the integration of video understanding with textual knowledge , requiring models to integrate information from different modalities across multiple reasoning steps. The study posits that while significant progress has been achieved in text-based RAG, substantial challenges remain in video understanding , particularly in scenarios lacking auxiliary text.

**Strengths:**

1. The paper introduces a novel multi-hop reasoning method that enables cross-modal retrieval, marking a significant departure from previous approaches.

2. It provides a systematic diagnosis demonstrating that existing video-RAG systems underperform not due to poor retrieval, but because of weak visual grounding capabilities .

3. A 'localization-first' method is proposed, which first localizes relevant video segments before generation, leading to significant performance gains .

**Weaknesses:**

1. Limited Scope and Scalability of the Benchmark: The scale of the EgoFact benchmark is modest (127 video clips , 396 queries ), which may limit the statistical robustness and generalizability of the findings. Furthermore, the corpora are drawn from narrow domains: videos are sourced only from a specific subset of Ego4D (e.g., repair scenarios) , and textual knowledge comes exclusively from WikiHow. This results in a limited variety of evaluated tasks, focusing heavily on procedural knowledge.

2. Limited Definition of "Multi-hop": The paper defines "2-hop" reasoning as a single, fixed pattern: "identify object in video -> retrieve textual fact". While this is a valid form of cross-modal reasoning, it fails to cover more complex reasoning chains (e.g., video-to-video, or synthesizing multiple sources).

3. Inconsistent Citation Format: The paper's citation format is inconsistent. For example, the in-text citations on lines 65-73 do not use the \citep style, which is inconsistent with other citations in the document.

**Questions:**

/

---

### Official Review · Reviewer_cT7T · 2025-10-30

**Soundness:** 1
**Presentation:** 2
**Contribution:** 2
**Rating:** 2
**Confidence:** 4

**Summary:**

The paper proposes EgoFact, a benchmark for multi-hop (1-hop/2-hop) and multimodal (video + text) RAG. The questions are divided into Local and Temporal types, with a total of 396 multiple-choice questions based on egocentric video clips from Ego4D. The authors also present an improved RAG pipeline derived from VideoRAG, introducing modifications to better handle multi-hop multimodal reasoning.

**Strengths:**

1. The problem dimensions are well defined, explicitly covering the different reasoning categories (1-hop/2-hop, Local/Temporal), which effectively distinguish between pure visual understanding and visual reasoning augmented with external knowledge.

2. The analysis is diagnostic in nature, separating retrieval success rate from evidence comprehension ability, revealing that the main bottleneck lies in video understanding rather than retrieval.

**Weaknesses:**

1. The claim of **being the first multi-hop multimodal benchmark is not fully accurate**. Previous works such as Dyn-VQA [1] and WebQA [2] have already explored multi-hop reasoning across modalities, which focus on image and text modalities. The paper lacks of clearly discuss how EgoFact differs from these existing multimodal RAG benchmarks (Dyn-VQA [1], WebQA [2], and M2RAG[3])  in terms of task design, modality coverage, and reasoning depth, instead of presenting it as the first of its kind.

[1] Benchmarking Multimodal Retrieval Augmented Generation with Dynamic VQA Dataset and Self-adaptive Planning Agent

[2] WebQA: Multihop and Multimodal QA

[3] Benchmarking Retrieval-Augmented Generation in Multi-Modal Contexts

2. The **dataset size** is too small (less than 400 samples), and the benchmark only performs local retrieval **without reporting the actual knowledge base size**. If the candidate pool is limited and contains few confusing distractors, the retrieval task becomes much easier and cannot reflect realistic retrieval difficulty.


3. The paper does not provide **the distribution of samples across different domains or scenarios**. Without clarity on domain diversity, it is unclear whether the benchmark can comprehensively evaluate multimodal reasoning beyond a narrow set of scenarios.


4. On the video side, GPT-4o-mini is used to generate frame-by-frame captions, which are then concatenated into clip descriptions and used to build temporal timelines and keywords. Subsequently, OpenAI’s Text-Embedding-3-small model is used for embedding and clustering. Since the entire video corpus construction pipeline **relies on the same family of APIs**, the resulting data shares a highly consistent linguistic style and vocabulary, which may **give distributional advantages** to models using the same or similar APIs.


5. The overall data construction process heavily depends on closed-source LLMs, introducing systematic bias. The author should also provide the details of how they conduct quality examination of samples for the Reliability check, whether it is automatic or human-annotated.


6. The paper only reports accuracy without providing **human upper bounds**. It also **omits essential diagnostic metrics** such as retrieval hit rate @k, localization IoU, which are important for identifying performance bottlenecks.


7. **Computation and cost analysis are missing**. The benchmark construction and evaluation involve extensive API calls, yet the paper only mentions a single A100 GPU environment without disclosing token counts, API costs, or time overhead, making it difficult to assess scalability and reproducibility.

**Questions:**

Please refer to the weakness sections.

---

### Official Review · Reviewer_G2wr · 2025-10-31

**Soundness:** 2
**Presentation:** 2
**Contribution:** 2
**Rating:** 2
**Confidence:** 4

**Summary:**

This paper introduces EgoFact, a benchmark for evaluating multimodal retrieval-augmented generation (RAG) on egocentric videos with both single-hop and multi-hop reasoning. The benchmark is conceptually well-motivated and carefully constructed. The accompanying “localization-first” method is a straightforward yet effective refinement that modestly improves video understanding performance. Overall, the paper identifies a valuable research gap and provides a clean empirical analysis.

**Strengths:**

1. The paper tackles a relevant problem of multimodal RAG involving video and text reasoning.
2. The diagnostic breakdown into retrieval and understanding components is clear and systematic.
3. The proposed localization-first step is simple yet practical, showing small but consistent improvements.

**Weaknesses:**

While the goal of advancing multi-hop multimodal RAG is commendable, this paper suffers from significant flaws in its benchmark design, experimental analysis, and the novelty of its proposed contribution.
1. The paper claims to address "multi-hop" reasoning. However, the query examples provided (e.g., in Table 4 ) suggest a very simple form of two-hop reasoning: (Hop 1) Identify an object or action in a video; (Hop 2) Retrieve an external factual statement about that object. This task seems to be a straightforward combination of object recognition and text-based factual lookup, rather than a complex reasoning process involving multiple logical steps or evidence chaining, which is what "multi-hop" typically implies in QA literature.
2. The benchmark's scale is another limitation. The entire video corpus consists of only 127 video clips, and the dataset contains 396 queries in total. Findings based on such a limited set of examples are less likely to be generalizable.
3. The authors use GPT-4o-mini to generate video captions. Given the extremely small sample size (127 clips), manual annotation by humans would have been not only feasible but also far superior for establishing a high-quality, reliable ground truth. Using a model to generate these intermediate representations introduces unnecessary noise and confounds the evaluation.
4. The paper's main claim that video understanding is the main bottleneck, not retrieval, is not convincingly supported and is even contradicted by its own results. Figure 4 shows that even for "Correct-answer cases", the video understanding success rate is dismally low. If correct answers don't require "understanding" (as per the paper's own metric), then the claim that "understanding" is the bottleneck is unfounded. Furthermore, Figure 3 shows that for incorrect answers, the video retrieval success rate for Local 2-hop, Temporal 1-hop, and Temporal 2-hop are all noticeably lower than for the correct cases, thus directly challenges the conclusion that retrieval is not a major bottleneck.
5. The proposed "localization-first" solution lacks novelty. The method involves sliding a 5-second window across the retrieved clip and selecting the segment with the highest mean query-video similarity using MetaCLIP. This is a standard, straightforward application of similarity matching and cannot be considered a novel technical contribution. It is a baseline-level heuristic, not a new framework.

Minor Concerns: The paper is unnecessarily verbose. Many sections are repetitive and "padded" with explanations that add little substance, making the paper difficult to read and obscuring the (already limited) core contributions.

**Questions:**

1. The paper's central claim is that video understanding, not retrieval, is the main bottleneck. However, Figure 4 shows that even in "Correct-answer cases," the video understanding success rate is very low (e.g., 40%). How can the model arrive at the correct answer if it fails the "understanding" check more often than not? Conversely, Figure 3 shows that for incorrect answers, retrieval success is often poor (50%). Could the authors please elaborate on how they reconcile this data with their primary claim?
2. The video corpus is quite small (127 clips). Why was GPT-4o-mini used to generate captions when high-quality human annotation would have been feasible at this scale? Could the authors comment on the validation process for these model-generated captions and address the concern that this introduces unnecessary noise, confounding the benchmark's reliability?
3. The paper claims to evaluate "multi-hop" reasoning. However, the examples in Table 4 suggest a simple two-step task: 1) identify an object in a video, 2) look up a fact about it. Could the authors justify why this constitutes "multi-hop reasoning" in the context of QA literature? Furthermore, how can the findings from such a small-scale benchmark (127 clips, 396 queries)  be considered generalizable?

---

### Official Review · Reviewer_exeh · 2025-11-01

**Soundness:** 2
**Presentation:** 2
**Contribution:** 2
**Rating:** 2
**Confidence:** 3

**Summary:**

This paper introduces EgoFact, a new benchmark for evaluating multi-hop, multimodal retrieval-augmented generation (RAG) systems, with a particular focus on egocentric videos integrated with external textual sources (e.g., WikiHow). It provides both single-hop and two-hop queries that require combining visual and textual evidence, and includes settings for local versus temporal reasoning. The paper identifies the limitations of existing video-RAG systems and proposes a localization-first framework, which enhances grounding by isolating relevant temporal segments prior to reasoning. Experiments show consistent gains over prior RAG baselines.

**Strengths:**

- The motivation of the paper is strong. EgoFact aims to fill the gap, i.e., multi-hop multimodal reasoning with grounded egocentric videos, beyond unimodal or text-heavy datasets.

- The finding that "retrieval is not the main failure point -- visual grounding is' is interesting.

- The localization-first idea is conceptually simple, easy to reproduce, and yields measurable accuracy gains.

**Weaknesses:**

- Please use the full text in Figure 1 to better illustrate the framework and highlight the unique contribution.

- The paper seems to be in progress, with
  - Limited Benchmark Scale: Only 127 30-second video clips and 396 queries are included. Although well-curated, this scale is small compared to benchmarks like Ego4D or VideoRAG datasets, raising concerns about generalization and statistical robustness.
  - Limited Experiment Evaluation and Related Work: Only GPT-4o-mini is used for generation; more systematic cross-model evaluation (e.g., Gemini, Claude) would strengthen generality
  - Limited Metric: Accuracy on multiple choice tasks may not capture partial correctness or grounding faithfulness; no human evaluation or explainability measure is provided.

**Questions:**

Please refer to the weaknesses above

**Details Of Ethics Concerns:**

The paper mainly uses Ego4D and WikiHow, clear data usage statements (licensing, consent, privacy) can be included in the paper.

---

### Note · Authors · 2025-11-13

**Comment:**

We thank all reviewers for their thoughtful feedback and for recognizing the importance of the research direction (exeh, G2wr, cT7T, Xtjw). We acknowledge that the current benchmark is limited in scale and therefore does not yet provide the level of robustness and coverage expected of a widely adopted evaluation suite. We also recognize the need to more clearly differentiate our benchmark from existing multimodal and multi-hop baselines to better highlight its unique contribution (cT7T, G2wr). We will continue to refine the design, expand the dataset, and improve the analysis, with the goal of scaling up the dataset into a well-established and clearly distinguishable benchmark for multimodal RAG systems. Thank you again for the constructive comments, which we believe will meaningfully strengthen the next iteration of this work.

**Withdrawal Confirmation:**

I have read and agree with the venue's withdrawal policy on behalf of myself and my co-authors.